# Exploring the Diversity and Ecological Dynamics of Palm Leaf Spotting Fungi—A Case Study on Ornamental Palms in Portugal

**DOI:** 10.3390/jof11010043

**Published:** 2025-01-07

**Authors:** Diana S. Pereira, Alan J. L. Phillips

**Affiliations:** Biosystems and Integrative Sciences Institute (BioISI), Faculdade de Ciências, Universidade de Lisboa, Campo Grande, 1749-016 Lisboa, Portugal

**Keywords:** *Arecaceae*, fungal diversity, fungal ecology, genomic fingerprinting, leaf spots, ornamentals, palm diseases, palm fungi

## Abstract

Palm trees (*Arecaceae*) are among the most popular ornamental plants worldwide. Despite extensive research on the fungi associated with *Arecaceae*, the diversity and ecological dynamics of fungi affecting ornamental palms remain poorly studied, although they have significant impact on palm health and economic value. Furthermore, while research on palm fungal diversity has traditionally focused on tropical assemblages, ornamental palms in temperate climates offer a unique opportunity to explore the diversity of palm fungi in non-native habitats. The present study conducted a preliminary assessment of the diversity and ecology of potential phytopathogenic fungi associated with foliar lesions on various ornamental palm host species in Portugal, combining morphological examination, PCR-based genomic fingerprinting, and biodiversity data analysis. The examination of 134 foliar lesions sampled from 100 palm trees resulted in a collection of 2064 palm leaf spotting fungi (PLSF), representing a diverse fungal assemblage of 320 molecular operational taxonomic units (MOTUs) across 97 genera. The overall fungal community composition revealed a distinct assemblage dominated by *Neosetophoma*, *Alternaria*, *Phoma*, and *Cladosporium*, with a profusion of infrequent and rare taxa consistent with a logseries distribution. Significantly positive co-occurrence (CO) patterns among prevalent and uncommon taxa suggest potential synergistic interactions enhancing fungal colonisation, persistence, and pathogenicity. The taxonomic structures of the PLSF contrasted markedly from tropical palm fungi, especially in the prevalence of pleosporalean coelomycetes of the *Didymellaceae* and *Phaeosphaeriaceae*, including recently introduced or not previously documented genera on *Arecaceae*. This novel assemblage suggests that climatic constraints shape the structure of palm fungal communities, resulting in distinctive temperate and tropical assemblages. In addition, the fungal assemblages varied significantly across palm host species, with temperate-native palms hosting more diverse, coelomycete-enriched communities. The present findings highlight foliar lesions as hyperdiverse microhabitats harbouring fungal communities with intricate interactions and a complex interplay of climatic, host, and ecological factors. With climate change altering environmental conditions, the identification of fungi thriving in or inhabiting these microhabitats becomes crucial for predicting shifts in pathogen dynamics and mitigating future fungal disease outbreaks. Understanding these complex ecological dynamics is essential for identifying potential phytopathogenic threats and developing effective management strategies for the health and sustainability of ornamental plants.

## 1. Introduction

Fungi play a pivotal role in the functioning and balance of ecosystems. They form diverse communities across a wide range of environments, performing essential biological functions. Yet, despite their ecological significance, fungi remain woefully neglected and understudied [1,2,3,4]. Recent estimates suggest that between 2.2 and 3.8 million fungal species exist worldwide, with over 90% still undiscovered and unnamed [5,6,7]. This lack of knowledge underscores the need for comprehensive efforts to inventory fungal biodiversity across various ecological niches for future research and long-term ecological sustainability [8,9,10,11,12]. While fungi are primarily recognised as decomposers essential for nutrient cycling, they also function as important symbionts and pathogens, influencing plant health and disease outcomes [13,14,15,16]. Understanding these dynamics is crucial for the effective management of plant hosts of human interest, including agricultural crops, forest trees, and ornamentals.

Among the diverse range of plant hosts, palm trees (*Arecaceae*) hold a particularly prominent position due to their economic value as agricultural crops in international trade and as highly prised ornamentals. Although primarily native to tropical and subtropical regions, these perennial monocotyledonous trees are now distributed globally due to their ornamental use [17,18,19]. In the Mediterranean region, where only the native palm species *Chamaerops humilis* and *Phoenix theophrasti* naturally occur, numerous palm species have become a distinctive feature of urban landscapes [20,21,22]. The health and attractiveness of ornamental plants are key elements in gardening, horticulture, and landscaping aiming to create and maintain functional, aesthetically pleasing spaces. Consequently, anything that diminishes their visual appeal, such as fungal diseases, can significantly reduce their economic and aesthetic value. Furthermore, the increasing demand for exotic palms and their import from overseas markets raise biosecurity concerns. The potential introduction of alien pathogens, allied with their ability to transition from weak to strong virulent agents outside their normal range, poses significant risks to local ecosystems and plant health [23,24,25]. Therefore, *Arecaceae* hosts represent an ideal case study for investigating fungal diversity and understanding the ecological dynamics that impact their health and ornamental value.

As with all plants, palm trees are susceptible to infection by pathogens, particularly fungal pathogens, and are vulnerable to diseases and pests that can cause significant damage, sometimes proving deadly and affecting a wide range of hosts [26,27,28]. Although a huge diversity of fungi is consistently found associated with palms, most are not pathogenic to palms [29,30]. Only a few fungal pathogens are deadly to palms, with most causing minor damage that can be easily managed [26]. Common deadly or potentially deadly fungal diseases in ornamental palms include Fusarium wilt (caused mainly by *Fusarium oxysporum*), Ganoderma root and butt rot or basal stem rot (caused by *Ganoderma* spp.), Gliocladium blight, pink bud rot or pink rot (caused by *Nalanthamala vermoesenii*, formerly *Gliocladium vermoeseni*), and Thielaviopsis trunk rot and related diseases (caused by *Thielaviopsis paradoxa* sensu lato) [31,32,33,34,35,36]. Diseases causing minor damage include diamond scale (caused by *Phaeochoropsis neowashingtoniae*) and other tar spots as well as several leaf spots and seedling, leaf, petiole, and rachis blights [37,38,39,40,41,42,43]. Since ornamental palms are valued for aesthetic purposes, diseases that are of minor consequence in a plantation, such as leaf spots, are a major problem for the ornamental nursery industry and landscapes.

The terms “leaf spot” and “leaf blight” are broadly used in the literature to describe a multitude of symptoms associated with plant leaf diseases. However, this can often lead to confusion and misinterpretation. In the present study, the definitions provided by Elliott [41] were adopted, with the term “foliar lesion” used for simplification. Leaf spots are typically discrete areas of diseased tissue, initially manifesting as water-soaked lesions. The lesions undergo colour changes—often yellow, grey, reddish-brown, brown, or black—and are sometimes surrounded by a distinctive halo or ring of contrasting colour. The distinction between a leaf spot and a leaf blight hinges on the extent of the damage inflicted upon the leaf blade. Leaf blights frequently result from the coalescence of numerous leaf spots, which may merge to form irregularly shaped areas of diseased tissue. As noted by Elliott [41], if the spots remain separated by green tissue, the disease is classified as a spot, but if they merge into larger, irregularly shaped areas, it is classified as a leaf blight.

Most leaf spots and leaf blights affecting palms are caused by fungi and are ubiquitous in both plantations and landscapes [26,27,28]. However, these diseases tend to be more problematic in younger palms [44]. Numerous fungal pathogens can be associated with palm leaf diseases, yet their symptoms are relatively similar. As a result, it is often not possible to allocate a specific name to a particular foliar symptom until the pathogen is identified. Potential fungal causes of palm leaf diseases, which have a wide host range across *Arecaceae* and other plant families, include the genera *Alternaria*, *Annellophora*, *Bipolaris*, *Botryosphaeria*, *Botrytis*, *Calonectria* (syn. *Cylindrocladium*), *Cercospora*, *Colletotrichum*, *Curvularia*, *Exserohilum*, *Fusarium*, *Gloeosporium*, *Graphiola*, *Nalanthamala*, *Pestalotiopsis* and related genera, *Phaeotrichoconis*, *Phyllachora*, *Pseudocercospora*, *Sclerotium*, *Seiridium*, and *Stigmina* [27,41]. Although the complete host range for each pathogen is unknown, it is assumed to be broad, with most palm species likely susceptible to at least one of them [41].

While palm endophytes and saprophytes have been intensively studied, relatively few studies on palmicolous fungi have addressed plant pathogens (for a literature review, see Pereira and Phillips [45]). Additionally, the great majority of these studies focused primarily on the host perspective, consisting more of phytosanitary measures rather than ecological and/or descriptive taxonomy studies of the phytopathogenic agents [26,31,32,33,34,35,39,40,41,42]. A review of the literature reveals a limited number of reports that have examined the assemblage of fungi associated with palm leaf diseases. Most of these studies have taken a descriptive taxonomic approach, providing valuable insights into the diversity and distribution of these fungi. Hyde and colleagues were among the first to recognise that these fungi are an important and understudied assemblage. Following a commentary on the diseases of palms by Chase and Broschat [46], later revised by Elliott et al. [26], a remarkable diversity of fungi associated with palm leaf diseases has been documented across various palm hosts and tropical regions. A new assemblage of fungi distinct from those commonly associated with foliar diseases was revealed, including species of *Astrosphaeriella*, *Capitorostrum*, *Everhartia*, *Maculatifrondis*, *Maculatipalma*, *Myelosperma*, *Oxydothis*, *Phyllosticta*, and *Pseudospiropes*. These studies highlight that, despite their importance, the fungal assemblages associated with palm leaf diseases remain significantly understudied [47,48,49,50,51,52,53,54,55,56,57,58,59,60]. While most of these studies focused on identifying the cause of the spots or blights, only one treated these symptoms as possible fungal communities. Fröhlich et al. [57] reported the biodiversity of fungi associated with palm leaf diseases in the rainforests and nurseries of North Queensland, using an integrated culture-dependent approach to isolate fungal assemblages from diseased leaf tissues.

The detrimental impacts of fungal pathogens on the ornamental palm industry have long been known [61]. Although the main fungal pathogens affecting palm trees have virtually all been identified, minor pathogens such as those causing foliar diseases remain poorly known and are continuously being reported, representing a glaring example of important research that has yet to be conducted, e.g., refs. [62,63,64,65,66,67,68,69,70,71,72,73,74,75]. Douanla-Meli and Scharnhorst [76] recently drew attention to the potential of palm foliage to act as a pathway for the dispersal of potentially pathogenic botryosphaeriaceous species. This is a cause for phytosanitary concern in destination countries, which are experiencing a steady increase in demand for tropical palm trees for ornamental use [20]. In Portugal, the identification of some biotic diseases affecting ornamental palms has been ongoing since 1998. These studies have been conducted with the objective of acquiring knowledge on phytosanitary problems that depreciate plants in green spaces. Among these, the most commonly reported fungal diseases are the pink rot and Thielaviopsis trunk rot [77,78,79]. Currently, except for these scattered studies, almost nothing is known about the diversity of fungi on palm trees in Portugal.

Palm trees are undoubtedly a key resource in the ornamentation of parks and gardens as well as shade trees in Portuguese cities. In this regard, a preliminary assessment of the diversity and ecology of potential phytopathogenic fungi associated with ornamental palm trees in Portugal was conducted, focusing on those fungi associated with foliar diseases. The present study aims to (1) identify the most prevalent mycota of palm leaf spotting fungi (PLSF) in ornamental palm trees in Lisbon, Portugal; (2) qualitatively and quantitatively examine the diversity of fungal communities associated with palm foliar lesions; (3) investigate the ecological dynamics of the fungal communities by examining composition, abundance distribution, and co-occurrence patterns; (4) evaluate the degree of genetic diversity in the most prevalent mycota using PCR-based genomic fingerprinting methodology; and (5) access host-related differences in the composition and diversity of the fungal communities recovered from different palm host species, with a particular focus on the differences between typical temperate and tropical palm hosts.

## 2. Materials and Methods

### 2.1. Specimen Collection and Examination

Diseased leaf segments and leaflets with foliar lesions were collected between September 2018 and May 2021 from ornamental palm trees in parks, gardens, cityscapes, and indoor environments in six civil parishes of Lisbon (Alvalade, Areeiro, Marvila, Olivais, Parque das Nações, and São Vicente) and one of Oeiras (União das Freguesias de Algés, Linda-a-Velha e Cruz Quebrada-Dafundo), Portugal (Figure 1). A total of 100 palm trees belonging to 9 distinct species were sampled, including *Chamaedorea elegans*, *Chamaerops humilis*, *Chrysalidocarpus lutescens*, *Phoenix canariensis*, *P. dactylifera*, *P. reclinata*, *P. roebelenii*, *Trachycarpus fortunei,* and *Washingtonia filifera* (Figure 2). Plant material was transported to the laboratory in paper envelopes and examined with a Leica MZ9.5 stereo microscope (Leica Microsystems GmbH, Wetzlar, Germany) for observations of lesion morphology and associated fungi. Specimens were air-dried and stored in a cardboard box at room temperature (18–20 °C). Morphological details of foliar lesions were observed on both adaxial and abaxial surfaces and used to categorise them into foliar lesion types. Foliar lesion shape, margin topography, colour and evolution over time (when possible), presence or absence of a distinctive border, halo or occasional coalescence, overall distribution on the segment or leaflet and, if applicable, the size were recorded.

### 2.2. Fungal Isolation

Isolations were made on 1/2 strength potato dextrose agar (1/2 PDA) (PDA; BIOKAR Diagnostics, Allonne, France) containing 0.05% chloramphenicol (CPDA) (chloramphenicol; Sigma-Aldrich, Oakville, ON, Canada) to reduce bacterial contamination [81]. Unless stated otherwise, cultures were incubated in continuous ambient light at room temperature (18–20 °C). A comprehensive isolation flowchart (Figure 3) was employed to capture the maximum culturable diversity of palm leaf spotting fungi (PLSF) present in each sample.

Segments and leaflets were first examined with a Leica MZ9.5 stereo microscope (Leica Microsystems GmbH, Germany) for the presence of spore-producing structures. In the absence of any signs of sporulation, the specimens were incubated for 1–3 weeks in a moist chamber and examined daily with the stereo microscope for signs of sporulation.

If fertile hyphomycete conidiophores were present, isolations were made by touching them with a sterile needle and spreading the conidial mass on CPDA. If sporocarps were found, they were first examined microscopically to ascertain whether they were ascomata or conidiomata. In the presence of conidiomata and non-bitunicate ascomata, the foliar lesion was incubated in a moist chamber to stimulate spore release. Conidia or ascospores were then transferred on a sterile needle and spread on CPDA. If, after seven days of incubation, no spores could be seen, the foliar lesion was surface sterilised and a few sporocarps were crushed in a drop of distilled sterile water (DSW) on a flamed microscope slide. The resulting spore suspension was diluted 100 times in DSW to reduce the chances of bacterial contamination [81] and spread on CPDA. In the presence of bitunicate ascomata, the foliar lesion was placed on a drop of DSW in an upside-down plate of CPDA to isolate spores that discharge forcibly and, consequently, impinged on the agar surface. In any of the isolation methods, after a suitable period of incubation, 10 single germinating conidia or ascospores were transferred separately onto 1/2 PDA to ensure the establishment of single-spore, pure cultures as described by Zhang et al. [82].

For all specimens, isolations were also made directly from foliar lesions. Pieces of tissue 1–2 mm^2^ were cut from the edge of the foliar lesions, surface disinfected in 5% sodium hypochlorite for 1 min, rinsed in three changes of sterile water, and blotted dry on sterile filter paper. Five leaf fragments were then plated onto CPDA and incubated until discrete colonies developed (2–7 d). Colonies were subcultured onto 1/2 PDA, and single-spore isolates were subsequently established as described by Zhang et al. [82]. Sterilisation efficiency was assessed by sterilised leaf fragment impressions onto CPDA [83] and by inoculating the last sterile water change used to rinse the leaf fragments onto CPDA. Asymptomatic leaf tissues for each sample subjected to the same sterilisation process were also plated onto CPDA to validate the isolation of fungal records from diseased tissue only.

Cultures of all isolates were stored in duplicate slants of quarter-strength PDA (1/4 PDA) about 2 cm in its widest part in 5 mL graduated microtubes and kept at 4 °C and at room temperature after being covered with 2 mL of sterile mineral oil.

### 2.3. Morphological Observation and Characterisation and Fungal Genera Identification

Pure cultures on 1/2 PDA were periodically examined under a Leica MZ9.5 stereo microscope (Leica Microsystems GmbH, Germany) for the development of microscopic reproductive structures. When necessary, cultures were induced to sporulate by culturing on 2% tap water agar (WA) (Bacteriological Agar Type E; BIOKAR Diagnostics, France) bearing healthy, double autoclaved (two cycles of 20 min, 121 °C and 1 bar with 48 h between each cycle) *Populus* sp. twigs or palm leaf pieces. In addition, cultures were incubated at 25–30 °C under a 12 h near-ultraviolet light/12 h dark cycle. Isolates with cultures that repeatedly failed to sporulate were identified solely by means of molecular analyses (see Section 2.5).

Microscopic structures were mounted in 100% lactic acid and examined by differential interference contrast (DIC) microscopy [84]. Sections 10 µm thick of some reproductive structures were made with a Bright 5040 Rotary Retracting Microtome with a solid-state freezer stage (Bright Instruments, Huntingdon, UK). Observations of micromorphological features were made with Leica MZ9.5 and Leica DMR microscopes (Leica Microsystems GmbH, Germany), and digital images were recorded with Leica DFC300 and Leica DFC320 cameras (Leica Microsystems GmbH, Germany), respectively. Measurements were made with the measurement module of the Leica IM500 Image Management System (Leica Microsystems GmbH, Germany). Photoplates were prepared with Adobe Photoshop CS6 Extended (Adobe, San Jose, CA, USA). Isolates were identified to genus level, when possible, by reference to the available literature, e.g., refs. [85,86,87,88]. All identifications were later presumptively confirmed by molecular analysis (see Section 2.5).

### 2.4. DNA Extraction and Genomic Fingerprinting

Genomic DNA (gDNA) was extracted from cultures of all isolates following a modified and optimised version of the guanidium thiocyanate method described by Pitcher et al. [89] and subjected to microsatellite/minisatellite primed (MSP)-PCR for genomic fingerprinting. Quality and quantity of gDNA were evaluated by 1% (*w*/*v*) agarose gel electrophoresis. Gels were stained with ethidium bromide (2.5 µg mL^−1^), visualised with an Alliance 4.7 UV transilluminator (UVITEC, Cambridge, UK), and the images recorded with Alliance software version 15.15 (UVITEC, Cambridge, UK). The gDNA concentrations were estimated using ImageJ software version 1.52a [90].

Morphological traits and cultural characteristics have often proven insufficient as standalone tools for the identification, differentiation, and classification of fungi. Therefore, additional techniques, such as the use of molecular markers, have been successfully employed to overcome these limitations. Among these, genomic fingerprinting has become a widely used molecular approach for microbial genotypic characterisation due to its high reproducibility and discriminatory power [91,92,93,94]. Genomic fingerprinting encompasses a diverse range of DNA-based methodologies that use DNA polymorphisms to generate practical markers for molecular typing across a range of species of fungi. These methods are extensively applied in fungal taxonomy as they enable the discrimination of isolates from intrageneric to strain levels. The MSP-PCR is a relatively robust and discriminatory PCR-based genomic fingerprinting that analyses the whole genome using single primers to generate DNA fingerprints effective for distinguishing between fungal species and strains. The primers used include small sequences complementary to microsatellites (tandem repeats of 1–5 base pairs) or minisatellites (tandem repeats of a basic motif 10–60 bp long), both of which are ubiquitous and present in multiple copies throughout eukaryotic genomes [95,96,97,98,99,100,101,102,103].

The PCR reactions were carried out with *Taq* polymerase, nucleotides, primers, PCR-water (ultrapure DNase/RNase-free distilled water), and buffers supplied by Invitrogen (Waltham, MA, USA). Amplification reactions were performed in a UNO II Thermocycler (Biometra, Göttingen, Germany). The MSP-PCR profiles were generated following a modified version of the protocol of Ramírez-Castrillón et al. [104], using the following primers: csM13 (5′–GAGGGTGGCGGTTCT–3′), derived from the minisatellite core sequence of the wild-type phage M13 [105,106] and (GTG)_5_ (5′–GTGGTGGTGGTGGTG–3′), which targets simple repetitive DNA microsatellite sequences [107,108]. The PCR reaction mixture for each primer consisted of 1× PCR buffer, 3 mM MgCl_2_, 25 pmol of the respective primer, 0.2 mM of each dNTP, 1 U of *Taq* DNA polymerase, and 50 ng of template gDNA and was made up to a total volume of 25 μL with PCR-water. Reaction mixtures with PCR-water instead of template DNA were used as negative controls. The following cycling conditions were used: initial denaturation at 95 °C for 5 min, followed by 40 cycles of denaturation at 95 °C for 1 min, annealing at 60 °C for 2 min and elongation at 72 °C for 2 min, and a final elongation step at 72 °C for 5 min. The PCR products (5 μL) were resolved by 1% (*w/v*) agarose gel electrophoresis, along with the 1 kb Plus DNA Ladder (Invitrogen, USA), in 0.5× TBE buffer (40 mM Tris, 45 mM boric acid, 1 mM EDTA, pH 8.3) with a constant voltage of 3.4 V cm^−1^ for 5 h. Gels were stained and visualised and the images recorded as previously described.

### 2.5. Genomic Discrimination and Molecular Identification of Fungal Genera

Percentage of similarity between genomic fingerprints was analysed by hierarchical cluster analysis (HCA) conducted using BioNumerics software version 6.6 (Applied Maths, Belgium). The isolates were clustered based on their csM13 and (GTG)_5_ MSP-PCR profiles in consensus dendrograms using the Pearson correlation coefficient (PCC) to generate the similarity matrices and the unweighted pair group method with arithmetic mean (UPGMA) as the clustering algorithm. The reproducibility cut-off level was calculated as the mean value of the reproducibility obtained for each primer independently. For this purpose, and for each primer, 10% of the isolates were randomly selected and their MSP-PCR profiles were repeated [109]. Dendrograms built based on these duplicates were used to estimate the reproducibility cut-off level, 95.05% for csM13 and 95.97% for (GTG)_5_, and to calculate the optimisation and curve smoothing parameters, 1.5% and 5.0% for csM13 and 2.0% and 2.5% for (GTG)_5_, respectively, that better paired the duplicates for each primer. A conservative estimate of the reproducibility cut-off level was set at 95% similarity, above which it is not possible to distinguish between isolates using the approach conducted. The conservative reproducibility cut-off level was used to calculate the isolation redundancy coefficient (IRC). The IRC was calculated considering the isolates belonging to the same genus and originating from the same foliar lesion that were paired above the 95% similarity threshold. These isolations were regarded as redundant, as they were likely to represent independent isolations of the same organism inhabiting a given foliar lesion.

To validate the cut-off levels that could be used to analyse the dendrograms, preliminary assessments were carried out using a set of isolates that had been fully identified based on phylogenetic analyses. Moreover, clustering patterns were also evaluated considering the morphological observation, characterisation, and identification (see Section 2.3). Three cut-off levels were set to evaluate the dendrograms (Figure A1).

A similarity cut-off level of 65% was used to delineate clusters of fungal genera, as this value of percentage of similarity was found to yield clusters of isolates that were congruent with genera identified through morphological analysis. In instances of below 65% similarity, isolates of different genera were found to be clustered together.

A similarity cut-off level of 75% was used to delineate clusters of molecular operational taxonomic units (MOTUs), as this value of percentage of similarity was found to yield clusters of isolates that were congruent with interspecific diversity as assessed by phylogenetic positioning. In instances of below 75% similarity, isolates from different species were clustered together. Within each genus represented by two or more isolates, isolates with a percentage of similarity ≥ 75% were considered to be closely related genetically and were therefore assigned to the same MOTU. A similarity cut-off level of 85% was then used to delineate clusters of genomic types (GTs) to assess the genetic diversity within each MOTU (intraspecific diversity).

In order to ascertain the identity of the fungal genera identified by morphological analyses, representative isolates obtained from different parishes and palm hosts from each cluster formed at 65% similarity were selected for molecular identification by DNA barcoding. For this purpose, part of the cluster of nrRNA genes (ITS), including the nuclear 5.8S rRNA gene and its flanking internal transcribed spacers ITS1 and ITS2, was amplified using the forward primer ITS5 (5′–GGAAGTAAAAGTCGTAACAAGG–3′) [110] and the reverse primer NL-4 (5′–GGTCCGTGTTTCAAGACGG–3′) [111]. PCR reactions were carried out with *Taq* polymerase, nucleotides, primers, PCR-water, and buffers supplied by Invitrogen (USA). Amplification reactions were performed in a UNO II Thermocycler (Biometra, Germany). The PCR reaction mixture consisted of 50–100 ng of gDNA, × 1 PCR buffer, 50 pmol of each primer, 200 μm of each dNTP, 2 mM MgCl_2_, and 1 U *Taq* DNA polymerase and was made up to a total volume of 50 μL with PCR-water. Reaction mixtures with PCR-water instead of template DNA were used as negative controls. The following cycling conditions were used: initial denaturation at 95 °C for 5 min, followed by 40 cycles of denaturation at 94 °C for 1 min, annealing at 52 °C for 30 s and elongation at 72 °C for 1.5 min, and a final elongation step at 72 °C for 10 min. The PCR products were checked on 1% (*w/v*) agarose gel electrophoresis stained with ethidium bromide and visualized on a UV transilluminator to assess PCR amplification. The amplified PCR products were purified and sequenced by Eurofins Genomics (Konstanz, Germany) in the forward direction using the primer ITS5. The newly generated ITS region sequences were used to obtain a presumptive identification of each selected isolate based on BLASTn searches against GenBank. The genera were depicted from the closest-hit taxa available in GenBank. A similar approach was used to identify all isolates whose cultures failed to sporulate as well as representatives of highly morphological similar isolates that had been clustered previously based on their genomic fingerprints.

### 2.6. Biodiversity and Ecological Data Analysis and Statistical Inference

Morphological analysis usually ensures identification of fungi to genus level. However, it is unreliable for identification at lower taxonomic ranks, as morphological characters may not always provide accurate groupings within a natural evolutionary framework [112]. Accordingly, although biodiversity is commonly measured at the species level, this study assessed it at the genus level, given that most of the isolates were only presumptively identified. Furthermore, considering that genetic diversity has been evaluated through genomic fingerprinting, structural diversity is also expressed in terms of MOTU. While preliminary analyses indicated a congruence between MOTU and interspecific diversity, most of the isolates have not been identified using contemporary molecular methods, i.e., DNA barcoding and phylogenetic positioning, to ensure species level identification [113]. To prevent the report of uncertain results, this study has refrained from referring to species in all biodiversity and ecological inferences, referring instead to MOTUs. In all the following equations, the terms “taxon” and “taxa” represent either genera or MOTUs, depending on the taxonomic level at which diversity is being measured.

#### 2.6.1. Population Status and Structural Diversity Measures

Fungal records are presented in terms of their percentage of abundance of occurrence (AO) and frequency of occurrence (FO), which were computed following Equations (1) and (2).
(1)$$\mathrm{AO}=\frac{\mathrm{number}\;\mathrm{of}\;\mathrm{records}\;\mathrm{of}\;a\;\mathrm{given}\;\mathrm{taxon}}{\mathrm{total}\;\mathrm{number}\;\mathrm{of}\;\mathrm{records}\;\mathrm{of}\;\mathrm{all}\;\mathrm{taxa}}\times 100$$


(2)
$$\mathrm{FO}=\frac{\mathrm{number}\;\mathrm{of}\;\mathrm{samples}\;\mathrm{in}\;\mathrm{which}\;a\;\mathrm{given}\;\mathrm{taxon}\;\mathrm{occurred}}{\mathrm{total}\;\mathrm{number}\;\mathrm{of}\;\mathrm{samples}\;\mathrm{studied}}\times 100$$


Based on the AO, taxa were grouped into four abundance classes: very frequent (>5%), frequent (>2–5%), infrequent (>1–2%), or rare (≤1%). AO and FO were computed both for the entire community of PLSF and for subsets thereof, based on the palm host from which they were recovered. Additionally, fungal records were analysed in order to express patterns of taxon co-occurrence (CO). The CO was evaluated using the probabilistic model of species CO by Veech [114], as implemented in R Statistical Software version 4.3.1 [115], with the R package *cooccur* version 1.3 [116]. As recommended by Griffith et al. [116], a threshold of <1 was employed to filter the pairs of taxa that are expected to share less than one sample, as inferred from their probabilities of CO. The purpose of the CO threshold is to remove from analysis taxa that simply do not have sufficient occurrence data [114].

Diversity is defined as a measure of the complexity of structure in an ecological community. This comprises two distinct attributes, which are evaluated by a quantitative expression of community structure (biodiversity measures), namely, species richness (S, number of species) and species evenness or equitableness (E, how similar species are in their abundances) [117]. To perform a more in-depth analysis of biodiversity, a series of diversity indices have been calculated in order to express and correlate both S (which, in this study, represents the number of genera or MOTUs) and E. Diversity of the communities of PLSF was quantified and compared according to Magurran [118] using Simpson diversity index (D), Shannon diversity index (H′), Shannon evenness index (J′), and logseries alpha (α) as richness–evenness indices and Margalef’s diversity index (D_Mg_) and Menhinick’s diversity index (D_Mn_) as species richness indices.

Simpson diversity index [119] was computed from Equation (3). It is referred to as a dominance measure since it is strongly affected by the abundance of the most common species [117]. As a measure, D reflects the probability of randomly choosing two individuals that belong to the same species [120].
(3)$$D=1-\sum_{i=1}^{S}\frac{n_{i}\left(n_{i}-1\right)}{N\left(N-1\right)}\;,$$
where (here and throughout) *n_i_* is the number of individuals (isolates) in the *i*th taxon, *N* is the total number of individuals, and *S* is the taxon richness (total number of taxa). Simpson diversity index varies from 0 to 1 and increases as the diversity decreases; thus, in this study, it was computed in its complement form (1 − D) to ensure that the index increases with increasing diversity [118].

Shannon diversity index [121] was computed from Equation (4). It represents a measure of the average degree of uncertainty in predicting the specific identity of an individual randomly chosen from a collection of *S* taxa and *N* individuals [117].
(4)$$H^{\prime }=\sum_{i=1}^{S}\left(p_{i}log{\;p}_{i}\right)\;,$$
where (here and throughout) *p_i_* is the proportional abundance of the *i*th taxon computed from *p_i_* = *n_i_*/*N*, and log is the common logarithm (base 10). Shannon diversity index increases as the number of species increases and as the distribution of individuals among species becomes more even [118].

Shannon evenness index [122] was computed from Equation (5). It quantifies the extent to which a certain community displays the maximum possible diversity given the observed richness [117].
(5)$$J^{\prime }=\frac{H^{\prime }}{LogS},$$
where H′ is the value of the Shannon diversity index. Log *S* is equivalent H′_max_ (maximum diversity), considering Equation (5) for Shannon diversity index and, therefore, J′ varies between 0 and 1 and could be used as a measure of entropy in the distribution of individuals among taxa [123].

Logseries alpha [124] was computed from Equation (6). It corresponds to the parameter alpha of the Fisher’s logarithmic series model and has been shown to be an appropriate index of diversity even when the logseries distribution is not the best descriptor of the underlying species abundance pattern [125,126].
(6)$$\alpha =\frac{N(1-x)}{x},$$
where *x* is estimated from the iterative solution of *S*/*N* = [(1 − *x*)/*x*] × [−ln (1 − *x*)].

Margalef species richness index (D_Mg_) [127] and Menhinick species richness index [128] were computed from Equations (7) and (8), respectively. These species richness indices have been developed to account for the fact that species richness is highly dependent on the collection effort [123].
(7)$$D_{Mg}=\frac{\left(S-1\right)}{LnN}\;,$$
(8)$$D_{Mn}=\frac{S}{\sqrt{N}}\;,$$
where Ln is the natural logarithm (base e).

In addition to the aforementioned diversity measures, fungal isolation was also used as a means of quantifying fungal richness, thereby illustrating the extent of multiple colonisations across samples derived from different host species. For this purpose, isolation rate (IR), i.e., the number of fungal records per sample, was computed following Equation (9).
(9)$$IR=\frac{number\;of\;fungal\;occourrences\;in\;a\;given\;host\;species}{number\;of\;samples\;examined\;for\;a\;given\;host\;species}$$

#### 2.6.2. Accumulation Curves and Abundance Distributions

Species abundance distribution (SAD) describes the relationship between diversity and the structural organisation of an ecological community. A characteristic pattern arises when the number of species and their relative abundances within a community are plotted. Therefore, SAD models are used to describe the distribution of commonness and rarity within an ecological community [129,130,131]. In the present study, SADs were constructed to assess the pattern of genus and MOTU abundances obtained from the community of PLSF and are referred to as genus abundance distribution and MOTU abundance distribution, respectively. SADs were generated by plotting the abundance of occurrence of each genus or MOTU in the sample set on a logarithmic scale against genus or MOTU, respectively, ranked from most to least abundant. Five SAD models that have been widely used in the ecological literature, including the broken stick [132], geometric series [133], logseries [124], lognormal [134], and neutral model [135,136] (for theoretical details, see Magurran [118] and McGill [137]), were selected to fit the MOTU abundance distribution, using maximum likelihood estimation. The MOTU abundance distribution was selected for fitting purposes in preference to the genus abundance distribution, as the former encompasses more information and provides a more suitable data frame for the application of theoretical SAD models. More importantly, all these models are founded upon particular ecological assumptions, which can assist with understanding specific processes involved in the assembly of foliar lesion communities. Models were fit in R Statistical Software version 4.3.1 [115] using the R package *sads* version 0.6.3 [138]. Fitting was carried out on the raw data, i.e., the vector of the number of records of each MOTU. As the choice of goodness-of-fit (GOF) test has been found to influence the results [139], two approaches were used to assess the GOF of the distributions obtained from the five SAD models, according to Magurran [118]. These included the Pearson’s chi-squared (*χ*^2^) and Kolmogorov–Smirnov (*D*) statistical tests at 95% confidence level. For this purpose, the observed values were compared with the predicted values for each abundance class, where the predicted values were derived from the statistical fitting of the SAD. Results were considered significant if *p*-value > 0.05, which indicates that a given predicted distribution conforms with the observation [118]. Sole reliance on traditional GOF metrics has been criticised as unreliable [139] and, therefore, information theoretic approaches (strength of fit, SOF), namely, the Akaike information criterion (AIC) corrected for small sample sizes (AICc) and the Bayesian information criterion (BIC), were also used to compare the fit of the different models. The lowest AICc and BIC values were taken to represent the best model to describe the MOTU abundance distribution, provided that the ΔBIC or ΔAICc to the next best model was >2 [140]. While statistical approaches to assess GOF are evidently more objective, visually analysing the fitted distribution can provide valuable information [141]. Thus, the MOTU abundance distribution was plotted along with each fitted SAD model, and the fit was visually inspected to determine if there were particular parts of the observed distribution that were better explained by different SAD models.

Species accumulation curves (SACs) are often used as a tool to evaluate the sampling effort and to compare the diversity of different sampling sites [118,142]. In the present study, SACs were constructed to estimate whether the sampling was thorough by plotting the cumulative number of genera (genus accumulation curves) or MOTUs (MOTU accumulation curves) recovered against the number of samples examined (selected at random) for each palm host and for the overall sample set. SACs and their statistical expectations (rarefaction curves) were generated based on randomised resampling (1000 permutations) of foliar lesion observations. Nonparametric estimators were used to estimate extrapolated taxon richness (Ŝ) that might exist if all possible taxa were found, including Chao 1 [143], first-order jackknife, second-order jackknife [144,145,146,147], abundance-based coverage estimator (ACE) [148,149,150], and bootstrap [147] (for theoretical details, see Magurran [118] and Gotelli and Colwell [151]). In addition, an asymptotic curve-fitting approach was used to extrapolate taxon richness. For this purpose, the SACs generated for the overall sample set were fitted to five different nonlinear self-starting species accumulation models widely used in the literature: the asymptotic regression [152], logistic function [153], Lomolino function [154], Michaelis–Menten function [155], and Weibull distribution [156] (for theoretical details, see Tjørve [157,158] and Dengler [159]) to predict their asymptote, which was used as an estimate of taxon richness. The *χ*^2^ statistical test was used to determine the GOF of each model, and the AICc and BIC were used as the SOF measures. The lowest AICc and BIC values were taken to represent the best model to describe the SACs, provided that the ΔAICc and ΔBIC to the next best model was >2. Only the asymptote of the best model(s) was used as an approximation of the extrapolated taxon richness. The coefficient of variation (CV) in species abundances, which is defined as the ratio of the standard deviation to the mean of species abundances in a community, was also computed. The CV parameter is used to characterise the degree of heterogeneity among species abundances. A higher CV reflects a greater degree of heterogeneity, whereas a CV of zero occurs only when all species have identical abundances, signifying a completely homogeneous assemblage [160]. All analyses were computed in R Statistical Software version 4.3.1 [115] using the following R packages: *vegan* version 2.6-6.1 [161], *SpadeR* version 0.1.1 [162], and *SPECIES* version 1.2.0 [163].

#### 2.6.3. Exploratory Analysis of the Assembly of Fungal Communities

##### Statistical Analyses of Composition and Diversity of Fungal Assemblages

Permutational multivariate analysis of variance (PERMANOVA) was carried out to assess differences in the composition of communities of PLSF among samples (with at least five fungal records) as a function of palm host species, parishes, or foliar lesion types (factors). PERMANOVA analyses were carried out on square-root-transformed data using similarity matrices based on Bray–Curtis dissimilarity [164] with 999 permutations performed on residuals under a reduced model. The data were square-root-transformed in order to scale down the influence of highly abundant taxa and thereby enhance the impact of less abundant ones in the analysis of similarity between fungal communities. The assumption of homogeneity of multivariate dispersion was primarily checked for each factor. Post hoc pairwise multiple comparisons were performed to explore significant effects using the Bonferroni correction as the *p*-value correction method.

Taxon richness and diversity of samples (with at least five fungal records) by host species were compared using the Kruskal–Wallis test (*H*) [165]. Post hoc pairwise multiple comparisons were performed to explore significant differences using the Dunn test [166,167] and the Bonferroni correction as the *p*-value correction method. For this purpose, diversity was measured using the Shannon diversity and evenness indices and the logseries alpha (see Section 2.6.1).

All analyses were computed in R Statistical Software version 4.3.1 [115] using the R packages *vegan* version 2.6-6.1 [161] and *stats* [115]. Analyses of the overlap between fungal assemblages documented in each host species were also conducted through the construction of Venn diagrams using the R packages *ggplot2* version 3.5.1 [168] and *ggVennDiagram* version 1.5.2 [169].

##### Principal Component Analyses and Hierarchical Cluster Analyses of Fungal Assemblages

Principal component analysis (PCA) was carried out to visualise differences in communities of PLSF among samples (with at least five fungal records) collected from different host species. The PCA analysis was performed on square-root-transformed AO of all MOTUs.

An HCA was conducted to assess the similarity among communities of PLSF obtained from different palm host species. Pairwise distances were estimated with Euclidean distance index to generate the dissimilarity matrices, and dendrograms were constructed by the UPGMA as the clustering algorithm. Dendrograms were generated based on the AO of MOTUs, and clusters were determined by visual inspection. GOF of the dendrograms was evaluated by means of the cophenetic correlation coefficient (c) [170].

The PCA and HCA were computed in R Statistical Software version 4.3.1 [115] using the R packages *factoextra* version 1.0.7 [171] and *cluster* version 2.1.4 [172], respectively.

## 3. Results

### 3.1. Disease Symptoms and Foliar Lesion Types

A clearly defined sample unit is of paramount importance in biodiversity studies, as it ensures the diminishing of potential bias that could compromise the accurate interpretation and establishment of ecological patterns and trends. In the present study, the sample unit is defined as a set of discrete, localised spots on diseased host leaves, which have been termed foliar lesions. To characterise each foliar lesion, a number of morphological characteristics were assessed, and a randomly selected representative spot was chosen for analysis of its culturable community of palm leaf spotting fungi (PLSF). A fungal record is, thus, defined as the isolation of a fungal taxon on a given sample.

All the palm host species examined supported a considerable wealth of diseased foliar tissue from which representative lesions were selected for further analysis. A total of 134 foliar lesions from 100 palm trees belonging to 9 distinct species, *Chamaedorea elegans*, *Chamaerops humilis*, *Chrysalidocarpus lutescens*, *Phoenix canariensis*, *P. dactylifera*, *P. reclinata*, *P. roebelenii*, *Trachycarpus fortunei,* and *Washingtonia filifera*, were collected and examined for associated fungi (Table 1).

Four distinct foliar lesion types were defined according to their general abaxial and adaxial surface morphological characteristics: tip die-back (TDB), large leaf spots (LLSs), small leaf spots (SLSs), and pinpoints and punctuations (PPs). Figure 4 and Figure 5 present illustrative schemes and representative samples of each foliar lesion type, respectively.

TDB foliar lesions were extensive necrosis on leaves beginning at their apices and progressing towards their bases. Morphologically, the foliar lesions were irregular in shape, with rounded ends, and exhibited different lengths depending on the age of the lesion. They were identical on both surfaces, occasionally paler on the abaxial surface, pale grey or pale brown to brownish in colour, usually becoming greyish and fragile or brittle, often with a dark brown border (<1–3 mm wide, rarely <5 mm wide), and rarely surrounded by an inconspicuous to visible yellowish, light brown, or light green halo. Mature lesions usually contained multiple immersed, semi-immersed, or erumpent fruiting bodies, typically found on both surfaces, although they were more prevalent on the adaxial surface (Figure 4 and Figure 5A–C).

LLSs were extensive blotches, rarely with necrotic tissue, that were not exclusively associated with foliar apices and were randomly distributed on the leaves. Morphologically, the LLSs were ellipsoidal to irregular in shape, measuring over 10 cm in length and often larger due to coalescence. The lesions had rounded to angular ends and were identical on both surfaces, occasionally paler on the abaxial surface, with a pale brown to brown, yellowish to greyish centre and a dark brown border (<1–2 mm wide, occasionally thicker, darker, and up to 10 mm wide). These lesions rarely became fragile or brittle and were occasionally surrounded by an inconspicuous pale whitish, yellowish, or brownish halo. Frequently, they displayed conspicuous concentric growth lines that often acquired the appearance of overlapping layers. Infrequently, mature lesions exhibited immersed to semi-immersed fruiting bodies, which often developed on sunken tissue regions (Figure 4 and Figure 5D–F).

The SLSs were small, discrete lesions that occasionally coalesced into larger spots and were randomly distributed on the leaves but rarely found along the main vein. Morphologically, SLSs were subglobose to broadly ellipsoidal or fusiform in shape, often becoming irregular, measuring 1–5 cm in length (sometimes larger than 5 cm), with rounded to angular ends. They were identical on both surfaces, though they occasionally appeared paler on the abaxial surface, displaying a centre with a range of colours from brown-grey, pale brown, yellowish to greyish, and sometimes blackish, and a dark brown border (up to 1 mm wide). These lesions rarely became brittle on the abaxial surface and were frequently surrounded by a conspicuous (though sometimes inconspicuous), pale brown or yellowish halo. Infrequently, mature lesions exhibited multiple immersed or semi-immersed fruiting bodies, predominantly on the adaxial surface. Additionally, there were instances where conidiophores were observed in association with stomatal apertures on the abaxial surface (Figure 4 and Figure 5G–I).

PPs were minute or very small, discrete lesions that occasionally coalesced into slightly larger spots. These were dispersed and randomly distributed along the foliar tissue but were rarely found along the main vein or leaf margins. The lesions were morphologically diverse, displaying oval, globose to subglobose, or ellipsoidal to fusiform shapes, and they were often irregular, measuring less than 1 cm in length (rarely up to 1.5 cm) with rounded to angular ends. They were identical on both surfaces and displayed a pale to yellowish hue, as well as brown to blackish shades. PPs were typically surrounded by a conspicuous or inconspicuous brownish, yellowish, or light green halo (Figure 4 and Figure 5J–L).

The occurrence of TDB foliar lesions was more prevalent (49% of all foliar lesions collected) compared with the other foliar lesion types (16% for LLSs, 18% for SLSs, and 17% for PPs). Additionally, TDB foliar lesions were generally the most prevalent foliar lesion type in palm hosts from which a considerable number of samples were collected (>10). However, no association between the type of foliar lesion and the host species was observed as, in general, all foliar lesion types were present in all palm host species (Table 1).

### 3.2. Overall Structural Diversity

A collection of 2064 fungal records associated with foliar lesions of ornamental palms was established. A total of 320 molecular operational taxonomic units (MOTUs) in 97 genera were recorded (Table 2). This included 160 MOTUs in 47 genera of coelomycetes (56% of all fungal records) and 114 MOTUs in 23 genera of hyphomycetes (39%). Furthermore, 19 genera of ascomycetes were recorded comprising 35 MOTUs (4%). The remaining MOTUs consisted of eight in seven genera of basidiomycetes and three in one genus of zygomycetes, both assemblage types comprising less than 1% of all fungal records.

Patterns of diversity and abundance can be examined qualitatively by comparing the taxonomic distribution of fungal taxa. The percentage abundance and frequency of occurrence of all genera and MOTUs are shown in Table 3 and Appendix A, respectively. The most common genera, with at least 150 fungal records, were *Neosetophoma* (22% of all fungal records), *Alternaria* (18%), *Phoma* (8%), and *Cladosporium* (7%). Accordingly, these genera were regarded as very frequent and had a frequency of occurrence (FO) of over 30%. In some instances, such as *Alternaria* and *Cladosporium*, the FO was over 50%, with both genera documented in more than half of the samples analysed (Table 3).

Eight genera, namely *Fusarium*, *Epicoccum* (each with about 4% of all fungal records), *Neodidymelliopsis*, *Penicillium*, *Didymella*, *Libertasomyces*, *Stemphylium,* and *Sclerostagonospora* (each with about 3%), were regarded as frequent, with an abundance of occurrence (AO) ranging from 49 to 76 fungal records. *Diaporthe*, *Keissleriella*, *Chaetomium,* and *Aspergillus* made up the infrequent group with very similar AOs of around 1%, comprising from 22 to 30 fungal records (Table 3).

The remaining 81 genera were represented by fewer than 20 fungal records and were therefore regarded as rare. Most of these genera corresponded to coelomycetes and were documented in fewer than 10 samples, resulting in a FO below 7%. Furthermore, over 50% of the rare mycota comprised genera represented by two (doubletons) or, especially, one (singletons) fungal record (Table 3).

A similar pattern was observed with regard to structural diversity upon analysis of the data in terms of MOTUs. A total of 307 MOTUs were represented by a maximum of 20 fungal records and were therefore regarded as rare (Appendix A). Consequently, the remaining 3 frequency groups comprised only 13 MOTUs (Figure 6, Appendix A). While four genera were regarded as very frequent, only two of them had MOTUs that were assigned to the same frequency group. These were *Neosetophoma* MOTU 188 (14% of all fungal records) and MOTU 185 (6%) and *Alternaria* MOTU 009 (8%) and MOTU 015 (7%) (Figure 6, Table 3). Furthermore, the FOs of these MOTUs exhibited considerable variation, with values ranging from approximately 20% for *Alternaria* MOTU 015 and *Neosetophoma* MOTU 185 to 25% for *Neosetophoma* MOTU 188 and 51% for *Alternaria* MOTU 009 (Appendix A).

Only 3 MOTUs, namely *Phoma* MOTU 243 (3% of all fungal records) and MOTU 235 (2%) and *Neodidymelliopsis* MOTU 168 (3%), were regarded as frequent, while 6 MOTUs were regarded as infrequent, with AOs ranging from 21 to 34 fungal records (Figure 6, Appendix A). The very frequent and infrequent MOTUs were affiliated with the genera that were regarded as very frequent and four of the genera that were regarded as frequent, namely, *Neodidymelliopsis*, *Epicoccum*, *Stemphylium,* and *Fusarium* (Table 3 and Appendix A). As a result, only 8 of the 97 genera documented exhibited commonly recorded MOTUs and can thus be regarded as the most prevalent PLSF in foliar lesions of palms in Portugal (Figure 6).

As expected, in general, taxa with higher AOs corresponded to taxa with higher FOs (Table 3). However, the relationship between AO and FO was nonlinear, with only a few exceptions warranting further consideration. Given that the data did not follow a normal distribution, the correlation between AO and FO was tested using Spearman’s rank correlation coefficient (*r*_s_). A statistically significant, very strong, positive monotonic correlation was observed between AO and FO for both genera [*r*_s_(95) = 0.99, *p* = 2.20 × 10^−6^] and MOTUs [*r*_s_(318) = 0.99, *p* = 2.20 × 10^−6^]. However, although the correlation remained very strong, it was less pronounced when tested only on the most commonly documented taxa [*r*_s_(95) = 0.84, *p* = 1.93 × 10^−5^ for genera, *r*_s_(318) = 0.86, *p* = 0.02 × 10^−2^ for MOTUs]. This is related to the few exceptions previously noted. For instance, in the very frequent mycota, *Neosetophoma* exhibited the highest AO but the lowest FO; conversely, *Cladosporium* was the very frequent genus with the lowest AO, but it ranked as the genus with the second-highest FO, with *Alternaria* being the genus with the highest FO but the second-highest AO. A similar result was observed in the frequent mycota, with *Penicillium* and *Stemphylium* emerging as the genera with the highest and second-highest FOs, respectively, despite not occupying the top three positions in the AO; conversely, *Neodidymelliopsis* ranked as the genus with the third-highest AO but the lowest FO (Table 3). Comparable outcomes can also be observed in the assemblage of the most documented MOTUs (Appendix A).

#### 3.2.1. Composition of Mitosporic Fungi and Taxa Abundance Distributions

A great diversity of typical genera of mitosporic fungi was documented. A synopsis of their distribution is presented in Figure 7, while Figure 8 provides an overview of their morphological diversity. Only a small percentage of the genera documented corresponded to the typical sexual morphs of ascomycetes, basidiomycetes, or zygomycetes. As a result, communities of PLSF were predominantly composed of taxa that are typical asexual morphs of ascomycetes, i.e., hyphomycetes and coelomycetes. However, most genera in both assemblages were represented by a small number of isolates and account for the rare mycota (Table 3). The percentage AO of coelomycetes was greater than that for hyphomycetes and constituted over half of all fungal records (56% vs. 39%), suggesting that this assemblage type has a substantial role in determining the composition of the communities of PLSF (Figure 7). A similar result was observed with respect to taxon richness at the genus and MOTU levels. The genus richness of coelomycetes was twice that of hyphomycetes (47 vs. 23), with a total of 46 additional MOTUs in comparison with the latter (Table 2).

The most representative coelomycete genera were *Neosetophoma* (42% of all coelomycete records), *Phoma* (16%), *Epicoccum* (7%), *Neodidymelliopsis* (6%), *Didymella*, *Libertasomyces,* and *Sclerostagonospora* (5% each) (Figure 7). Thus, the main families of the assemblage of coelomycetes of PLSF were *Phaeosphaeriaceae*, *Didymellaceae,* and *Libertasomycetaceae* in the order *Pleosporales*. Regarding hyphomycetes, the most representative genera were *Alternaria* (49% of all hyphomycete records), *Cladosporium* (20%), *Fusarium* (10%), *Penicillium* (8%), and *Stemphylium* (7%) (Figure 7). Thus, the main families of the assemblage of hyphomycetes of PLSF were *Pleosporaceae*, *Cladosporiaceae*, *Nectriaceae,* and *Aspergillaceae* in the orders *Pleosporales*, *Cladosporiales*, *Hypocreales,* and *Eurotiales*, respectively.

Although genus richness was very expressive, most coelomycete genera were affiliated with families within the *Pleosporales*, suggesting a narrower taxonomic distribution at higher taxonomic ranks for these taxa. The taxonomic distribution of coelomycetes included 20 families in 7 classes and 3 orders. However, most corresponded to taxa represented by a minute number of records, including 15 singleton genera that express the high diversity of rare taxa. In contrast, the assemblage of hyphomycetes showed a wider taxonomic distribution at higher taxonomic ranks, including 19 families in 12 classes and 5 orders. Several of these comprised well-represented genera, with only five genera being singletons.

The aforementioned taxonomic distribution is reflected in the diversity measures, which indicate that the coelomycete assemblage is more diverse and even, especially in relation to genera (Table 4). Most diversity indices computed displayed higher values for the coelomycete assemblage, particularly those indices directed towards richness, such as the logseries alpha (9.83 vs. 4.43 and 50.25 vs. 36.46 for coelomycetes vs. hyphomycetes, respectively) and the Margalef (6.52 vs. 3.29 and 22.53 vs. 16.92) and Menhinick (1.38 vs. 0.82 and 4.69 vs. 4.04) richness indices (Table 4).

Differences between the values of the Simpson and Shannon diversity indices were less pronounced, although the coelomycete assemblage continued to exhibit, on the whole, a higher level of diversity (Table 4). These results, together with the values for the Shannon evenness index, express the high dominance of the very frequent mycota in both assemblages, especially the genera *Neosetophoma* and *Alternaria* and their MOTUs for coelomycetes and hyphomycetes, respectively (Figure 6, Table 4).

In order to gain a deeper insight into the structure of the community of PLSF, species abundance distributions (SADs) were constructed for the occurrence of both genera (Figure 9A) and MOTUs (Figure 9B). The assemblages of mitosporic fungi apparently influence the pattern of the PLSF community, which is well represented by either the genus or MOTU abundance distribution plots. Both plots are indicative of an assemblage with a high dominance pattern with few genera and MOTUs (very frequent and frequent mycota) with a percentage AO several orders of magnitude higher than the remaining taxa (infrequent and rare mycota) (Figure 9, Table 3). Both coelomycete and hyphomycete assemblages contribute similarly to the steeper slopes, which express a high dominance (commonness). However, the coelomycete assemblage contributes more to the shallower slopes, which express a profusion of rarity in the community of PLSF. Indeed, approximately 50% of the genera and 40% of the MOTUs composing the long tail of rare taxa in the abundance distribution plots are coelomycetes (Figure 7).

As theoretical SAD models are based on specific ecological assumptions that can help to understand the processes involved in the assembly of biological communities, five SAD models were fitted to the MOTU abundance distribution. A synopsis of the fit metrics for each model tested is presented in Table 5, and their expected SADs, along with the observed SADs for the MOTUs, are presented in Appendix A. The observed and expected SAD patterns were similar. Still, upon visual inspection, the logseries, neutral model, and lognormal appear to represent better the pattern observed in the MOTU abundance distribution. These three theoretical models seem to more accurately reflect the high unevenness characterising the assemblages of very frequent and rare taxa. This allows for observing the notable, steep decline in the ranked taxa abundance between the two assemblages, thereby illustrating the large proportion of rare taxa (Appendix A).

The predictive ability of the SAD models varied substantially as suggested by the strength-of-fit (SOF) measures (Table 5). However, the results of the goodness-of-fit (GOF) statistical tests showed that all SAD models exhibited a good fit and conformed with the observed SAD (*p* > 0.05), with the logseries and neutral model presenting the best performance (Table 5). The observed MOTU abundance distribution thus meets the ecological assumption in all competing SAD models. Still, the logseries and neutral model performed better compared with the other models tested following the *p*-values of the GOF tests. In addition, following the AIC and BIC values, the logseries was depicted as the most suitable theoretical SAD model to explain the community patterns for the assemblages of PLSF. Both AIC and BIC weights indicate that the logseries is more likely than the neutral model. The logseries presented AIC and BIC normalised probabilities of 75% and 95%, respectively, of being the best model to describe the MOTU abundance distribution, while the neutral model presented 25% and 5%, respectively (Table 5). The AIC and BIC values indicated that the lognormal, geometric series, and broken stick models had a very low normalised probability (<1%) of being suitable models to describe the observed SAD, confirming the visual inspection of the plotted modelled SAD for the geometric series and broken stick (Appendix A).

#### 3.2.2. Taxonomic Structure and Genetic Diversity

The fungal assemblage documented in association with the foliar lesions of palms presented a broad taxonomic distribution. Figure 10 provides an overview of the taxonomic structure of the communities of PLSF, which were found to comprise 54 families, 23 orders, and 9 classes in 3 phyla. However, nearly all fungal records obtained belonged to the phylum *Ascomycota* (99.13%), with the class *Dothideomycetes* being the most representative (82.70%), followed by *Sordariomycetes* (10.90%), *Eurotiomycetes* (0.04%), and a few other classes that were only occasionally documented.

With regard to orders, most fungal records are members of the *Pleosporales* (73.59%), including the hyphomycete and coelomycete genera with the highest percentage AO as well as most of the rare coelomycete genera. Most of these genera were affiliated with three highly represented families, namely *Phaeosphaeriaceae* (26.94%), *Pleosporaceae* (20.45%), and *Didymellaceae* (19.14%). However, a considerable number of less-represented families in *Pleosporales* were documented, such as *Libertasomycetaceae* and *Didymosphaeriaceae. Cladosporiales* was the second most represented order (7.41%) due to the occurrence of *Cladosporium* (*Cladosporiaceae*), followed by *Hypocreales* (4.32%) due to the occurrence of *Fusarium* (*Nectriaceae*), *Eurotiales* (3.88%) due to the occurrence of *Penicillium* and *Aspergillus* (*Aspergillaceae*), and many other less represented orders, such as *Sordariales* (1.89%), *Botryosphaeriales*, *Diaporthales* (1.50% each), *Xylariales* (1.45%), and *Amphisphaeriales* (1.02%) (Figure 10).

A total of 0.48% of the fungal records were identified as belonging to the phylum *Basidiomycota*. This was represented by two classes (*Agaricomycetes* and *Wallemiomycetes*), three orders (*Agaricales*, *Polyporales*, and *Wallemiales*), and seven families (*Physalacriaceae*, *Psathyrellaceae*, *Fomitopsidaceae*, *Meruliaceae*, *Polyporaceae*, *Phanerochaetaceae*, and *Wallemiaceae*). It is noteworthy that *Graphiola phoenicis* was also identified by its fruiting bodies in several specimens, but never in association with the discrete lesions selected for further investigation of the associated fungi. Accordingly, the *G. phoenicis* yeast state was never recovered from any foliar lesion upon isolation. The remaining 0.39% of the fungal records belonged to the phylum *Mucoromycota* and were all identified as belonging to the genus *Rhizopus* (*Rhizopodaceae*, *Mucorales*, *Mucoromycetes*) (Figure 10).

The genetic diversity of the PLSF was evaluated through a hierarchical cluster analysis (HCA) of the csM13 and (GTG)_5_ MSP-PCR profiles obtained for each fungal record. This approach yielded highly reproducible and complex genomic fingerprints comprising several bands ranging from 200 to 3500 base pairs. Based on these genomic fingerprints, fungal records were clustered into consensus dendrograms to determine the number of genomic types (GTs) within each MOTU. The genomic fingerprints obtained with the primer (GTG)_5_ were, in general, more complex and discriminatory than those obtained with the primer csM13. However, the combined analysis of the profiles of both primers presented a higher discriminatory power, enabling the resolution of some clusters that were not discriminated when the primers were used individually. The overall clustering pattern enabled the calculation of a substantially low isolation redundancy coefficient (IRC) of 6.54%, which accounted for 135 fungal records whose isolation was considered redundant. In addition, most of these fungal records were from the most commonly documented MOTUs of *Alternaria* and *Neosetophoma*. Given that the occurrence of these MOTUs is several times higher than that of the others, it can be assumed that the previous ecological observations pointed out were based on the isolation of effectively different fungal records and that the associated error is negligible.

To prevent any potential bias resulting from insufficient data, only those taxa with a percentage AO of at least 1.0% were included in the genetic diversity analysis. In general, genera with higher percentage AOs presented a higher number of MOTUs and GTs (Figure 11). However, it is noteworthy that some specific examples did not follow this trend, suggesting new insights for interpreting the patterns of the communities of PLSF. While the assemblage of coelomycetes was found to be more diverse than hyphomycetes, especially in terms of taxon richness and evenness (Table 4), by frequency group, the hyphomycete genera presented a relatively higher number of MOTUs, which displayed greater genetic diversity, i.e., a higher number of GTs. For instance, in the very frequent mycota, despite being the most frequently documented genus, *Neosetophoma* exhibited the lowest number of MOTUs and GTs, followed by *Phoma*, which was the third most documented genus; conversely, *Alternaria* and *Cladosporium* were found to be particularly diverse. Similarly, in the frequent mycota, *Fusarium*, *Penicillium,* and *Stemphylium* were found to exhibit greater genetic diversity than most of the coelomycete genera such as *Neodidymelliopsis*, *Didymella*, *Libertasomyces*, and *Sclerostagonospora*, which displayed a lower number of MOTUs with a greater degree of genomic homogeneity. A comparable contrasting result was observed in the infrequent mycota between the coelomycetous genus *Keissleriella* and the hyphomycetous genus *Aspergillus* (Figure 11).

### 3.3. Fungal Assemblages and Biodiversity Analyses

The bootstrapping permutational multivariate analysis of variance (PERMANOVA) test revealed significant differences (*p* < 0.05) in fungal communities among samples of the palm host species [F(5) = 2.17, *p* = 0.001], foliar lesion types [F(3) = 1.41, *p* = 0.016], and parishes [F(3) = 2.18, *p* = 0.001]. However, while the assumption of homogeneity of multivariate dispersion between palm host species and foliar lesion types was met [F(5) = 0.92, *p* = 0.500 and F(3) = 1.69, *p* = 0.174, respectively], it was not met between parishes [F(3) = 3.97, *p* = 0.013]. Accordingly, the impact of parishes on the composition of the communities of PLSF was disregarded for the subsequent analyses. Although foliar lesion types clearly affect the community composition of PLSF, they account for a relatively small amount of the variation (4.0%) when compared with the impact of palm host species (14.0%).

#### 3.3.1. Fungal Assemblages and Host Species

Nine distinct palm species were sampled and examined for associated fungi. More samples of *Chamaerops humilis* (32), *Phoenix canariensis* (31), and *Chrysalidocarpus lutescens* (19) were collected than of *T. fortunei* (14), *P. reclinata* (13), and *P. dactylifera* (12), and therefore, differences in the number of taxa collected from each host species are likely to be a reflection of this. *Washingtonia filifera* (8 samples), *Phoenix roebelenii* (3), and *Chamaedorea elegans* (2) were only occasionally surveyed and thus were not considered for ecological analyses, as the number of samples (less than 10) was too small for any supported trends to be observed (Table 1).

##### Quantitative and Qualitative Diversity

The overall diversity of the taxa documented in each palm host species was compared (Table 6). The fungal communities of all six species appear to be similarly diverse when assessed using the Simpson and Shannon diversity indices. This suggests that these indices may not be suitable for quantifying the biodiversity of foliar lesion fungal communities, particularly due to their high dominance patterns. For instance, the lower values of the evenness index for *Chamaerops humilis* and *Trachycarpus fortunei* are likely responsible for reducing the values of the Simpson and Shannon diversity indices. This highlights that *Phoenix canariensis* and *P. dactylifera* harbour the least diverse fungal communities as indicated by their high evenness index yet lower Shannon diversity indices (Table 6).

A comparison of the diversity of PLSF across different palm host species revealed notable differences when assessed through species richness indices. In general terms, *C. humilis* was shown to harbour the most diverse fungal communities (α = 72.56, D_Mg_ = 26.08, and D_Mn_ = 6.48), followed by *Chrysalidocarpus lutescens* (α = 59.32, D_Mg_ = 17.10, and D_Mn_ = 6.20), *P. reclinata* (α = 55.90, D_Mg_ = 18.45, and D_Mn_ = 5.95), and *T. fortunei* (α = 40.22, D_Mg_ = 14.89, and D_Mn_ = 4.95). The fungal communities associated with foliar lesions of *P. canariensis* and *P. dactylifera* were observed to be the least diverse (α = 35.83 and 29.87, D_Mg_ = 11.45 and 9.07, and D_Mn_ = 4.82 and 4.38, respectively), as also suggested by the Simpson and Shannon diversity indices (Table 6). The results obtained for *P. canariensis* are of particular interest, as it was the host species with the second-highest number of samples collected, with a similar number of samples to *C. humilis* (Table 1). Therefore, this lends further support to the hypothesis that the observed results are indeed host-related and not merely a reflection of sample bias.

Differences in the diversity of PLSF between different palm host species were also expressed taking into account the fungal richness per sample. The isolation rate (IR), which expresses the degree of multiple colonisations of the samples of a given host, was particularly high in samples from *P. reclinata* (25.38), followed by *C. humilis* (22.00), *T. fortunei* (21.50), and *C. lutescens* (12.11). The ratio between the number of MOTUs and the number of samples (number of MOTUs per sample) was also evaluated. This ratio can be used as a measure of diversity, as it expresses the number of MOTUs documented to the number of samples collected for a given host species. Therefore, this ratio is an indirect measure of the MOTU richness for a given host species and is normalised with the number of samples examined, which can diminish sampling effort biases. Similarly, at the MOTU level, the same four palm species were depicted as harbouring the most diverse fungal communities, with *P. reclinata* appearing to be particularly diverse (Table 6). Although *P. reclinata* samples appear to be particularly rich in both fungal records and number of MOTUs, the small sample size suggests that this palm species may be under-sampled (Table 1). Consequently, the fungal richness per sample may be overestimated to a certain extent due to the inclusion of multiple colonised samples. Naturally, this effect is diluted in diversity measures that account for a greater number of parameters.

Notwithstanding the apparent discrepancies in richness and diversity measures between the fungal communities documented among different palm host species, the statistical significance of these differences was evaluated by considering samples from the same host as potential replicates. The Kruskal–Wallis test revealed significant differences (*p* < 0.05) in the MOTU richness [*H*(5) = 25.26, *p* = 0.001], Shannon diversity index [*H*(5) = 21.11, *p* = 0.008], Shannon evenness index [*H*(5) = 15.02, *p* = 0.01] and logseries alpha [*H*(5) = 15.78, *p* = 0.007] among samples of the six palm host species sampled. Post hoc pairwise multiple comparisons using the Dunn test with Bonferroni adjustments for the four measures were significant (*p* < 0.05) for only the following combinations *C. humilis* versus *P. canariensis*, *P. canariensis* versus *P. reclinata*, *P. dactylifera* versus *P. reclinata,* and *P. canariensis* versus *T. fortunei*. No other differences were statistically significant (*p* > 0.05). The exact *p*-values obtained for significant comparisons of MOTU richness, Shannon diversity index, Shannon evenness index, and logseries alpha were as follows: *p* = 0.02, 0.04, 0.003, 0.008, respectively, for *C. humilis* versus *P. canariensis*; *p* = 0.0003, 0.0007, 0.049, 0.006, respectively, for *P. canariensis* versus *P. reclinata*; *p* = 0.035, 0.049, 0.049, 0.007, 0.009, respectively, for *P. dactylifera* versus *P. reclinata*; *p* = 0.022, 0.016, 0.004, 0.007, respectively, for *P. canariensis* versus *T. fortunei*.

Qualitative examination of patterns of diversity and abundance may be conducted by comparing the taxonomic distributions of fungal taxa between the palm host species sampled. The biodiversity pattern observed in each host species was similar to that observed in the overall fungal community (Figure 9). Thus, a high profusion of infrequent and rare taxa was recorded together with a few very frequent genera. The top ten genera and MOTUs by assemblage type for each palm host species are presented in Figure 12 and Appendix A, respectively. Several trends are apparent, and differences can be seen in the mycota occurring on different palm host species.

(1) The fungal assemblages of *C. humilis*, *P. reclinata,* and *T. fortunei* (group A) exhibited an enrichment in coelomycetes, with *Neosetophoma* being a particularly abundant genus in their foliar lesions. Although *Alternaria* was the second most abundant genus in group A, its AO was less pronounced when compared with the other palm host species sampled (Figure 12). With regard to MOTUs, *Neosetophoma* MOTU 188 was the most abundant in *C. humilis* and *T. fortunei*, while *Neosetophoma* MOTU 185 was the most abundant in *P. reclinata*, followed by the former with an identical AO (Appendix A). In general, the top coelomycete assemblage appears to exhibit greater diversity in terms of genus and MOTU richness in group A than the top hyphomycete assemblage. *Chamaerops humilis* displayed a greater number of shared genera and MOTUs with *T. fortunei* than with *P. reclinata* (Figure 12 and Appendix A).

(2) The fungal assemblages of *C. lutescens*, *P. canariensis,* and *P. dactylifera* (group B) showed either an enrichment in hyphomycetes or assemblages of hyphomycetes and coelomycetes with very similar relative proportions. *Alternaria* was the most abundant genus, followed by *Cladosporium* or *Phoma*, in group B, while *Neosetophoma* had a much lower AO or was only occasionally documented in comparison with group A. For instance, *Neosetophoma* was not among the top ten genera documented in *P. dactylifera* (Figure 12). With regard to MOTUs, *Alternaria* MOTU 009 was the most abundant in *P. canariensis* and *P. dactylifera*, and together with *Alternaria* MOTU 015, it was also the most abundant in *C. lutescens* (Appendix A). It is noteworthy that some non-mitosporic fungi are among the top ten genera and MOTUs documented in group B, including *Chaetomium* in *C. lutescens*, *Pyronema* in *P. canariensis*, and *Fasciatispora* and *Nothodactylaria* in *P. dactylifera*. Although *Chaetomium* was also identified as one of the most abundant genera in *T. fortunei*, its AO was residual compared with that observed in *C. lutescens*. This pattern is particularly interesting in *P. dactylifera*, as *Fasciatispora* MOTU 122 and *Nothodactylaria* MOTU 200 were among the top five most documented MOTUs (Figure 12 and Appendix A).

(3) In general terms, the AOs of the most documented MOTUs appear to differ between the two aforementioned groups of palm host species. While *Neosetophoma* MOTU 188 was more common in group A, either *Neosetophoma* MOTU 185 or *Neosetophoma* MOTU 188 was more common in group B, and their occurrence was highly host-related. *Alternaria* MOTU 009 was more prevalent in group B, while *Alternaria* MOTU 015 tended to be more prevalent or as prevalent as *Alternaria* MOTU 009 in group A. Of note, *P reclinata* in group A and *C. lutescens* in group B exhibited relative abundance patterns of taxa that were intermediate to the two groups. In some instances, *P. reclinata* resembled the relative abundance patterns of taxa from group B (Appendix A). The analysis of qualitative and quantitative fungal diversity documented in *Washingtonia filifera* was disregarded due to potential sample bias. However, the observed patterns were more closely aligned with those of group B, with similar relative proportions of coelomycetes and hyphomycetes and *Alternaria* and *Phoma* as the most common genera (Appendix A).

The above trends are consistent with the PERMANOVA results, which revealed that the palm host species had a statistically significant influence of 14.0% on the composition of the fungal communities documented. A post hoc pairwise multiple-comparisons PERMANOVA test with Bonferroni adjustments further revealed that this influence was significant (*p* < 0.05) between samples of *C. humilis* versus *C. lutescens* (*p* = 0.015), *C. humilis* versus *P. canariensis* (*p* = 0.045), *C. humilis* versus *P. dactylifera* (*p* = 0.03), *C. lutescens* versus *P. canariensis* (*p* = 0.015), *C. lutescens* versus *P. reclinata* (*p* = 0.015), *P. canariensis* versus *P. reclinata* (*p* = 0.015), and *P. dactylifera* versus *P. reclinata* (*p* = 0.015). No other differences were statistically significant (*p* > 0.05).

Despite the existence of patterns of similarity between the fungal communities documented in the six palm host species, each palm host species exhibited specific taxa whose occurrence was exclusive (most rare taxa, including singletons and doubletons) or taxa whose occurrence was preferential. Figure 13 presents the general biodiversity patterns of taxon richness and overlap by host species.

In general, *C. humilis* was the host species in which a greater number of MOTUs per genus was documented, consistent with the quantitative diversity analyses that showed *C. humilis* to harbour the most diverse fungal communities (Appendix A). In relative terms, *Fusarium* and *Diaporthe* were the most speciose genera found in *C. humilis*, with other common genera in terms of the number of species including *Didymella*, *Libertasomyces,* and *Stemphylium* as well as many other coelomycete genera such as *Coniothyrium*, *Epicoccum*, *Neodidymelliopsis,* and *Neosetophoma* (Figure 13A and Appendix A).

*Phoenix reclinata* was the host species with the second most MOTUs per genus, followed by *T. fortunei* (Appendix A). In relative terms, the most speciose genera in *P. reclinata* were *Hendersonia* and *Phaeosphaeria*, with other common coelomycete genera in terms of the number of species including *Coniothyrium*, *Keissleriella,* and *Sclerostagonospora*, whereas *Aspergillus* was relatively the most speciose genera in *T. fortunei*, followed by many coelomycete genera that were similarly common in terms of the number of species, such as *Neosetophoma*, *Neodidymelliopsis,* and *Phoma* (Figure 13A and Appendix A).

Most genera were not particularly speciose in *C. lutescens*, and especially in *P. canariensis* and *P. dactylifera*. The most speciose genus in *C. lutescens* was *Penicillium*, followed by *Alternaria*, *Cladosporium,* and *Diaporthe*, all three of which were less speciose than in *C. humilis* (Figure 13A and Appendix A).

The degree of overlap between genera and MOTUs among combinations of the six palm host species sampled was found to be very low, with values ranging from 3% to 30%. The overlap of genera between the six palm host species was less than 9% (8.60%), while between five, four, three, and two it was less than 13% (8.60–12.80%) and from just over to less than 16% (9.68–16.13%), 20% (17.75–20.43%), and 30% (13.98–30.11%), respectively (Figure 13B,D). With regard to MOTUs, the overlap between the six palm host species was less than 3% (2.72%), while between five, four, three, and two it was from just over to less than 4% (2.72–4.42%), 5% (2.72–6.46%), 11% (3.74–11.22%), and 23% (6.12– 23.13%), respectively (Figure 13C,D).

The genera documented in all six palm host species included the very frequent mycota (*Alternaria*, *Cladosporium*, *Neosetophoma,* and *Phoma*) and four genera of the frequent mycota (*Epicoccum*, *Libertasomyces*, *Penicillium,* and *Stemphylium*). Conversely, while three of the MOTUs documented in all six palm host species corresponded to the most frequently documented mycota, including *Alternaria* MOTU 009 and *Cladosporium* MOTU 058 and MOTU 059, the remaining five corresponded to rare taxa, including *Cladosporium* MOTU 056 and MOTU 060, *Epicoccum* MOTU 114, *Libertasomyces* MOTU 158, and *Phoma* MOTU 245.

In general, the fungal taxa overlapping values were found to be greater in combinations of palm host species involving *C. humilis*, *T. fortunei,* and/or *P. reclinata*, which corresponded to the previously identified group A for the qualitative biodiversity trends. However, in some instances, *C. lutescens* was also depicted in such combinations. For example, the top three overlap of genera and MOTUs between two palm host species were for the following combinations: *C. humilis*–*P. reclinata*, *T. fortunei*–*C. lutescens*, and *C. humilis*–*T. fortunei* or *C. humilis*–*C. lutescens*, respectively (Figure 13D). *Chamaerops humilis* was the host species with the highest number of unique taxa, comprising 14% of all genera (Figure 13B) and 24% of all MOTUs (Figure 13C), which aligns with the previous quantitative analyses that depicted *C. humilis* as the host species with the most diverse fungal communities (Table 6). This figure is approximately double and triple, respectively, the numbers of unique taxa observed in the host species with the second-highest numbers, namely, *C. lutescens*, *P. reclinata,* and *T. fortunei* (8% and 8–9%, respectively). *Phoenix canariensis* and *P. dactylifera* were the host species with the lowest numbers of unique taxa (Figure 13B,C), which also aligns with the previous quantitative analyses that depicted them as the host species with the least diverse fungal communities (Table 6). Most unique taxa correspond to rare mycota, including singletons and doubletons, and therefore, *C. humilis* was the host species with the highest number of rare taxa.

The differences observed between the fungal assemblages documented in the different palm host species are not readily apparent when analysed at the sample level. However, they become more evident when the data from the sample set of a given host are analysed as a whole (Figure 14).

A principal component analysis (PCA) was conducted to visualise differences in the communities of PLSF among the samples collected from the different host species. Around 32% of the total variation in the data was represented by the first two PCA axes, indicating that the data did not fit the PCA model very well. Principal component (PC) 1 showed a total of 23.27% of the variance, mainly related to *Neosetophoma* MOTU 188. Conversely, PC 2 accounted for 10.30% of the variability, mainly related to *Alternaria* MOTU 015 and *Neosetophoma* MOTU 185, which exhibited a negative correlation. *Alternaria* MOTU 015 and *Neosetophoma* MOTU 185 also explained some of the variance in PC 1, and the latter were more correlated with *Neosetophoma* MOTU 188 than the former (Figure 14A).

The PCA results corroborate the previous analyses that showed that the AO of the most common *Alternaria* and *Neosetophoma* MOTUs contribute to the differences observed in the fungal communities of different palm host species. The remaining MOTUs had small scores on both PCs and showed different patterns of correlation; only some of them are shown in Figure 14A. For instance, *Alternaria* MOTU 009, *Epicoccum* MOTU 111, and *Neodidymelliopsis* MOTU 168 are more correlated with *Alternaria* MOTU 015; *Phoma* MOTU 243 and other *Neosetophoma* MOTUs are more correlated with *Neosetophoma* MOTU 185 and MOTU 188.

Although the PCA suggests some grouping of samples by palm host species, particularly for *C. lutescens*, *P. reclinata*, *P. canariensis*, *P. dactylifera,* and *T. fortunei*, which had smaller and/or narrower 95% confidence ellipses, there was a considerable overlap between samples from different host species. This is particularly true for *C. humilis* and *P. reclinata* (Figure 14A). However, a small amount of variability is accounted for by the first 2 PCs, and a total of 24 PCs would be required to account for at least 80% of the total variance. Despite differences in the abundance of the more common taxa, communities of PLSF among palm host species are mostly shaped by the presence of rare taxa, whose considerable variability could not be reliably reduced to the limited number of PCs used. In addition, it was diluted by the similarities in the most common taxa, which showed a broad distribution among samples of different palm host species.

To investigate further the influence of palm host species on the diversity and composition of their associated PLSF communities, an HCA was carried out based on the AO of the MOTUs. The dendrogram presented a cophenetic correlation coefficient (c) value greater than 0.90, revealing that the clustering obtained was reliable and very well fit. Two main clusters were evident, comprising *T. fortunei* and *C. humilis* (C1) and the remaining palm host species (C2). These clusters were highly congruent with the groups delineated according to fungal biodiversity trends, except for *P. reclinata*, whose top fungal assemblage seemed more similar to *T. fortunei* and *C. humilis* (GA) but clustered with host species of GB in the HCA (Figure 14B).

##### Assessment of the Sampling Process

To ascertain whether an adequate number of samples had been collected, the relationship between increasing sample size and taxa recovery was examined. Species accumulation curves (SACs) for genera and MOTUs were constructed for both the overall fungal community and for each palm host species.

The accumulation curves for the overall fungal community at the genus and MOTU levels showed a similar pattern and did not reach an asymptote, indicating that the number of samples collected was insufficient to recover all potential taxa. The observed genus and MOTU richness displayed a steeply rising rarefied SAC. This pattern is consistent with communities comprising a high proportion of rare species and a few abundant species, as evidenced by the relatively low inflection point on the ordinate axis and the extended upward slope towards the potential asymptote. Furthermore, although an asymptote could not be seen, both curves exhibited an asymptotic trend, suggesting that the sampling effort had been substantial (Figure 15). However, the genus accumulation curve exhibited a more pronounced inflection point with a more noticeable change in curvature slopes (Figure 15A), suggesting that the communities of PLSF are more thoroughly sampled at the genus level than at the MOTU level. In contrast, the MOTU accumulation curve was less flattened towards the end with a steeper slope, characteristic of a community with great diversity (Figure 15B).

Extrapolated taxon richness suggested a similar range of unsampled taxa for both genera and MOTUs. None of the richness estimators reached a plateau, and as expected, the different functions yielded substantially different estimates (Table 7).

The asymptotic approach was the method that indicated the lowest number of potentially unsampled taxa, with asymptotic values very close to the observed taxon richness (103 for genera and 324 for MOTUs) (Table 7). Although all five SAC models tested showed a good fit (*p* > 0.05), the AIC and BIC values indicated that the Weibull distribution presented the best performance. However, the asymptotic approaches were not considered in the present analysis, as the accumulation curves indicated that the genera and MOTUs recovered from the foliar lesions of palms were far below their potential diversity (Figure 15). It is likely that the expected distribution for each model was highly influenced by the observed data. In addition, this approach assumes that all species detection probabilities are equal, resulting in an underestimation of true species richness where heterogeneity exists. The coefficient of variation (CV) for genera and MOTUs was very high (CV = 2.96 and 3.53, respectively), indicating a high degree of heterogeneity among taxa abundances, as already illustrated by the SACs for both assemblages. Consequently, the estimates obtained by the asymptotic approaches yielded substantially low estimates for each assemblage.

The nonparametric models indicated an extrapolated taxon richness ranging from 111 to 158 genera and 381 to 505 MOTUs (Table 7) with mean values of 141 and 483, respectively. Of these, only 97 genera and 320 MOTUs were recorded, suggesting that about 61.39% to 87.39% (mean = 68.79%) and 59.26% to 83.99% (mean = 67.38%) of the potential diversity of genera and MOTUs, respectively, from the foliar lesions of palms was recorded. Extrapolated taxon richness was generally higher for the Chao 1 and jackknife estimators and lower for the bootstrap estimator.

The SACs produced for each palm host species at the genus and MOTU levels showed a pattern comparable to those observed for the overall community. Notably, none of the curves reached a plateau, indicating that there are yet more fungal representatives of the foliar lesion communities that remain to be recovered (Figure 16).

When comparing the diversity of the fungal communities among the six palm host species based on the initial slopes of the SACs (Figure 16), two groups were evident, which were consistent with the previous quantitative analysis of fungal diversity (Table 6). In addition, the extrapolated taxon richness also indicated diversity trends comparable to those observed with the genus and MOTU accumulation curves (Table 8). The fungal communities of *C. humilis*, *P. reclinata,* and *T. fortunei* appeared to be the most diverse, as evidenced by the steeper slope gradient in their SACs when compared with those of *P. canariensis* and *P. dactylifera*. Therefore, the former are expected to harbour a potential greater number of unrecorded taxa. Indeed, *C. humilis* was identified as the host species harbouring the most potentially diverse communities of PLSF, with 52 genera and 172 MOTUs accounting for approximately 64–87% and 52–81%, respectively, of its potential fungal diversity, with an expected average of 71 genera and 276 MOTUs. The second most diverse host species was *T. fortunei*, with 44 genera and 88 MOTUs accounting for approximately 48–83% and 37–79%, respectively, of its potential fungal diversity, with an expected average of 74 genera and 172 MOTUs (Table 8). *Phoenix reclinata* and *C. lutescens* also presented relatively high expected fungal diversities, as evidenced by their SACs (Figure 16), with expected averages of 50 and 65 genera and 181 and 201 MOTUs, of which 58–80% and 46–60%, respectively, were recovered (with the lower percentages corresponding to *C. lutescens*) (Table 8). *Chrysalidocarpus lutescens* exhibited intermediate slopes, which were clearly less steep than those observed for *C. humilis*, *P. reclinata,* and *T. fortunei* (Figure 16). However, the nonparametric models indicated a considerably high extrapolated taxon richness for *C. lutescens*, suggesting that its fungal diversity is far from well documented (Table 8). *Phoenix canariensis* and *P. dactylifera* were identified as the host species harbouring the least potentially diverse communities of PLSF, with predicted mean extrapolated taxon richness of 50 genera and 104 MOTUs and 36 and 79, respectively, of which approximately 64% of genera and 56% of MOTUs were recovered for *P. canariensis* and 69% and 53% were recovered for *P. dactylifera* (Table 8).

The confidence intervals (CIs) for the genus accumulation curves were relatively wider (Figure 16A) than the CIs for the MOTU accumulation curves (Figure 16B), indicating that genus composition across the samples is less consistent than the MOTU composition. Trends in potential fungal diversity on foliar lesions of each palm host species are also illustrated by the CIs of the SACs. *Chamaerops humilis* displayed notably wider CIs, implying a greater variability in genus and MOTU composition between samples (Figure 16). A comparable result was observed for the genus composition of samples of *T. fortunei* and *C. lutescens* (Figure 16A). In contrast, the CIs for *P. canariensis* and *P. dactylifera* were relatively narrower, suggesting a lower variability in their genus and MOTU composition between samples (Figure 16).

#### 3.3.2. Fungal Assemblages and Foliar Lesion Types

Four distinct foliar lesion types were defined based on their general morphological characteristics. More TDB samples (65) were collected than LLS (22), SLS (24), and PP (23), and therefore, differences in the numbers of taxa collected from each foliar lesion type are likely to be a reflection of this (Table 1). The PERMANOVA results revealed that foliar lesion types had a statistically significant influence on the composition of the fungal communities documented. However, it accounted for a relatively small amount of the variation (4.0%). In addition, a post hoc pairwise multiple comparisons PERMANOVA test with Bonferroni adjustments further revealed that this influence was significant (*p* < 0.05) only between TDB versus SLS samples. The exact *p*-values obtained for the post hoc pairwise multiple comparisons PERMANOVA tests of foliar lesion types were as follows: *p* = 0.324 for TDB versus PP; *p* = 1.000 for TDB versus LLS; *p* = 0.018 for TDB versus SLS; *p* = 1.000 for PP versus LLS; *p* = 1.000 for PP versus SLS; *p* = 0.636 for LLS versus SLS. Indeed, no major trends in the fungal communities were observed among the foliar lesion types.

##### Number of Taxa in a Single Foliar Lesion

There were no major differences or trends in the biodiversity of the fungal communities documented in each foliar lesion type, except for the relative frequency of coelomycetes and hyphomycetes. While coelomycetes were more prevalent than hyphomycetes in TDB and LLS, the opposite was observed in SLS and PP, where hyphomycetes were the most abundant assemblage type, particularly in the PP samples (Figure 17).

All foliar lesion types presented exclusive genera, consisting mainly of representatives of the rare mycota. However, these were more prevalent in the TDB samples, which had the highest median MOTU count, with the boxplot indicating a greater diversity of MOTUs compared with the other foliar lesion types. Indeed, the interquartile range (IQR) for the TDB samples suggested a wide distribution of MOTU counts, and the large whiskers extending to almost 35 MOTUs indicated high variability. Similarly, the mean number of MOTUs was also higher in the TDB samples, suggesting that on average, this foliar lesion type harbours a higher number of MOTUs (Figure 17). Assessing the distribution of fungal occurrences on each sample, it can be observed that the TDB samples have a wide range of MOTUs per sample, with high diversity even at the upper end (MOTUs from 10 to over 30) (Table 9). This is consistent with the boxplot analyses, which showed no outliers for the distribution of MOTUs for this foliar lesion type, despite the presence of samples with exceptionally high MOTU counts (Figure 17). For instance, 17 out of 19 samples in the highest categories of number of MOTUs per sample (20 < n ≤ 40) correspond to TDB samples. In addition, TDB samples are represented across all categories of number of MOTUs per sample (from n = 1 to n ≤ 20) (Table 9), highlighting that TDB samples support a wide range of fungal colonisation, comprising from single MOTU to highly diverse MOTU assemblages.

The LLS, SLS, and PP samples showed low median and mean values for the number of MOTUs, with narrower and similar IQRs, suggesting that these foliar lesion types are more consistent in MOTU counts across samples with less variability (lower fungal diversity). Outliers were observed only in the LLS samples, which presented two markedly atypical samples (Figure 17). These should be regarded as extraordinary outliers given their considerable divergence from the norm, with a magnitude exceeding three times the IQR. Therefore, similarly to the TDB samples, the LLS samples also showed a wide range of MOTUs per sample. However, the number of MOTUs per sample in the LLS samples was typically situated within the intermediate range, with hyperdiverse samples representing exceptional outlier cases (Figure 17, Table 9). The SLS and PP samples tended to support fewer MOTUs per sample, with few samples having a number of MOTUs per sample equal to or greater than 10 (Table 9). These results suggest that SLSs and PPs might support more specialised or limited fungal communities consistent with the homogeneity depicted by the IQRs observed for these foliar lesion types (Figure 17).

The overall boxplot showed intermediate median and mean values. However, the spread of the box was more moderate compared with the TDB samples and more similar to that observed for the LLS, SLS, and PP samples. In addition, the several outliers observed for the overall sample set corresponded to TDB samples, further suggesting that TDB samples can support an atypical, exceptionally high number of MOTUs (Figure 17). Therefore, although a potentially richer fungal diversity can be observed in foliar lesions of palms, they will mostly support a fungal diversity of 1 to 10 MOTUs (Table 9). The mean number of MOTUs per sample was eight. However, given the presence of extreme and discrepant values (outliers) on an asymmetric distribution of MOTU richness (Figure 17), the median is a more suitable measure of central tendency in the present analysis, as it is less strongly affected by outliers. The median number of MOTUs per sample was 5, which is the number of MOTUs that are most likely to occur in a single foliar lesion of a palm, although the maximum number of MOTUs observed in a single foliar lesion in this study was 38.

##### Taxa Co-Occurrence

Pairwise taxon co-occurrence (CO) analyses were conducted to further examine the structuring of the communities of PLSF. The results are presented as CO matrices at the genus and MOTU levels in Figure 18. Additionally, Appendix A summarise the percentage of total pairwise interactions for each taxon that were classified as positive, negative, or random, and Appendix A summarise the significant taxon pairwise interactions and their respective probabilities. At the genus level, a total of 4656 taxa pair combinations were analysed, with 4204 pairs (90.29%) removed due to an expected CO value of less than 1 sample. Of the remaining 452 pairs, 340 (75.22%) showed random associations, while 112 (24.78%) exhibited positive associations (Figure 18A, Appendix A). At the MOTU level, a total of 51,040 taxa pair combinations were analysed, with 50,377 (98.70%) removed for having an expected CO of less than 1 sample. The remaining 663 pairs revealed 563 (84.92%) random associations and 100 (15.08%) non-random associations. Of these, 98 (14.78%) were positive and 2 were negative (0.30%) (Figure 18B, Appendix A).

At either the genus or MOTU level, the communities of PLSF are mostly composed of random associations between taxa, suggesting that there is no significant relationship in their occurrence together or apart. Therefore, most taxa seem to colonise the foliar lesions opportunistically (Figure 18). However, a quarter of the associations at the genus level were positive, meaning that several genera were found together significantly more often than expected by chance. Notably, no negative associations were observed at the genus level, suggesting that most taxa do not compete strongly in this environment (Figure 18A, Appendix A). Similarly, although the majority of the MOTUs were not preferentially co-occurring, about one-seventh of the associations at the MOTU level were positive, reflecting significant CO patterns that might point to ecological or biological relationships (Figure 18B, Appendix A).

Most positive associations seem to involve the most frequently documented mycota, namely, those taxa that were regarded as frequent and very frequent. Moreover, these are mostly associated with representatives of the coelomycete assemblage (Figure 6, Table 3 and Appendix A). To illustrate, the top five genera involved in positive CO are *Neosetophoma* (18 of all positive CO pairs at the genus level), the most prevalent genus, and *Fusarium* (16), *Didymella* (15), *Neodidymelliopsis* (12) and *Alternaria* (11). Other genera that often co-occur significantly with other genera are *Epicoccum*, *Libertasomyces,* and *Phoma* (Appendix A). Similarly, the top five MOTUs involved in positive CO were *Neosetophoma* MOTU 188 (37 of all positive CO pairs at the MOTU level) and MOTU 185 (11), *Alternaria* MOTU 015 (21), and *Phoma* MOTU 235 (12) and MOTU 243 (5). The first three are among the most prevalent MOTUs (Appendix A). It is noteworthy that, despite *Alternaria* being the second most frequently documented genus, its CO with *Neosetophoma* appears to be mostly random (Figure 18A, Table 3 and Appendix A). Indeed, although a significantly positive CO was detected between *Neosetophoma* MOTU 188 and *Alternaria* MOTU 015, most of the remaining pairs of MOTUs belonging to both genera appear to co-occur randomly (Figure 18B, Appendix A). Similarly to *Alternaria*, although *Cladosporium* was the fourth most prevalent genus, its CO patterns were mainly characterised by random associations (Figure 18A, Table 3 and Appendix A).

Infrequent and rare taxa are predominantly implicated in random CO patterns given their low AO, which makes it less likely that they will engage in significant associations with other taxa. Nevertheless, the observation of significantly positive associations between some of the infrequent and rare mycota with the frequent and very frequent mycota may suggest that the latter exert an influence in the establishment of the former in foliar lesion environments or vice versa (Figure 18). For instance, *Hendersonia* and *Phaeosphaeria*, although regarded as rare genera, presented a higher proportion of significantly positive pairings than random associations. A similar pattern was observed for *Didymella*, *Fusarium,* and *Neodidymelliopsis*. The remaining genera involved in significantly positive associations showed a higher proportion of random interactions (Figure 19, Appendix A). At the MOTU level, only *Phoma* MOTU 246, also regarded as rare, presented a higher proportion of significantly positive pairings than random associations. However, a high proportion of significantly positive versus random associations was also observed for *Alternaria* MOTU 015 and MOTU 019, *Hendersonia* MOTU 144, *Stagonosporopsis* MOTU 298, and *Neosetophoma* MOTU 188 (Appendix A). The pairwise CO analyses at the MOTU level provide a finer resolution of the functional and ecological ties between taxa. As a result, two significantly negative associations were observed, namely, between *Botrytis* MOTU 039 and *Alternaria* MOTU 009 and *Neosetophoma* MOTU 188 and *Stemphylium* MOTU 303, hinting at potential competition or exclusion between these MOTUs (Figure 18B, Appendix A).

## 4. Discussion

The aetiology of plant diseases has traditionally been understood through the concept of the “disease triangle”, which involves a susceptible host, a virulent pathogen, and an abiotic environment conducive to infection [173]. However, an essential element—the plant-associated microbiome—has been largely overlooked. Growing evidence now underscores the significant influence of plant-associated microbes on plant health and disease outcomes, e.g., refs. [174,175,176,177]. In the context of an increasingly globalised world and with a growing awareness of the environmental challenges posed by climate change, it is imperative to adopt a more integrated and comprehensive approach to study the factors that impact plant health. The global transportation of palms for use in interiorscapes and landscapes, especially the movement of tropical species to non-native environments, represents a key pathway for the introduction of potential fungal pathogens into new regions [26,27]. Therefore, while “instant” landscapes may offer immediate aesthetic benefits, they may also cause long-term detrimental effects on crop production, ecosystems, and the environment. Furthermore, minor damage-causing diseases such as leaf spots pose great challenges to the ornamental palm industry due to the potential loss of their high aesthetic value [178].

Given that ornamental palms are widely used as decorative plants in cityscapes, garden compositions, and indoor environments in Portugal, the present study aimed to conduct a preliminary assessment of the diversity and ecology of the potential phytopathogenic fungi associated with ornamental palm trees in Lisbon. For this purpose, foliar lesions were used as a case study to explore the composition and dynamics of communities of palm leaf spotting fungi (PLSF). Palm foliar lesions were recognised as hyperdiverse microhabitats, harbouring a wide variety of fungi exhibiting complex ecological dynamics and biodiversity patterns influenced by the palm host species. To the best of the authors’ knowledge, this is the first study to conduct a comprehensive and integrated analysis of the fungal assemblage associated with palm foliar lesions.

### 4.1. Biodiversity and Ecological Dynamics of Palm Leaf Spotting Fungal Communities

A total of 2064 isolates were obtained from 134 foliar lesions sampled from 100 palm trees. These were classified into 97 genera and 320 molecular operational taxonomic units (MOTUs). The overall composition of the fungal communities associated with foliar lesions of palms revealed a well-defined pattern of dominance by a few genera—*Neosetophoma*, *Alternaria*, *Phoma,* and *Cladosporium*—alongside a remarkable profusion of infrequent and rare genera. This trend was similarly reflected at the MOTU level, with four highly prevalent MOTUs, namely, *Neosetophoma* MOTU 188 and MOTU 185 and *Alternaria* MOTU 009 and MOTU 015. In addition to these 4 very common taxa, the PLSF community was composed of approximately 10 moderately common taxa and a considerable number of low-frequency taxa. This distribution pattern, characterised by a few highly abundant taxa and a wide array of taxa with only limited occurrences, mirrors well-documented trends in plant, animal, and fungal biodiversity studies, e.g., refs. [129,179,180,181,182,183], underscoring the complexity of palm foliar lesions as hyperdiverse microhabitats with distinct ecological dynamics. Similar findings have also been reported in investigations of both palmicolous endophytes and saprobes, e.g., refs. [184,185,186,187], further corroborating that palm foliar lesions harbour rich and structured mycobiota. For instance, Fröhlich et al. [184] observed that the endophyte communities of two *Licuala* palms were composed of a single dominant xylariaceous anamorph, approximately 10 less common but equally ubiquitous species, and a large number of species occurring at very low frequencies.

Although the precise ecological processes underlying the establishment of PLSF communities are unknown, it is clear that only a small number of taxa play a critical role in maintaining the overall community structure. An analysis combining genus- and MOTU-level compositions identified just 8 of the 97 genera as core components of the PLSF communities. In addition to the four dominant genera, *Neodidymelliopsis*, *Epicoccum*, *Stemphylium,* and *Fusarium* also contributed to the prevalent MOTUs found in palm foliar lesions. Most of these eight core genera are well-documented pathogens and saprophytes, suggesting that biotic interactions and saprophytic activity are key drivers in structuring the fungal communities associated with palm foliar lesions.

#### 4.1.1. Ecological Significance of Dominant Taxa and Functional Implications in Disease Development

Several ecological studies have suggested that a small number of common species play a major role in shaping patterns of species richness [188] and community composition [189,190,191]. However, the dynamics of commonness and rarity among taxa within biological communities continue to intrigue researchers, with various ecological processes frequently identified as potential drivers of these patterns [183,192,193,194,195,196]. In the present study, the ecological dynamics underlying the structuring of PLSF communities were tentatively explored by constructing species abundance distributions (SADs) for both genera and MOTUs. Both SADs clearly supported the fungal community structure observed, evidencing highly skewed, J-shaped distributions. In addition, commonly studied SAD models were fitted to the MOTU abundance distribution to gain a deeper understanding of the ecological dynamics of the PLSF, as such models differ in how the taxa are assumed to partition limited resources within a community [126,197]. The MOTU abundance distribution was best described by the logseries model [124]. The logseries model is characterised by one or a few factors governing the distribution of taxa coupled with random intervals between taxa arrivals into a habitat, which results in a pattern where only a few taxa become dominant [117,126].

The steep slopes observed in the SAD plots in this study are indicative of the presence of a highly dominant assemblage represented by the most frequently documented genera and their corresponding MOTUs. This suggests that PLSF communities are shaped by a limited number of factors related to the establishment of symptoms, likely involving the interaction between the potential fungal phytopathogens or early colonisers and host leaf tissue. Once these taxa establish the lesions, subsequent mycota arrive randomly, colonising the already damaged tissue where a limited portion of nutrients remains available. Indeed, the logseries model predicts that the dominant species pre-empt the largest portion of limiting resources, resulting in non-equilibrium assemblages [118,198]. This dynamic was evident in the PLSF communities, where *Neosetophoma*, *Alternaria*, *Phoma*, and *Cladosporium* presented occurrences several orders of magnitude higher than those of the rare and infrequent taxa. This underscores the prominent role of a few dominant species in shaping the overall community structure. Furthermore, these results align with the statistically significant positive co-occurrence (CO) patterns, which primarily involved the most frequently documented mycota, hinting that dominant taxa may facilitate the colonisation by infrequent and rare taxa.

The logseries has long been recognised as a suitable model to describe the species abundance distribution of phylloplane and rhizoplane fungi, as plant-surface fungal assemblages are often regarded as non-equilibrium ecological units [117]. Similar patterns have been documented for fungal communities associated with leaf litter, e.g., ref. [199]. Therefore, it is not surprising that communities of PLSF can also be described by the logseries model. Additionally, the logseries model has been used to characterise samples from small, stressed, or pioneer communities [198,200]. It is noteworthy that foliar lesions are likely to represent stressed microhabitats, given that the host plant’s immune system attempts to suppress disease progression [201]. Consequently, host defences can act as a disturbance that prevents most fungal taxa from becoming abundant, with only a few species able to cope with the biotic stress, which results in a marked disparity in taxa abundances. Similar findings have been reported for xerotolerant and heat-stimulated soil fungi, which are favoured following fire disturbances and also conform to a logseries distribution [202].

The ecological implications of the logseries pattern observed here are profound, especially considering the restricted environment of the palm foliar lesions. It is likely that *Neosetophoma*, *Alternaria*, *Phoma,* and *Cladosporium* play key ecological roles either as primary pathogens, secondary invaders, or early colonisers, and various factors may be related to their dominance. Together, these four genera accounted for over 55% of all records of PLSF, indicating a sizable role in the establishment and expansion of foliar lesions. Notably, these genera are often reported in association with plant diseases, including leaf spots, particularly in plant tissues already under stress [24,203,204,205]. The necrotrophic behaviour of *Alternaria* species, for instance, has been widely documented and is essential for the pathogenesis and progression of associated diseases. Numerous recent outbreaks of *Alternaria* leaf spot diseases have been documented across a variety of plant hosts worldwide [206]. In date palms, *Alternaria* has recently been regarded as the most common pathogen [207], and other reports link it to diseases in additional palm host species, e.g., ref. [208]. The high prevalence of *Alternaria* in PLSF communities, as the second most frequently documented genus and with two very frequent MOTUs, underscores its significance as a potential phytopathogen associated with palm foliar diseases.

Species of *Cladosporium* and *Phoma* are similarly recognised as common as leaf-spotting fungi and agents of other plant diseases in economically important crops, causing significant yield losses, e.g., refs. [209,210,211,212,213]. A few previous records have also reported species of both genera in association with palm foliar diseases and healthy tissues, e.g., refs. [64,65,214,215,216,217,218], and therefore, the high prevalence of these fungi in the communities of PLSF was to be expected. Conversely, the high prevalence of *Neosetophoma*, a relatively recently introduced genus [219], suggests it has a more complex role in palm pathology than previously known. *Neosetophoma* was the most frequently documented genus, and it also presented the most frequently documented MOTU in the palm foliar lesions. While previous studies have reported *Neosetophoma* as a leaf-spotting pathogen of grasses and as an endophyte and saprophyte on a wide range of hosts, e.g., refs. [220,221,222,223], only one species has been documented in association with decaying palm material [224]. Therefore, the high prevalence of *Neosetophoma* in palm foliar lesions indicates a potentially significant role as a pathogen in palm foliar diseases that was previously unknown.

In contrast to *Neosetophoma*, species of *Alternaria*, *Cladosporium,* and *Phoma* are typically considered to be cosmopolitan, ubiquitous epiphytes and saprophytes, often acting as secondary invaders in stressed or weakened plant tissues [215,225,226]. Their relatively lower abundance compared with *Neosetophoma* further suggests differences in the pathogenicity of PLSF, reinforcing the role of *Neosetophoma* as an important potential pathogen in palm foliar diseases. Although species of *Alternaria*, *Cladosporium,* and *Phoma* are commonly reported in association with foliar diseases, including those affecting palms [26,27,28], they are likely opportunistic fungi, causing secondary infections in previously compromised tissues. The emergence of opportunistic fungi following primary infections of plant hosts is well documented [227,228]. For example, *Fusarium solani* has been identified as an opportunistic pathogen of the ornamental plant *Kalanchoe blossfeldiana*, manifesting only after prior infection by *Berkeleyomyces basicola* (syn. *Thielaviopsis basicola*) [229]. In such cases, the genera often linked to secondary colonisation or saprotrophy tend to thrive in environments where the host tissue integrity is already compromised, suggesting that their role within PLSF communities may be more related to decomposition than parasitism. This has been demonstrated for pestalotioid fungi, a group that is among the most ubiquitous within palm canopies and is frequently isolated from healthy tissues across a wide variety of palms. Pestalotioid fungi typically require wounds to infect plant tissues, acting as secondary invaders of previously established diseases by primary pathogens, which has been well-documented in *Elaeis guineensis* [26,42,230,231,232,233,234].

Primary pathogen infections or environmental stresses likely alter nutrient availability on the leaf surface, facilitating the proliferation of existing epiphytic or saprophytic fungi, which then colonise the leaf as secondary, minor pathogens that expand the already established lesion [227,228,235]. Several studies have documented similar pathogen interactions in co-infected plants, suggesting that positive CO patterns, as detected in this study, may reflect underlying synergistic relationships. For example, Tao et al. [176] found that the distribution of potentially pathogenic *Alternaria* on plant leaves determines the foliar fungal communities around the brown spot, with members of *Botryosphaeria*, *Paraphoma,* and *Plectosphaerella* acting as facilitators that exacerbate disease severity and expansion. Similarly, infection of maize by *Fusarium verticillioides* has been shown to facilitate subsequent colonisation by related fungi [236], and purple blotch disease of onion leaves has been found to stimulate colonisation by opportunistic fungi causing secondary infections [237]. In communities of PLSF, many infrequent and rare mycota may also represent primary pathogens, whose establishment is enhanced by the presence of common epiphytes and saprobes, which in turn facilitate the establishment of the former. Although the dominant mycota are the primary determinants of the community structure, the uncommon mycota may act as colonisation facilitators, and therefore both taxa engage in synergistic interactions that influence disease development.

The cosmopolitan nature of common saprophytes and epiphytes undoubtedly accounts for the high prevalence of certain taxa in the foliar lesions. This was clear when examining the relationship between the abundance of occurrence (AO) and frequency of occurrence (FO) of the most commonly documented taxa. While the AO reflects the overall prevalence of a given taxon in foliar lesions, the FO indicates how frequently it is observed across samples. Interestingly, within different abundance classes, the taxa with the greater AO were not necessarily the taxa with the greater FO, i.e., the most frequently observed. For instance, among very frequent mycota, *Neosetophoma* exhibited the highest AO but the lowest FO, being recorded in approximately 33% of the samples, while *Alternaria*, despite being less abundant overall, was recorded in more than 60% of the samples. A similar trend was observed among frequent mycota, with *Neodidymelliopsis* presenting the lowest FO yet one of the highest AOs, opposite to *Penicillium* and *Stemphylium*, which were recorded in more than 20% of the samples. As observed for *Neosetophoma*, *Neodidymelliopsis* is a relatively uncommon and recently introduced genus, primarily documented as a saprophyte [238]. On the other hand, *Penicillium* and *Stemphylium* represent ubiquitous genera encompassing plant pathogenic, endophytic, and saprophytic species with worldwide distributions [239,240]. The intricate and complex ecological patterns observed in PLSF reveal the multifunctional nature of these fungal assemblages and highlight the presence of important, hitherto unknown genera associated with palm foliar diseases. Among these are several coelomycete genera documented as well-known plant pathogens, which compose the long tail of rare taxa inhabiting palm foliar lesions.

#### 4.1.2. Ecological Significance of Uncommon Taxa and Impact of Knowledge Gaps in Palm Health

Rare taxa are increasingly recognised as crucial components in ecosystem functioning [241,242,243]. In this study, the rare mycota comprised over 80% of the documented genera and over 95% of the MOTUs, contributing considerably to the taxonomic diversity of the PLSF. Although the dominant mycota in PLSF communities were recognised as keystone taxa in community dynamics and overall structure, the uncommon mycota appear to engage in interactions that facilitate disease development. Many of these taxa included coelomycete genera widely recognised as plant pathogens, such as *Ascochyta*, *Colletotrichum*, *Coniothyrium*, *Didymocyrtis*, *Hendersonia*, *Neofusicoccum*, *Nothophoma*, *Paraconiothyrium*, *Parastagonospora*, *Phaeosphaeria*, *Phyllosticta,* and *Stagonosporopsis*, e.g., refs. [204,205,244,245].

Many *Stagonosporopsis* species are phytopathogens that can cause devasting diseases on plants from various families, e.g., refs. [246,247], yet associations between this genus and *Arecaceae* are virtually undocumented. Similarly, species of *Ascochyta* and *Nothophoma* occur mainly as pathogens with wide host ranges, but no records currently link them to palms, e.g., refs. [248,249]. Other genera, such as *Parastagonospora* and *Phaeosphaeria*, are often latent pathogens known to cause significant crop losses, e.g., ref. [250], and their associations with *Arecaceae* are limited to a few reports [205,251,252]. Likewise, genera such as *Colletotrichum* and *Neofusicoccum*, well-established phytopathogens affecting a wide range of hosts, are known for their heightened aggressiveness in plants under environmental or biological stress [253,254], yet their associations with *Arecaceae* have only recently been uncovered. For instance, several *Colletotrichum* species have been identified as anthracnose pathogens in areca palms in Hainan, China [255], while others have been associated with foliar diseases of other arecaceous hosts, e.g., refs. [74,256]. Similarly, the occurrence of *Neofusicoccum* and other *Botryosphaeriaceae* members on palms has been recently reviewed, and the lack of knowledge regarding the associations of these cosmopolitan fungi with *Arecaceae* has been emphasised [257].

The substantial presence of common phytopathogenic coelomycetes in PLSF, many of which have not been reported in association with *Arecaceae*, underscores the functional complexity of the fungal communities inhabiting palm foliar lesions and highlights the importance of the rare mycota in disease dynamics. Although the ecological roles of rare taxa remain largely unexplored, they are believed to enhance community stability by serving as a reservoir capable of rapidly responding to environmental changes [241]. For instance, despite their low abundance, rare taxa have been shown to play a pivotal role in driving fungal community responses to soil salinisation [258], shaping plant community composition in Alpine Grassland Soils [259], and regulating soil multifunctionality in *Eucalyptus* plantations [260]. Therefore, either as primary or latent pathogens activated by biotic or abiotic stressors, rare palmicolous coelomycetes may act as a reservoir of pathogenic potential within PLSF communities, becoming active as disease agents or facilitators under certain conditions. This hypothesis aligns with growing evidence that, while often overlooked, rare taxa hold critical functional roles within ecological systems [242].

Beyond their potential roles in palm foliar diseases, the substantial presence of a large number of rare taxa reflects a high taxon richness and diversity, which likely supports the resilience of PLSF communities through enhanced genetic and physiological diversity. Understanding the diversity and ecology of rare mycota is particularly important in the context of global climate change, due to which plant hosts increasingly face environmental pressures. In such scenarios, rare mycota with pathogenic potential may become ecologically important in biological interaction and successional dynamics, possibly leading to the emergence of previously unknown fungal pathogens [261,262]. For instance, Guo et al. [263] identified *Diaporthe ueckeri* as the causal agent of brown blotch disease on bottle palms in Guangdong, China, which severely impacts plant growth and ornamental value. *Diaporthe* was one of the infrequent taxa recorded in the present study, and its significance on palms has been recently revised by Pereira et al. [264] and Pereira and Phillips [265], who emphasised the limited understanding of its role in palm health. This knowledge gap applies to many coelomycetes documented in this study, which included not only the most frequent mycota but also numerous doubleton and singleton taxa that substantially contributed to the rich taxonomic diversity observed in the palm foliar lesions.

### 4.2. Taxonomic Structure of Palm Leaf Spotting Fungal Communities

*Arecaceae* hosts are widely recognised as hotspots of fungal diversity [45], a characteristic clearly reflected in the PLSF communities observed in this study. However, the taxonomic structure of the fungal assemblages associated with palm foliar lesions in Portugal differed markedly from the typical fungal assemblages found on palms in the tropics, hereafter referred to as tropical palm fungi (TrPF). A comparison of the most commonly documented fungal taxa of typical TrPF and PLSF at different taxonomic ranks is presented in Table 10. Before drawing any conclusions, it is important to note that all palms sampled in this study were ornamentals, which certainly has influenced the observed results.

As previously discussed, several coelomycetous taxa frequently documented in this study belong to recently introduced genera as well as genera that have not been (significantly) noted on arecaceous hosts before, despite their broad occurrence as saprophytes, epiphytes, and pathogens across various host families. In light of the influence of climatic factors in shaping differences among palm fungal assemblages, these genera are discussed here as potential components of an emerging assemblage of typical temperate palm fungi (TePF) (Table 10).

#### 4.2.1. Differences in the Typical Mycota of Tropical Palm Fungi

A distinctive assemblage of fungi in the phylum *Ascomycota* is consistently found in association with palms in the tropics, insomuch as they are often referred to as “palm fungi” or “palmicolous fungi” [45,266,267,268]. While the present study similarly identified ascomycetes as the predominant group among PLSF, notable taxonomic differences were observed compared with previously documented assemblages of palmicolous endophytes, saprobes, and pathogens (Table 10). Although a diverse array of *Ascomycota* species is associated with *Arecaceae* hosts, the majority are non-pathogenic to palms [27]. Consequently, the potential phytopathogenic fungi reported here were expected to differ from the typical assemblage of TrPF. Moreover, several genera identified here are common fungal pathogens with wide host ranges both within *Arecaceae* and other plant families, including, for instance, *Alternaria*, *Botryosphaeria*, *Bipolaris*, *Botrytis*, *Colletotrichum*, *Fusarium*, *Pestalotiopsis* and related genera, and *Phyllosticta* [27]. However, except for *Alternaria* and *Fusarium*, the remaining genera appeared as uncommon mycota in the present study, with a few recorded as singletons, such as *Botryosphaeria*, *Bipolaris*, and *Phyllosticta*. This observation raises questions about the consistency and representativeness of these genera as common leaf-spotting fungi on palm trees.

Reports of PLSF often seem to reflect specific cases where a certain fungal species or genotypic variant has become particularly virulent or produces only weak symptoms during pathogenicity testing, likely acting as a secondary invader of compromised plant tissues. For example, *Bipolaris heliconiae* was recently identified as the causal agent of frog-eye-like leaf spots on *Chrysalidocarpus lutescens* (syn. *Dypsis lutescens*) in Bangalore, India, illustrating the emergence of a novel, previously unreported disease [269]. Similarly, several species of *Neopestalotiopsis* have recently been implicated in leaf spot diseases of various *Arecaceae* hosts in nurseries in Brazil, uncovering new diseases affecting the commercial value of palms [270,271]. Pestalotioid species have also been regarded as weak pathogens of various arecaceous hosts in China [272]. Additionally, *Neopestalotiopsis* and *Colletotrichum* species were identified as the causal agents of leaf spots in *Chamaedorea costaricana* (syn. *Chamaedorea quezalteca*) plantations in Mexico [273]. Despite these documented cases, the genera involved were notably underrepresented in the foliar lesions of palms in Portugal. This observation could be attributed to climatic factors, as most reports are associated with tropical regions. However, given the cosmopolitan distribution of most associated genera, it is unlikely that climate alone explains their scarcity in this study. The limited occurrence of *Phyllosticta* in this study provides another striking example, as it is a geographically widespread genus of phytopathogens with a diverse host range [274]. Furthermore, *Phyllosticta* has been extensively documented in association with palms, with several species frequently implicated in leaf spot diseases, e.g., refs. [60,68]. It is worth noting that *Phyllosticta* species are also recorded as endophytes [274] inclusive in palm hosts, e.g., ref. [275], suggesting that their latent phytopathogenic nature may underlie many cases where *Phyllosticta* is implicated as a palm leaf spotting fungus.

Several other genera frequently identified as foliar fungal pathogens and commonly noted on palms were not recorded in the present study. These include *Cercospora* and its allied genera, *Calonectria*, and *Curvularia*. As with the previously discussed examples, most reports of these genera on palms come from tropical regions. Given their widespread distribution, however, the absence of representatives on palm foliar lesions in Portugal is surprising, especially considering the relatively extensive sampling conducted. For instance, several species of cercosporoid fungi have been documented as leaf-spotting pathogens on palms in Brazil [276], New Zealand [277], Puerto Rico [278], Thailand [59,279], the USA [280], and a few other tropical countries, although further taxonomic clarification is still needed for many of these species [281]. Similarly, *Calonectria* leaf spot is one of the most frequently documented diseases affecting a vast array of arecaceous hosts worldwide, impacting both ornamental and agriculture crops [43,282,283], with occurrences also documented on ornamental palms in Europe, e.g., ref. [284].

Few studies have investigated the fungi responsible for foliar lesions on palm trees in a manner comparable to the present research, which used comprehensive culture-dependent isolations. Consequently, comparisons of the mycota identified can be somewhat subjective. Most studies addressing PLSF have focused on economically important palms such as the date palm (*Phoenix dactylifera*), e.g., refs. [285,286,287,288,289,290,291,292], and the oil palm (*Elaeis guineensis*), e.g., refs. [63,293,294,295,296,297]. For instance, Al-Nadabi et al. [287] examined the fungi associated with 198 symptomatic leaf samples from date palm trees in Oman and found that *Alternaria* was the most prevalent genus, followed by species of *Curvularia*, *Nigrospora*, *Didymella*, and *Fusarium*. Recently, *Alternaria* has also been identified as the primary cause of leaf spots on date palms in Basrah, Iraq [292], and it is frequently regarded as a significant date palm leaf spotting fungus in various studies alongside species of *Bipolaris*, *Cladosporium*, *Curvularia*, *Fusarium*, *Helminthosporium*, *Neoscytalidium*, *Nigrospora*, pestalotioid fungi, *Phoma*, and many other genera; see [286,290,291] and references therein. In the case of *Elaeis guineensis*, most studies on leaf-spotting fungi have identified pestalotioid fungi and species of *Colletotrichum* and *Curvularia* as the principal causes of foliar diseases [233,296,297,298]. While several genera identified in these studies were also commonly found in the present study, the relative proportions of most of these fungi were notably low despite their widespread occurrence in nature, suggesting that PLSF communities in Portugal harbour a distinct, previously unreported assemblage of palm mycota. Furthermore, when comparing with fungal communities associated with leaf spots on other plant families, the present results were also particularly distinct, e.g., refs. [174,175].

Apart from the differences observed in widespread fungi, the differences observed between PLSF and TrPF are clearly shaped by climatic factors, reflecting distinct fungal diversity patterns in temperate versus tropical regions. Climate is widely recognised as a significant driver in various aspects of fungal biogeography, influencing not only the global distribution of common fungi but also the composition and diversity of fungal communities [299,300]. Accordingly, climate has proved to be an influential factor in shaping the diversity and distribution of fungi associated with palm trees [30], and the findings of this study are consistent with such observations. While *Linocarpaceae*, *Oxydothidaceae*, *Phomatosporaceae*, and *Xylariaceae* are major components of fungal assemblages on tropical palms [45], these families were absent from the assemblages of PLSF, which were primarily represented by members of *Didymellaceae*, *Phaeosphaeriaceae*, and *Pleosporaceae* (Table 10). Similarly, although TrPF typically belong to the orders *Amphisphaeriales*, *Chaetosphaeriales*, *Phomatosporales*, and *Xylariales* [45], these orders were either absent or poorly represented in the assemblages of PLSF, with approximately 74% of the assemblages comprising members from the *Pleosporales* (Table 10). Consequently, this led to *Dothideomycetes* being the most important class in PLSF, whereas typical TrPF are primarily composed of *Sordariomycetes*, although several *Dothideomycetes* taxa are also commonly found on palms [45] (Table 10).

Although one might assume that the differences observed between PLSF and TrPF simply reflect the type of assemblage sampled given that most studies on palm fungi primarily report on saprophytes, these differences are consistent even when examining other assemblages of palmicolous fungi, including endophytes and pathogens. Indeed, typical TrPF have also been reported in association with palm foliar diseases. For example, *Astrosphaeriella* and *Oxydothis* species have already been reported as palm leaf spotting fungi [50,52]. Research on the fungi associated with leaf spots of palms in north Queensland has also revealed a distinct assemblage compared with the PLSF identified in Portugal, with pestalotioid species predominating among the anamorphic fungi [57]. Notably, several leaf pathogens with a host range restricted to palms were also absent from the assemblage of PLSF in Portugal. For instance, none of the typical tar spot fungi associated with the *Arecaceae* family were detected in the present study, which was to be expected, as their geographic distribution is known to be quite limited and confined to tropical regions [37]. These include genera in the family *Phaeochoraceae* in *Phyllachorales*, such as *Cocoicola*, *Phaeochora*, *Phaeochoropsis,* and *Serenomyces*, which comprise saprophytic and biotrophic species known to occur only in association with palms, e.g., refs. [301,302,303,304]. Similarly, typical palm leaf pathogens such as *Annellophora phoenicis*, *Stigmina palmivora,* and species of *Exserohilum* were not detected in the present study, as these pathogens present a relatively limited geographic range confined to tropical countries [27,305,306]. For instance, *A. phoenicis* has been so far associated with collections from Malaysia, Myanmar, New Guinea, Panama, and Sierra Leone, with some doubtful reports from India, the USA (Texas), and Thailand [307]. The only typical palm leaf pathogen identified in the present study was *Graphiola phoenicis*, which was noted in several specimens of different *Arecaceae* hosts. However, the yeast state of *G. phoenicis* was never isolated from any lesions, suggesting that it does not compose the PLSF communities under investigation. *Graphiola* species are parasites almost exclusively associated with *Arecaceae* hosts, exhibiting a wider geographic range, and they are considered the sole genus of palm leaf pathogens in the phylum *Basidiomycota* [27,38].

Despite the substantial prevalence of *Pleosporales*, the assemblage of PLSF displayed a broad taxonomic distribution, encompassing several genera from a wide range of fungal orders and families that occurred infrequently. The relative proportion of *Ascomycota*, *Basidiomycota*, and zygomycetes in PLSF communities were comparable to those documented in previous studies on palmicolous fungi [45]. However, the basidiomycetes isolated in this study differed from those reported in the limited research available on palmicolous basidiomycetes, which has primarily focused on endophytic basidiomycetes associated with *Elaeis guineensis* [308,309]. These records underscore the lack of comprehensive studies on palmicolous basidiomycetes, as most of the basidiomycete taxa isolated here have not been previously reported from *Arecaceae*. It is likely that these basidiomycetes identified as part of the PLSF communities are endophytes that have transitioned to a saprophytic lifestyle following lesion establishment, as, apart from species of *Graphiola*, no basidiomycete has been found to cause leaf spots on palms. Previous research has already shown that fungal endophytes harbour a diverse range of fungi including latent pathogens and dormant endophytes, e.g., refs. [310,311,312,313,314]. Furthermore, recent studies on palmicolous fungi have also indicated that the spines of *Calamus castaneus* can harbour fungal pathogens of different crops as endophytes, including species of the genera *Colletotrichum*, *Diaporthe*, *Fusarium*, and *Neopestalotiopsis* [315], all of which were detected in the present study.

Palm leaf spotting fungal communities also differed substantially from those of palmicolous endophytes, but they were more similar to the communities documented on temperate palms than those found on tropical palms. Fröhlich et al. [184] observed that tropical palmicolous endophyte assemblages showed greater affinities to other hosts from tropical regions than to temperate hosts. Although both temperate and tropical palms share many endophyte genera, the relative proportions of these taxa are often quite different. Endophytic assemblages of tropical palms tend to be dominated by xylariaceous taxa, while the proportion of such endophytes in temperate palms is generally lower, a difference that has been attributed to climatic effects on fungal composition [184,316,317,318]. In contrast, Taylor et al. [318] found that the assemblages of palmicolous endophytes on the temperate palm *Trachycarpus fortunei* were dominated by species of *Colletotrichum*, *Diaporthe,* and *Phylosticta* and included common saprobes and pathogens that are isolated in both temperate and tropical regions but are less common in the latter. The low relative proportions of non-xylariaceous taxa reported in studies on temperate palms are reflected in the PLSF assemblages recorded from ornamental palms in Portugal.

It has long been asserted that the fungi associated with palms in their native habitats are not necessarily the same as those observed when they are moved into the landscape of temperate regions of the world [27]. Indeed, the assemblages of PLSF in Portugal presented more affinities to common genera identified in temperate palms and other temperate hosts and comprised several ubiquitous, plurivorous taxa often recovered from a variety of hosts across both temperate and tropical regions. Taylor et al. [30] showed that fungi associated with palms in their native habitats consistently correlated with multiple palm species, especially in tropical regions. However, the fungal assemblages found on the same palms grown outside their tropical range were composed of relatively more ubiquitous fungi with a much broader host range across different plant families. In addition, according to Taylor et al. [30], in temperate regions, the dominant tropical palm mycota were replaced by more ubiquitous, plurivorous taxa, along with fungi from other groups such as coelomycetes. For instance, the fungal assemblage of the temperate palm *Trachycarpus fortunei* was composed of more temperate and widespread taxa [30], consistent with the results of the present study. While palms are naturally adapted to tropical and subtropical climates, those in Portugal are subject to temperate conditions, resulting in a mycobiota reflective of temperate fungal diversity. Therefore, investigating the fungal diversity on palms in Portugal provides valuable insights into the assemblage of palmicolous fungi in a temperate climate context. Two major groups of mycota can be recognised in the assemblage of PLSF in Portugal: (a) widespread, plurivorous genera such as *Alternaria*, *Aspergillus*, *Chaetomium*, *Cladosporium*, *Diaporthe*, *Epicoccum*, *Fusarium*, *Penicillium*, *Phoma*, *Stemphylium*, and numerous ubiquitous genera, particularly hyphomycetes, with limited occurrences; and (b) uncommon or ubiquitous coelomycetous genera such as *Neosetophoma*, *Neodidymelliopsis*, *Didymella*, *Libertasomyces*, *Sclerostagonospora*, *Keissleriella*, and various other genera with limited occurrences. The latter group, which includes common coelomycetes often associated with plant diseases, has not been previously noted as a distinct assemblage on palms. However, it appears to compose a relevant component in the present study, potentially representing an assemblage of typical TePF.

#### 4.2.2. Coelomycetous Taxa as Components of Temperate Palm Fungal Assemblages

Host specificity and host recurrence are often documented in assemblages of palmicolous fungi in the tropics [9,45]. However, considering the widespread nature of the fungi identified in palm foliar lesions in Portugal, it seems unlikely that the great majority of them are specific to temperate palms. Only a few taxa, which occurred with limited frequency, might be indicative of host specificity towards *Chamaerops humilis*, the sole palm species native to the Mediterranean region sampled in this study. For example, *Palmeiromyces chamaeropicola*, documented here as *Palmeiromyces* MOTU 203, was published as a new *Teratosphaeriaceae* taxon in Pereira and Phillips [69] and is a candidate for such host specificity. Members of *Teratosphaeriaceae* and related families are frequently regarded as highly host-specific [319], and *P. chamaeropicola* was proposed to be an obligate biotroph based on its growth characteristics, reinforcing its potential specificity towards *C. humilis*. However, any conclusions regarding the ecology and physiology of *P. chamaeropicola* remain premature, as its current known distribution is limited to a single plant in Lisbon [69].

Although unlikely to be host-specific to palms in temperate regions, several coelomycetous taxa identified in the present study exhibited notable recurrence in diseased foliar tissues of ornamental palms. This observation is particularly noteworthy, as some of these taxa are either widespread, speciose, or have only recently been introduced, with very limited or no prior documentation of their association with *Arecaceae*. Genera such as *Sclerostagonospora* and *Keissleriella* are predominantly temperate, which may explain their scarcity in fungal assemblages of tropical palms. Furthermore, along with *Neodidymelliopsis* and *Libertasomyces*, they are relatively uncommon genera with only a few species described, which might also account for their apparent lack of association with *Arecaceae*. In contrast, *Didymella* is a speciose genus with a worldwide distribution, commonly found as saprobes on dead plant material and as opportunistic pathogens of herbaceous and woody plants across a broad range of hosts [245]. However, their association with arecaceous hosts appears to be relatively rare. For instance, *D. sinensis* was collected by Tennakoon et al. [320] from dead leaves of *Roystonea regia* in Taiwan, a species that has otherwise been reported in association with only a few other hosts. Likewise, *Neosetophoma*, although a recently introduced genus with around 30 described species, has been reported from a wide range of hosts, but its association with *Arecaceae* is restricted to *N. trachycarpi* on decaying rachides of *Trachycarpus fortunei* from Guizhou, China [224]. Interestingly, *T. fortunei* is a palm native to warm temperate regions of China, which further supports a close association between *Neosetophoma* and related coelomycetous taxa with temperate palms or palms thriving in temperate environments.

The high prevalence of coelomycetous taxa in palm foliar lesions in Portugal reveals a previously undocumented fungal assemblage of palmicolous fungi. Despite the dominance of mitosporic fungi in PLSF communities, coelomycetes were more abundant, accounting for 56% of all fungal records, which also reflected a higher taxon richness at both the genus and MOTU levels. Taxon richness and abundance are key components in shaping the diversity of the fungal communities associated with palm foliar lesions. While both hyphomycetes and coelomycetes contribute to the pronounced dominance indicated by the steeper slopes of the abundance distributions, coelomycetes contribute more to the shallower slopes that reflect the profusion of rare mycota. As previously discussed, the ecological importance of rare taxa is increasingly being recognised [241,242,243], and the rare coelomycetous taxa documented here may function as reservoirs of pathogenic potential, possibly playing critical roles in disease establishment and progression.

One of the main differences noted by Taylor et al. [30] concerning the biogeographical distribution of microfungi associated with palms in tropical versus temperate habitats was the relatively higher abundance of coelomycetes in temperate regions alongside only a few typical TrPF. The present findings align with those of Taylor et al. [30] and further suggest that coelomycetes may represent a characteristic component of TePF assemblages. The ratio between the relative abundances of coelomycetes and hyphomycetes found in this study (1.44) is remarkably similar to the ratio found by Taylor et al. [30] for temperate regions (1.32, based on data presented by Taylor et al. [30] for *T. fortunei*). This similarity is particularly noteworthy given that these ratios were derived from considerably different contexts: (a) while the present study included multiple ornamental palm species in a temperate climate setting, the value extrapolated from data in Taylor et al. [30] is based solely on *T. fortunei*, used as a model species to study palm fungi biodiversity in temperate regions; (b) the present study focused on fungal communities involved in palm foliar diseases, whereas Taylor et al. [30] examined fungal communities found on different decaying parts of palms.

The coelomycete assemblage was found to be more diverse and more evenly distributed compared with hyphomycetes, which implies greater genetic and physiological diversity. However, they are likely to be less generalist due to their closer association with arecaceous hosts, as indicated by the taxonomic structure and diversity of the most commonly documented genera. At higher taxonomic ranks, coelomycetes presented a narrower taxonomic distribution, with most taxa affiliated with families of the *Pleosporales*, pointing to a more specific mycota compared with hyphomycetes. Furthermore, by frequency group, the hyphomycetes displayed greater genetic diversity, as evidenced by a relatively higher number of MOTUs and genomic types (GTs) per MOTU among the most commonly documented genera. Given the congruence of MOTUs with interspecific diversity, the most common genera of hyphomycetes in palm foliar lesions appear to harbour more species and show higher intraspecific diversity compared with coelomycetes. This pattern likely reflects the more ubiquitous and cosmopolitan nature of most hyphomycetes documented, such as *Alternaria*, *Cladosporium*, *Fusarium*, *Penicillium*, and *Aspergillus*, which exhibit greater intraspecific diversity due to their common association with a wide range of hosts and adaptation to diverse environments [321].

Genetic diversity, expressed in terms of MOTUs and GTs, was assessed using genomic fingerprinting techniques, specifically microsatellite/minisatellite primed (MSP)-PCR. MSP-PCR fingerprinting has demonstrated utility in the molecular typing of fungal strains, and its application in the identification of yeast and filamentous fungi species has been extensively documented, e.g., refs. [98,103,322,323,324,325,326,327,328,329,330,331,332]. For instance, Alves et al. [98] reported that MSP-PCR and related PCR-based fingerprinting techniques enabled the differentiation of 27 species in the *Botryosphaeriaceae*. In this study, genomic fingerprinting using csM13 and (GTG)_5_ primers was employed as a proxy to estimate interspecific diversity (via MOTUs) and intraspecific diversity (via GTs). Both primers produced highly complex and discriminatory band profiles, but (GTG)_5_ proved more robust in delineating MOTUs compared with csM13. Nevertheless, the combined analysis of the profiles of both primers yielded greater discriminatory power, enabling the resolution of clusters that could not be differentiated when the primers were used individually. This outcome aligns with expectations, as the discriminatory capacity of each primer is influenced by the specific taxonomic group under investigation, and polyphasic approaches generally exhibit enhanced differentiation capabilities among microbial isolates [333,334].

Although phylogenetic analyses are required to confirm the correspondence between MOTUs and species, preliminary findings suggest a strong correlation, indicating that most MOTUs, i.e., clusters formed at similarity values ≥ 75%, likely represent distinct species, consistent with previous studies. For example, Alves et al. [98] showed that clusters formed at similarity values ≥ 80% corresponded to known *Botryosphaeriaceae* species. Therefore, it is reasonable to assume that the biodiversity analyses previously discussed likely reflect ecological scenarios at the species level. Moreover, they are supported by a substantially low isolation redundancy coefficient (IRC) of 6.54%, which accounted for 135 fungal records. Most of these fungal records were from the most commonly documented MOTUs of *Alternaria* and *Neosetophoma*. Given that the occurrence of these MOTUs is several times higher than that of the others, it can be assumed that the previous ecological observations pointed out were effectively based on the isolation of effectively different fungal records and that the associated error is negligible. Therefore, despite the fact that some of the *Alternaria* and *Neosetophoma* records were derived from redundant isolations, the considerable number of distinct fungal records for these genera still indicates that they were the most prevalent and abundant in the fungal communities under investigation. It appears that the calculation of IRC based on genomic fingerprints may prove an invaluable tool in the field of ecological studies. When combined with suitable molecular markers, the IRC can serve as a fine-tuning tool in the analysis of fungal community structure.

It is worth nothing that the number of true species is likely underestimated, as initial morphological analyses were used to group fungal records by genus, which were subsequently subjected to hierarchical cluster analysis (HCA) of genomic fingerprints. However, MSP-PCR and related fingerprinting methods analyse highly variable genomic regions, making them more suitable for taxonomic studies at or below the species level [96,335]. Consequently, due to cryptic speciation, it is probable that distinct species have been grouped within the same MOTU and that distinct genera have been erroneously assumed to belong to the same genus based on morphological analyses, particularly in coelomycetous genera that lack clear morphological distinctiveness.

The coelomycetes that appear characteristic TePF assemblages include several families of *Phoma*-like taxa, whose classification remains controversial and is still under ongoing revision [336,337]. Therefore, it is likely that the scarcity of some of these fungi on palmicolous assemblages is also influenced by inaccurate documentation of *Phoma*-like species, often resulting from outdated identification approaches such as relying solely on morphological characteristics. *Phoma*-like taxa identified in the present study as prominent palmicolous fungi include members of the *Didymellaceae*, such as *Didymella* and *Neodidymelliopsis*, along with several less abundant taxa, including *Allophoma*, *Ascochyta*, *Dimorpha*, *Ectophoma*, *Nothophoma*, and *Stagonosporopsis*; members of the *Phaeosphaeriaceae*, including *Neosetophoma* and *Sclerostagonospora*, as well as less abundant taxa, such as *Didymocyrtis*, *Hendersonia*, *Neostagonospora*, *Parastagonospora*, and *Phaeosphaeria*; and other taxa like *Keissleriella* (*Lentitheciaceae*), *Libertasomyces* (*Libertasomycetaceae*), and *Coniothyrium*-like taxa such as *Paraconiothyrium* (*Didymosphaeriaceae*) and *Coniothyrium* (*Coniothyriaceae*). These taxa, along with other related families in *Pleosporales*, are poorly documented on palms. For instance, *Coniothyrium palmarum* was frequently reported in association with leaf spots on *Phoenix dactylifera* in India and Cyprus [85], and the isolates commonly used to represent *C. palmarum* in phylogenetic studies were isolated from a dead petiole of *Chamaerops humilis* in Italy (May 1971) and from leaf spots on *Phoenix dactylifera* in Israel (August 1973) [338]. However, since these early collections, no new isolates of this species have been reported, insomuch as there is no DNA sequence available for the type specimen of *C. palmarum*. Therefore, the systematics of *Coniothyrium* remain unclear. Although some recent collections of palmicolous coelomycetes have been made, they mostly do not represent the prevalent mycota found in palm foliar lesions in Portugal. For instance, *Phaeosphaeria nodulispora* was reported as an endophytic fungus on leaves of *Cocos nucifera* in Pernambuco, Brazil [339], and *P. hyphaenes* was documented on leaves of *Hyphaene* sp. in South Africa [252]. In the present study, *Phaeosphaeria* was a rare taxon. Likewise, *Parastagonospora phoenicicola* on *Phoenix canariensis* and *Wojnowiciella dactylidis* on *Dypsis* sp. were documented by Crous et al. [251,340] in New Zealand, and both genera represented rare taxa in this study, with *Wojnowiciella* being a singleton. More recently, Xiong et al. [341] reported the first instance of a *Paraconiothyrium* species on palms, namely *P. guangdongensis* on a rotting leaf of *Licuala peltata* var. *sumawongii* in China. *Paraconiothyrium* was also a rare taxon in the present study. Similarly, Kularathnage et al. [342] reported *Ectophoma phoenicis* on dead petioles of *P. roebelenii* in China, which was another rare, doubleton genus identified in this study.

To better understand the role of pleosporalean coelomycetes as prominent components of palm fungal assemblages, more systematic collection of palm fungi in temperate regions is required, including taxa from different ecological groups such as saprobes and endophytes. Additionally, the status of the palm host species should not be disregarded, as most palms studied here are grown solely as ornamentals in Portugal, which is likely affecting the results observed. Therefore, a survey of the fungal assemblage of wild representatives of *C. humilis*, which is native to the southernmost region of Portugal, would be essential to draw more robust conclusions regarding the composition of TePF.

### 4.3. Host-Related Patterns in Communities of Palm Leaf Spotting Fungi

Differences in palm fungal assemblages between temperate and tropical regions have been found to be more related to climatic influences than to the hosts sampled [30]. Recent surveys across various ecosystems also suggest that the biodiversity and specificity of palm fungi reflect more the habitats surveyed than the hosts sampled or geographical distribution [224]. However, several instances of host specificity and host recurrence demonstrate that palm host species also impact the composition of palmicolous fungal assemblages [45]. Accordingly, the present study revealed that palm host species significantly influenced the composition of the PLSF communities, accounting for up to 14% of the observed variation. Moreover, statistical analyses indicated that the composition of the fungal assemblages from the foliar lesions of combinations of different palm host species were significantly different, further underscoring the influence of host identity on fungal colonisation.

The palm host species sampled in this study exhibit considerable variation in their morphology, ecology, and historical biogeography [17], factors that have already been shown to influence the composition of palm-associated mycota [45,266]. *Chamaerops humilis*, commonly known as the European fan palm, is native to Europe and North Africa, typically growing in shrubs and rocky coastal outcrops around the western Mediterranean Basin. It is one of the most cold-hardy palms and is widely used in landscaping in temperate regions [17]. *Trachycarpus fortunei*, commonly named the Chinese windmill palm, is native to warm temperate China and is considered the most cold-hardy palm worldwide, thriving even in mountainous regions with severe winter conditions [17]. Due to its popularity as an exotic ornamental plant throughout European countries and beyond, *T. fortunei* has become naturalised in several regions of the world where it has been introduced, e.g., ref. [343]. *Chamaerops humilis* and *T. fortunei* were the only palm host species sampled that are native to temperate climates. *Chrysalidocarpus lutescens*, commonly known as the areca palm, is a tropical and subtropical palm species native to eastern Madagascar, where it thrives in moist rainforests and along riverbanks. It is often cultivated as a houseplant in warm temperate climates [17]. *Phoenix canariensis*, known as the Canary Island date palm, is native to the Canary Islands, growing on a wide variety of soils in riparian areas. Its adaptability to diverse soil types and its tolerance to drought make *P. canariensis* a popular ornamental in warm temperate regions [17]. *Phoenix dactylifera*, the date palm, is among the most economically important palm species and is believed to originate from North Africa and the Middle East, where it thrives in dry, desert climates [17]. Similarly to *P. canariensis*, *P. dactylifera* has been widely introduced over centuries and is one of the most grown and appreciated ornamental palms worldwide. Both *P. canariensis* and *P. dactylifera* are becoming naturalised in numerous regions globally, including Portugal [344,345]. *Phoenix reclinata*, commonly referred to as the Senegal date palm, is native to tropical Africa, the Arabian Peninsula, and Madagascar, where it grows in humid lowland woodlands, highland forests, and on open rocky hillsides. Although it is also cultivated as an ornamental in temperate countries, *P. reclinata* is primarily used as an ornamental palm in tropical and subtropical regions and has become naturalised in only a few countries outside tropical Africa [17].

Despite their varying climatic requirements, all the palm species examined are cultivated as ornamentals in warm temperate climates such as that of Portugal, and they were sampled in parishes of Lisbon and Oeiras. The diversity in their native climates, ecological preferences, and morphologies among the palm species suggests that differences in fungal communities could be influenced by the host-specific environmental conditions and adaptations of each species. To evaluate host-related differences in PLSF communities, biodiversity analyses were conducted based on the fungi isolated from samples of each host species. The results observed must be interpreted with care, as the apparent biodiversity trends may be a product of biased sampling. Overall, *C. humilis* was found to harbour the most diverse fungal communities, followed by *C. lutescens*, *P. reclinata*, and *T. fortunei*. In contrast, the foliar lesions of *P. canariensis* and *P. dactylifera* exhibited the least diverse fungal communities, with particularly depauperate fungal assemblages. Notably, the differences in fungal diversity between the foliar lesions of *P. canariensis* and *P. dactylifera* and those of the other palm hosts were found to be mostly statistically significant, indicating that these two tropical palms support markedly less diverse fungal communities under the temperate conditions in which they were sampled.

Although native to tropical and subtropical regions, *P. canariensis* and *P. dactylifera* have been cultivated for centuries, resulting in naturalised populations now established in Portugal and other temperate countries. The proliferation of these palms beyond their natural geographic range likely contributes to the observed depauperate and poorly diverse fungal assemblages. This biodiversity trend is especially noteworthy in *P. canariensis*. Despite one of the highest sampling efforts, fungal communities associated with this species were among the least diverse, further supporting that the observed results are driven by host-related factors rather than merely reflecting sampling bias. Such phenomena of depauperate fungal assemblages on naturalised hosts outside their native range are well documented in various plant–fungal associations, including palms. For instance, fungal endophytes from leaves and twigs of *Quercus ilex* were found to be less diverse in naturalised stands compared with native ones and were dominated by cosmopolitan and non-specific taxa [346]. Similarly, ectomycorrhizal fungal communities associated with pines frequently exhibit reduced diversity compared with those within their native range [347]. Comparable findings have also been reported for endophytic and saprophytic assemblages of palm hosts by Taylor et al. [30,318]. For example, the plurivorous nature of the fungi documented on *Cocos nucifera* by Taylor et al. [30] has been linked to its long history of cultivation. Most previous studies on the fungal assemblages of *P. dactylifera* and *P. canariensis* have primarily focused on fungal pathogens. As previously discussed, the limited overlap between the typical foliar pathogens of these two palms and the fungi documented in the present study suggests a climatic influence, especially as most previous research has been conducted in countries where these palm species are native, e.g., refs. [290,291,292]. Additionally, arbuscular mycorrhizal fungal communities in date palm plantations and the natural vegetation of surrounding desert habitats of Southern Arabia were found to be quite diverse and distinct from those documented on other native vegetation [348]. While not entirely a directly comparable example, this investigation demonstrates that *P. dactylifera* can support highly diverse and unique fungal communities when grown within its natural geographic range. Limited research has been conducted on fungal communities of *P. dactylifera* and even less on *P. canariensis*. Therefore, further studies are needed to better understand the results of the present study.

The exceptional diversity observed in the fungal communities associated with *C. humilis* further supports the role of climate in influencing the biodiversity of palm fungi. Notably, *C. humilis* was the only palm host species in this study that occurs naturally in Portugal, thriving along the southern coasts of Algarve and appearing sporadically on the Alentejo coast [349]. Previous research on the ascomycete fungi associated with *C. humilis* has documented the presence of a few typical TrPF, such as *Anthostomella*, *Frondisphaeria*, *Arecophila*, and *Phaeochora*, as well as several temperate and widespread taxa [267]. In the present study, only a single fungal record, *Fasciatispora* MOTU 122, was identified as a typical TrPF. The remaining taxa consisted of widespread, plurivorous genera and, especially, uncommon and ubiquitous coelomycetes genera, which have been considered here as characteristic of TePF. Genera such as *Neosetophoma*, *Libertasomyces*, *Neodidymelliopsis*, *Didymella*, and *Sclerostagonospora* were highly prevalent in the PLSF communities of *C. humilis*, which were predominantly composed of coelomycetes. In addition, *C. humilis* was the host species with the highest number of unique taxa, comprising 14% of all genera and 24% of all MOTUs. Given that *C. humilis* is a temperate palm cultivated as an ornamental in a country where it naturally occurs, it is not surprising that its fungal communities harbour a characteristic and distinct mycota. Thus, despite being depauperate in typical TrPF, the foliar lesions of *C. humilis* seem to harbour a coelomycetous assemblage typical of temperate palms or temperate climates, as suggested by previous studies [30,267]. A similar trend was observed in *T. fortunei* and *P. reclinata*, which, along with *C. humilis*, were recognised as the group of palms in this study (group A) enriched with coelomycetes.

*Trachycarpus fortunei* is also considered a temperate palm, insomuch as it has been previously used as a case study to investigate the composition of palm fungi in temperate climates. Taylor et al. [30] showed that, although some typical TrPF were present in *T. fortunei*, its mycota was predominantly composed of ubiquitous fungi, with an enrichment in coelomycetes compared with the fungal assemblages found on tropical palms. Similarly, studies conducted on the endophytic fungi associated with *T. fortunei* revealed an assemblage with more affinity with unrelated temperate hosts than with tropical palm hosts [318]. The results obtained here align with those of Taylor et al. [30,318]. No typical TrPF were detected in *T. fortunei*, and instead, its mycota was dominated by coelomycetous taxa and was more similar to that documented in fungal communities of *C. humilis*, with a high prevalence of *Neosetophoma* and *Neodidymelliopsis* as representatives of TePF.

Contrary to expectations for a tropical palm, *P. reclinata* exhibited a mycota qualitatively similar to that of temperate palms like *C. humilis* and *T. fortunei*. Specifically, an enrichment in coelomycetous taxa was observed, with a high prevalence of *Neosetophoma*, *Keissleriella*, and *Sclerostagonospora*, which are characteristic of TePF. All the specimens of *P. reclinata* were sampled from the Parque das Nações parish in Lisbon, where most of the specimens of *C. humilis* were also found. In Parque das Nações, palms were planted in densely populated green spaces alongside a variety of plant species, including other arecaceous hosts, designed to create an exotic landscape architecture. This close proximity between different plant species, particularly *P. reclinata* and *C. humilis*, likely facilitated the exchange of fungal spores, which could explain the similarities between the mycota of *P. reclinata* and that of the temperate palms. Indeed, *C. humilis* and *P. reclinata* exhibited the highest value of overlapping MOTUs among all sampled palm species, and several TePF were found to be relatively more speciose in *P. reclinata*. Only a single taxon, *Fasciatispora* MOTU 122, represented by two fungal records, was identified as a typical TrPF in *P. reclinata*. The detection of this taxon on *C. humilis* might also reflect the close proximity of specimens of these two host species. However, drawing any significant assumptions would require further collections and examination of *Fasciatispora* MOTU 122. This taxon was detected only on *P. canariensis* (as a singleton) and *P. dactylifera*, hinting at a potential remnant of a tropical fungal association within a temperate environment. Due to the limited research on the fungal assemblages of *P. reclinata*, further conclusions remain premature. Very few studies have documented fungi on *P. reclinata*, and the literature offers only isolated reports, e.g., refs. [350,351,352], underscoring the need for more systematic investigations of fungal associations with this arecaceous host.

A similar lack of systematic research into fungal associations is evident for *C. lutescens*, with few isolated reports, e.g., refs. [269,279,353,354]. *Chrysalidocarpus lutescens*, unlike *P. reclinata*, conformed to expectations for a tropical palm by exhibiting a mycota qualitatively similar to that of tropical palms such as *P. canariensis* and *P. dactylifera* (regarded here as group B). Specifically, an impoverishment in coelomycetous taxa was observed, with a high prevalence of widespread, plurivorous genera such as *Alternaria* and *Cladosporium*. In addition, some non-mitosporic fungi were among the top ten taxa in palms of group B, including *Chaetomium*, *Pyronema*, *Fasciatispora,* and *Nothodactylaria*. These taxa might represent remnants of a tropical fungal association as previously discussed for *Fasciatispora*, a typical TrPF [45]. A remarkable distinction between groups A (temperate palms and *P. reclinata*) and B (tropical palms) was the prevalence of *Neosetophoma*. Although *Neosetophoma* was recorded as the most abundant genus in palm foliar lesions in Portugal overall, it was markedly underrepresented or nearly absent in tropical palms, including *Washingtonia filifera*, which was excluded from the main analyses due to a limited number of samples.

Tropical palms were not only depauperate in typical TrPF, but also depauperate in typical TePF, ultimately harbouring fungal assemblages dominated by more ubiquitous, plurivorous taxa that are not characteristic of any specific climate or region. This pattern is evidently a climatic influence and conforms with previous studies on palmicolous assemblages of tropical palms growing outside their natural ranges [30]. Indeed, the HCA based on the AO of MOTUs provided further support for this differentiation. The HCA grouped the palm host species into two groups that were highly congruent with their geographical distribution and climate, with temperate palms (*C. humilis* and *T. fortunei*) clustering together, clearly separate from the tropical palms, including *P. reclinata*. Interestingly, while the fungal assemblages of *P. reclinata* appeared qualitatively more similar to those of temperate palms, the quantitative analyses indicated greater similarity to the tropical palms (*C. lutescens*, *P. canariensis*, and *P. dactylifera*) when the fungal assemblages were analysed as a whole. It should be noted that the fungal assemblages of the tropical palms *C. lutescens* and *P. reclinata* were also highly diverse, albeit not as diverse as those observed in *C. humilis*. This suggests that, despite the climatic mismatch between these tropical palms and the temperate environment in which they are cultivated, they still manage to host a wide variety of fungal taxa, an aspect consistently documented in studies of fungal colonisation on arecaceous hosts [45]. For example, *C. lutescens* and *P. reclinata* presented a similar number of unique taxa to *T. fortunei* and a higher number compared with *P. canariensis* and *P. dactylifera*, which were remarkably impoverished in fungal diversity. Moreover, the high fungal colonisation of *C. lutescens* and *P. reclinata* resulted in relatively high values of fungal taxa overlap with temperate palms and, in some instances, even higher overlap than that found between the temperate palms themselves, as they shared the cosmopolitan mycota documented. Nevertheless, the overall diversity levels of these tropical palms did not reach the richness documented for *C. humilis*, possibly reflecting the native status and ecological adaptation of *C. humilis* to the local environment.

Although most differences between samples of temperate and tropical palms reside in the infrequent and rare mycota, especially coelomycetous taxa, it is noteworthy that some of the most frequently documented taxa also contribute to the distinction between fungal assemblages of these two groups of arecaceous hosts. The high variability among fungal communities across different samples results in these host-specific differences being somewhat diluted at the sample level, even though they become apparent when samples are analysed as a whole. This was evident in the principal component analysis (PCA), which revealed considerable overlap between samples from different host species, indicating that the poor fit of the model allows for only broad and cautious interpretations. Most of the observed grouping of samples was related to *Neosetophoma* MOTU 188 and MOTU 185 and to *Alternaria* MOTU 015, the latter two exhibiting a negative correlation. Specifically, while *Neosetophoma* MOTU 188 appeared to be more prevalent in temperate palms, particularly *C. humilis*, *Neosetophoma* MOTU 185 was more prevalent in tropical palms, including *P. canariensis* and *P. reclinata*. Likewise, *Alternaria* MOTU 009 was more prevalent in tropical palms, especially *P. canariensis* and *P. dactylifera*, whereas *Alternaria* MOTU 015 was prevalent in temperate palms, along with *Alternaria* MOTU 009. Nonetheless, these biodiversity trends are only apparent and not yet conclusive. Further collections, including other ecological groups such as saprophytes and endophytes, would help to clarify whether certain MOTUs are genuinely more prevalent in temperate palms due to biological constraints related to the biogeography of palm host species. Additionally, identification of these MOTUs to the species level using contemporary molecular methods is essential to determine whether they represent distinct species or genotypic variants of the same species resulting from adaptive divergence to hosts with different climatic constraints.

The results of this study are preliminary, yet they provide valuable insights into the complex interactions between PLSF and their hosts. Although a thorough sampling effort was undertaken, many fungal representatives within the foliar lesion environment likely remain uncollected and uncharacterised, as evidenced by the species accumulation curves (SACs) plotted for both genera and MOTUs. Indeed, none of the curves levelled off, which suggests that a more comprehensive effort is needed to capture the entire diversity of PLSF communities. Nonetheless, the sample size was sufficiently large to reveal illustrative results that express discernible ecological and biodiversity trends in each palm host species and across the overall fungal community. Extrapolated taxon richness estimates for the overall community indicate that around 70% of the potential diversity of the genera and MOTUs was recorded. The SACs for the entire PLSF community displayed an asymptotic trend, suggesting substantial representation of fungal diversity, particularly at the genus level, with a more pronounced inflection point in the genus accumulation curve compared with the MOTUs. This suggests that the communities of PLSF are more thoroughly sampled at the genus level than at the MOTU level. In addition, the wider confidence intervals in the SACs for MOTUs, particularly for individual palm host species, imply greater variability in species composition compared with genera, suggesting that additional sampling could yield further novel MOTUs while genus-level diversity may be approaching saturation. Drawing definitive conclusions is premature, as nonparametric models predicted highly similar ranges of unrecorded diversity and also failed to level off, making these estimates useful but not conclusive indicators of the potential fungal diversity present. Furthermore, several studies have already discussed that such diversity estimators present several limitations and should be used cautiously [355,356,357,358]. The taxa yet to be recorded are expected to be mostly infrequent and rare, as the community was best described by a logseries distribution with a long tail of rarely recorded genera. It is plausible that many of these unrecorded genera will be coelomycetous fungi, as the present study demonstrated that coelomycetes constitute the majority of infrequent and rare genera and contribute significantly to the high fungal diversity observed.

Given the important role of rare mycota in the fungal diversity of PLSF communities, it is more accurate to use species richness estimators that account for the informative power of rare taxa regarding the number of undetected species. Among the nonparametric estimators applied in the present study, Chao 1, abundance-based coverage (ACE), and jackknife estimators are particularly suitable for datasets skewed towards low-abundance species [118,120,151,359] characteristic of the PLSF fungal communities under investigation. The Chao 1 is designed to infer the presence of unseen species primarily based on the number of rare species such as singletons and doubletons. It is particularly effective in datasets where rare species dominate and where species abundances are uneven [143,148,151]. Similarly, the second-order jackknife estimator accounts for species richness based on the frequency of singletons and doubletons, offering a refined estimate compared with the first-order jackknife, which ignores the doubletons in its estimations. This estimator also provides a more accurate estimation of the unseen diversity in situations where the abundance of rare taxa is considerable [145,146,151,360]. The ACE estimator accounts for species richness by dividing the species into rare and abundant categories based on a pre-set threshold, focusing on taxa below this threshold. It is particularly suitable for datasets with numerous low-abundance species, as it explicitly accounts for these by categorising them separately, thereby providing robust estimates for communities with both rare and more common taxa. The robustness of the ACE is particularly beneficial in diverse and complex ecological settings such as foliar lesions, where the abundance of fungal species can vary widely [148,149,150,151,361]. In the present study, the estimates for taxon richness derived from Chao 1, ACE, and second-order jackknife estimators were notably similar, suggesting that these methods provide a good approximation of the true diversity present in the PLSF communities. Notably, all three estimators yielded higher estimates of fungal diversity compared with the first-order jackknife and the bootstrap estimator. The bootstrap estimator, which relies on repeated random resampling to calculate species richness and the likelihood of resampling each species [147], tends to perform well when the dataset is relatively well sampled and lacks a strong bias towards rare species. However, it can substantially underestimate species richness in cases like the present study, where the community structure is dominated by rare taxa [118,151]. Consequently, it is likely that the lower estimates derived from the first-order jackknife and bootstrap represent underestimations of the actual diversity in these communities given the high proportion of rare taxa. These findings emphasise the importance of applying estimators that adequately address the specific characteristics of the fungal community, particularly those skewed towards low-abundance species, to ensure the accuracy of the biodiversity assessments.

Evaluating the adequacy of sampling size is of paramount importance in the present study, given that all the biodiversity patterns and ecological observations may be influenced by sampling bias. Most collections were conducted randomly, both in terms of collection sites and the host species, which may not fully represent the true diversity. While ornamental palms are commonly distributed throughout Portuguese cities, their canopies are often difficult to access due to their substantial height, limiting comprehensive inspection for foliar diseases. This limitation could affect the representativeness of the sampled material, potentially leading to an underestimation or skewed understanding of the true diversity and ecological interactions of the associated fungal communities. Addressing these constraints is essential to ensure that the observed patterns genuinely reflect the underlying ecological dynamics rather than artefacts of insufficient or biased sampling.

The SACs generated for each palm host species at both the genus and MOTU levels were consistent with previous analyses, with initial slopes suggesting host-related variations in fungal diversity. Fungal communities associated with *C. humilis*, *P. reclinata*, and *T. fortunei* appeared to be the most diverse, as indicated by the steeper gradient in their SACs, whereas *P. canariensis* and *P. dactylifera* harboured the least diverse fungal assemblages. Accordingly, species richness estimators also depicted *C. humilis* as hosting the most potentially diverse fungal communities, followed by *T. fortunei*, although all palm host species were found to have yet-undetected fungal diversity, implying that further sampling would likely reveal additional taxa. These results reinforce the presence of varying levels of fungal richness among different palm host species, highlighting differences in community composition that may be shaped by both host-specific and environmental factors. Interestingly, a low overlap of fungal taxa was observed among combinations of the six palm host species, ranging from 3% to 30%. This suggests that, regardless of their biogeographical origins or status as ornamentals, each palm host species supported distinct fungal communities. Such low overlap in fungal taxa has been consistently reported in palm fungal communities, e.g., refs. [29,268,362,363,364], and it is thought to be one of the factors contributing to the overall high fungal diversity associated with palms [45]. Moreover, some interesting hints regarding the unique association of PLSF with *C. humilis* have been unveiled for the first time, suggesting potential host-specific dynamics worth exploring further. These findings set the stage for understanding palm foliar lesions in Portugal as hyperdiverse microhabitats harbouring previously unknown mycota.

### 4.4. Foliar Lesions as Hyperdiverse Microhabitats with Co-Occurrence Patterns

The visual assessment of plant disease symptoms has long been a cornerstone practice in phytopathogen identification, relying on the observation of changes in the colour, morphology, and texture of plant tissues as the disease progresses. However, this traditional approach is inherently subjective and susceptible to variation and human error, especially given the context of climate change and shifting environmental conditions that may alter disease expression [24,365]. Moreover, it is well established that a single foliar lesion may harbour multiple potential phytopathogens, often with overlapping symptoms, thereby complicating the accurate identification of the primary causative agent through visual inspection alone, e.g., refs. [174,175,366]. In the present study, four distinct foliar lesion types—tip die-back (TDB), large leaf spots (LLSs), small leaf spots (SLSs), and pinpoints and punctuations (PPs)—were defined based on the overall morphological features of both the abaxial and adaxial leaf surfaces. Despite this categorisation, no major differences or discernible trends in fungal biodiversity were observed across the different foliar lesion types, which accounted for only 4.0% of the variation in the composition of fungal communities. The great majority of the samples were found to harbour more than one MOTU, although nearly all lesion types also included a few samples that appeared to be infected by a single fungal species, with a wide range of taxa implicated. These included some of the most commonly documented MOTUs, such as *Alternaria* MOTU 009 and *Neosetophoma* MOTU 185, both isolated singly from PP samples, and *Phoma* MOTU 243, isolated singly from a TDB sample. Only the relative frequencies of hyphomycetes and coelomycetes showed differences across the four foliar lesion types, with TDB and LLS enriched in coelomycetes, while SLS and PP exhibited a higher abundance of hyphomycetes, particularly the PP samples.

The size of the foliar lesions was one of the primary traits used to categorise foliar lesion types in this study, and it can explain part of the differences found between the foliar lesion types. While TDB and LLSs were typically large, often covering entire leaflets or leaf segments and thus unmeasurable, SLSs and PPs were smaller and usually allowed for specific and measurable definitions. The size of foliar lesions can be influenced by both the host plant and the associated fungal assemblage. For instance, leaf spots caused by *Septoria* spp. can vary substantially in size, from barely visible to covering up to one-third of the leaf area, depending on the host and fungal species involved [173]. The ability of certain fungal pathogens to infect larger areas of leaf tissue can affect the entire structure of the fungal community inhabiting the affected tissue, as larger lesions provide increased amounts of dead tissue and available nutrients, which can be readily utilised by other fungal colonisers. Indeed, nutrient acquisition is a key determinant of colonisation success during parasitic interactions [367,368,369]. This factor could explain why TDB lesions were found to harbour a greater diversity of MOTUs compared with the other foliar lesion types as well as a higher number of exclusive genera. In TDB lesions, necrosis consistently spread extensively throughout the leaves, and in several cases, the tissue could essentially be regarded as dead material. Therefore, it is not surprising that a greater MOTU richness was documented colonising some of the TDB lesions, which exhibited a considerable variability in the distribution of fungal occurrences per sample despite the lack of outliers in the boxplot analysis. This observation aligns with previous studies that have documented the succession of fungal communities in the phylloplane as leaves transition through senescence and eventually die. This succession often results in an increase in fungal diversity, as the senescence process grants phylloplane fungi easier access to the nutrients made available in the decomposing leaf biomass, e.g., refs. [370,371,372]. Additionally, senescent leaves may foster shifts in certain fungal taxa from an endophytic lifestyle to a saprotrophic one, further contributing to increased fungal diversity, e.g., refs. [373,374,375,376,377]. Interestingly, this transition from an endophytic to a saprotrophic lifestyle has also been suggested as a driver of the high fungal diversity observed in palm tissues [45], and it has been discussed occasionally in the present study. This dynamic process provides an explanation for the extraordinarily high fungal richness observed in highly necrotic TDB lesions, which could result from additional fungal species colonising an already established leaf-spotting community driven by the increased availability of nutrients. A similar pattern likely occurs in highly necrotic LLS samples, where brittle tissue was also observed to harbour atypically high taxon richness per sample, as evidenced by their identification as extraordinary outliers in the boxplot analysis. It is also worth noting that the presence of a larger quantity of diseased or necrotic tissue may facilitate the establishment of communities enriched in coelomycetous taxa, as an expanded leaf area provides more substrate to produce reproductive structures (pycnidia).

In several foliar lesions, a more or less conspicuous yellowish to light-green halo was frequently observed surrounding the damaged tissue. These halos represent areas of tissue chlorosis resulting from either chlorophyll destruction or a failure of chlorophyll biosynthesis. Various leaf-spotting fungi are known to induce such chlorotic halos through the production of phytotoxins [173]. For instance, leaf spots caused by *Alternaria* spp. are commonly associated with the development of chlorotic halos surrounding necrotic tissue as a result of toxin production [378,379,380]. Only the PP samples consistently exhibited chlorotic halos surrounding necrotic plant tissue, suggesting that these lesions are more often associated with fungal elements capable of producing toxins as part of their infection processes. Whether these toxins are produced by a single fungal species or by multiple fungal members of the community would require further verification through pathogenicity testing.

Although the number of MOTUs recorded per foliar lesion varied widely, with extreme cases of over 30 MOTUs observed, the typical number of MOTUs found in a single foliar lesion of palms is expected to be around 5. This finding suggests a diverse range of colonisation scenarios within foliar lesions, underscoring the need for a nuanced understanding of the interplay between primary pathogens and other fungal colonisers. The frequent assumption that any fungus isolated from symptomatic tissues is the primary pathogen without rigorous pathogenicity testing undermines the reliability of phytopathogen identification. This practice overlooks the potential presence of secondary invaders, including saprobes or opportunistic pathogens, that can colonise previously compromised tissues [381]. The pathogenicity of leaf-spotting fungi has long been recognised as often neglected or poorly tested, further casting doubt on the reliability of identification through conventional means. For instance, Hawthorne and Otto [382] conducted pathogenicity tests on fungi commonly isolated from necrotic lesions on the leaves, flowers, and fruit of kiwifruit (*Actinidia deliciosa*) and found that all the fungi were best classified as wound parasites incapable of infecting unwounded, healthy tissues. This finding may explain why multiple fungal species are often isolated from a single leaf spot. More recently, Yu et al. [383] reported that seven species could be considered primary pathogens of muskmelon (*Cucumis melo*) leaf spot in Eastern Shandong, China, several of which had not previously been identified.

Given the challenges in reliably identifying phytopathogens of foliar diseases, a community-level analysis of foliar lesions offers a more nuanced and holistic approach to understanding plant diseases. Only 8% of the samples studied here appeared to be colonised by a single fungus, while over 65% harboured between 2 and 10 distinct MOTUs, underscoring the complexity and inherent hyperdiversity of foliar lesions. Studying the entire fungal assemblage associated with foliar lesions provides deeper insights into the ecological dynamics at play in these hyperdiverse microhabitats, thereby allowing for a more comprehensive identification of potential pathogens. These findings highlight the limitations of relying solely on traditional, single-pathogen-focused assessments for accurate pathogen identification, emphasizing the importance of integrative approaches that can address the coexistence of multiple potential pathogens within a single lesion.

#### Co-Occurrence in Palm Leaf Spotting Fungal Communities

Interactions among organisms and environmental influences on coexistence within biological communities are often explored in ecology through the analysis of species CO. Co-occurrence patterns serve as indicators of the ecological processes that drive coexistence and maintain diversity within communities [384,385]. Co-occurrence refers to the frequency with which two or more taxa are found together across different samples, and it may be random, positive, or negative. Random CO suggests that the presence of one taxon does not influence the distribution of another, whereas positive and negative COs suggest that taxa are found together more often or less often, respectively, than expected by chance. Typically, positive COs suggest mutual affinity or shared habitat requirements, while negative COs imply avoidance or competition [114,386,387,388]. Thus, analysing CO patterns can provide insights into the underlying ecological processes and interactions that shape community composition. The concept of fungal co-occurrence has recently been explored in communities of palmicolous saprophytes, particularly in studies on *Nypa fruticans* in Brunei [389]. In the present study, the genera and MOTUs in foliar lesions predominantly exhibited random associations, indicating that fungal community assembly is largely neutral, with stochastic processes playing a prominent role. This observation is supported by the high strength-of-fit (SOF) values obtained from the neutral model, although the best overall fit was provided by a logseries distribution. These results are consistent with the existing literature that suggests several factors governing the structuring of biological communities, with abundant and rare taxa often presenting as sub-communities being driven by different ecological forces, e.g., refs. [390,391,392,393]. For example, Jiao and Lu [392] found that neutral processes played a more significant role in shaping the abundant fungal sub-community compared with the rare sub-community in agro-ecosystems of Eastern China. More recently, Unterseher et al. [390] reported that the analysis of abundant and rare MOTUs in plant-associated fungal communities of the phyllosphere revealed two distinct species abundance distributions, highlighting the ecological significance of the infrequent fungal component that followed a logseries distribution. This suggests that the structuring forces at play within fungal communities may vary depending on the relative abundance of taxa, with abundant species more likely to follow neutral dynamics while rare taxa could be shaped by other, perhaps more deterministic, ecological processes. However, it is important to note that the present study is preliminary, and the data gathered here do not yet allow for any definitive conclusions about the distinct dynamics between rare and abundant mycota. Further studies are needed to explore these relationships in depth, particularly by employing next-generation sequencing (NGS) methodologies such as DNA metabarcoding, which could help to reveal the undetected mycota that may have been overlooked. Indeed, current analyses suggest that at least approximately 30% of the potential PLSF diversity remains uncollected, underscoring the need for a more comprehensive survey of these fungal communities. Future efforts could provide a more nuanced understanding of how abundant and rare taxa are structured and maintained within these hyperdiverse foliar microhabitats.

Despite the predominance of random associations, a notable proportion of the fungal community exhibited positive CO patterns. Approximately a quarter of the genera and 15% of the MOTUs showed significant positive pairwise associations, indicating a trend towards coexistence among these taxa. Several mechanisms could explain these positive associations: (a) mutualistic or complementary relationships, where co-occurring fungi complement each other in nutrient acquisition; (b) shared environmental preferences or tolerance, leading to frequent cohabitation due to comparable habitat requirements; and (c) facilitation through secondary colonisation, where one taxon alters or degrades plant tissue, thereby creating conditions conducive for subsequent colonisers to establish. Notably, many of the taxa documented in this study are likely opportunistic fungi, which is reflected in their positive pairwise associations, often involving the most prevalent taxa. This suggests a fungal assemblage shaped by shared ecological opportunities rather than by direct competition. Such ecological dynamics have previously been reported in controlled conditions among communities of fungal saprophytes, where dominant fungi, particularly primary decayers, exert an influence on the growth and diversity of co-occurring fungi. For instance, Pouska et al. [394] observed that the presence of *Fomitopsis pinicola* influenced species richness and the composition of fungal communities in *Picea abies* logs from an old-growth mountain spruce forest in the Bohemian Forest, Czech Republic. Similarly, Maria and Sridhar [395] noted that the dominant fungus *Lignincola laevis* shaped associated fungal communities colonising *Avicennia officinalis* in a southwestern Indian mangrove. More recently, Sarma and Hyde [389] reported that *Linocarpon bipolaris* and *L. appendiculatum* played a similar role in influencing fungal colonisers of the palm *Nypa fruticans* in the Tutong River, Brunei. Several other studies have shown the importance of fungal interactions in shaping microbial communities. Abrego et al. [396] reported that positive co-occurrences play an important role in root-associated fungal communities. Banik et al. [397] showed that intraspecific interactions among five species of wood-decay fungi can alter decay rates and the dynamics of interspecific interactions. Thus, it can be hypothesised that dominant colonisers within palm foliar lesions exhibit a certain degree of accommodation for other fungi, suggesting a mutualistic or synergistic relationship that facilitates coexistence within these hyperdiverse communities.

Most positive associations involved representatives of the coelomycete assemblage, further hinting at the notorious role of this group in palm foliar diseases. The top five genera involved in positive COs were *Alternaria*, *Didymella*, *Fusarium*, *Neodidymelliopsis,* and *Neosetophoma*, all of which, except *Didymella*, have previously been recognised as core components of PLSF communities in Portugal. At the MOTU level, those of *Alternaria*, *Neosetophoma*, and *Phoma* were the most frequently involved in positive COs. Interestingly, despite being the two most prevalent taxa in PLSF, *Alternaria* and *Neosetophoma* predominantly exhibited random rather than positive associations. The intricate relationship between the MOTUs of these two genera, as previously indicated by the PCA results, hints at potential host-related patterns in their interactions. Similarly, *Cladosporium,* despite being the fourth most prevalent genus, also exhibited mostly random associations. These patterns may reflect differences in the functional roles of the taxa, with some more actively engaged in pathogenicity, while others are primarily involved in saprophytism. Likewise, the observation of significant positive associations between some of the infrequent and rare taxa with the most prevalent mycota suggests potential ecological dependencies. As previously discussed, it is plausible that frequent and very frequent mycota facilitate the establishment of rare taxa in foliar lesions, or vice versa, through mechanisms such as habitat modification or secondary colonization, which is widely documented in the existing literature, e.g., refs. [176,236,237]. Notably, rare genera like *Hendersonia* and *Phaeosphaeria* exhibited a higher proportion of significant positive associations compared with random ones, hinting at non-random ecological interactions. This trend was also seen for *Didymella*, *Fusarium*, and *Neodidymelliopsis*, as well as for *Phoma* MOTU 246, with a few other MOTUs exhibiting a high number of positive associations, providing insights into their possible roles in the broader community. The MOTU-level analysis also highlighted two significant negative associations, namely, *Botrytis* MOTU 039 and *Alternaria* MOTU 009 and *Neosetophoma* MOTU 188 and *Stemphylium* MOTU 303, which may indicate competitive interactions or exclusion relationships between these MOTUs. Such negative associations hint at possible antagonistic dynamics within the fungal communities, where resource competition or habitat incompatibility could limit the coexistence of these taxa. It is noteworthy that both *Botrytis* MOTU 039 and *Alternaria* MOTU 009 were isolated singly, which highlights their potential antagonistic behaviour.

The presence of both positive and negative COs among various MOTUs emphasises the complex and nuanced ecological interactions within foliar lesions. However, the predominance of positive associations suggests facilitative processes, where some fungi may aid the colonisation of others, with limited evidence of competition. Notably, positive associations between uncommon and common mycota may indicate that primary colonisers alter the habitat to support later arrivals. Further research, particularly pathogenicity testing, is needed to clarify the specific roles of these taxa and determine their actual influence on plant health, especially given the growing evidence of synergistic interactions between microbial pathogens in plant disease complexes [398,399]. Pathogenicity tests will help distinguish between true pathogens, which actively contribute to disease development, and secondary colonisers, which take advantage of pre-existing tissue damage. Understanding these interactions will not only enhance the ecological knowledge of foliar fungal communities but also help refine disease management strategies by focusing on key pathogens rather than harmless cohabitants. Such strategies are crucial for developing accurate diagnostics and effective management solutions for plant diseases in these hyperdiverse microhabitats.

## 5. Conclusions

Understanding plant–pathogen associations is essential for maintaining the health, aesthetic value, and sustainability of ornamental plants as well as for the effective management of agricultural crops and forest ecosystems. This understanding is becoming increasingly important in the context of global trade and the potential transboundary movement of plant pathogens. Although no major fungal outbreaks have yet been identified in *Arecaceae* hosts, the recent impact of the red palm weevil (*Rhynchophorus ferrugineus*), which has decimated *Phoenix canariensis* populations across the Mediterranean basin, serves to illustrate the potentially devastating consequences that pests and diseases can have when introduced to hosts growing outside their native range. The health of ornamental palms can be compromised by various fungal pathogens, often manifesting as foliar diseases that detract from their visual appeal and vitality. Despite their popularity as ornamental plants, there is an almost complete lack of knowledge about the fungi associated with palms in Portugal, underscoring the need for targeted research to assess fungal diversity and potential pathogenic threats. In this regard, the present study provides a foundational understanding of the diversity and ecology of potential phytopathogenic fungi associated with ornamental palm trees in Portugal, elucidating the influence of a complex interplay of climatic, host, and ecological factors on communities of palm leaf spotting fungi (PLSF).

The presence of dominant genera such as *Neosetophoma*, *Alternaria*, *Phoma*, and *Cladosporium* alongside a wide diversity of rare mycota suggests that communities of PLSF are shaped by a combination of common, potentially pathogenic species and rare taxa that may act as reservoirs of pathogenic potential. While the role of the dominant taxa in establishing and expanding foliar lesions was evident, the ecological significance of the rare mycota requires further investigation. These taxa could become active disease agents or facilitators under specific environmental conditions or biotic stressors, underscoring the need for more comprehensive research to elucidate their functional roles within PLSF communities. Moreover, the pathogenic potential of *Neosetophoma* in palms remains largely unexplored and warrants further studies. Although identified as the most prevalent genus in the palm foliar lesions, its association with palm diseases has not been documented prior to this study. While PLSF communities were predominantly characterised by random associations, the emergence of significant positive co-occurrence (CO) patterns, particularly between prevalent and uncommon taxa, highlights the complex and interactive nature of these assemblages. These findings indicate that palm foliar lesions function as dynamic and hyperdiverse microhabitats where diverse fungal taxa coexist and interact, potentially influencing the progression and severity of host diseases.

The dominance of coelomycetes in the foliar lesions of the palms sampled in this study underscores their potential role as key components of PLSF communities in Portugal. Furthermore, this dominance suggests that they may represent an emerging assemblage of typical temperate palm fungi (TePF), significantly contributing to the diversity of palm fungal assemblages in temperate environments. These results indicate that climatic factors influence the diversity of PLSF, as evidenced by the divergence between TePF and tropical palm fungi (TrPF) as well as several commonly reported PLSF. Despite the presence of some typical TrPF, such as *Fasciatispora*, the overall composition of the PLSF reflected a distinct assemblage consistent with previously documented trends in temperate regions, where tropical palm mycota are typically replaced by ubiquitous, plurivorous taxa. If the host substrate alone were the primary determinant, ascomycetes frequently associated with tropical palms would also be expected on the same palm hosts in temperate regions. However, the observed differences, both here and in previous studies, suggest that these patterns are primarily driven by climatic constraints. Indeed, almost no typical TrPF were recorded in this study, with ubiquitous fungi assuming greater prominence. This further supports the hypothesis that climate plays a pivotal role in structuring palm-associated mycota, with temperate environments favouring the proliferation of coelomycetous taxa. It is important to note that no wild representatives of palms were included in this study, and the fungal assemblages recorded may reflect this limitation. While climate is evidently an influence, a survey of the mycota associated with a warm temperate palm such as *Chamaerops humilis*, which is endemic to the Algarve region of Portugal, where wild populations are available, would provide a more robust baseline for understanding these patterns. Moreover, it could also clarify whether TePF represent a typical assemblage of phytopathogens or if they include representatives of palmicolous saprophytes and endophytes, thereby offering deeper insights into the ecological dynamics of palm-associated fungi in temperate environments.

The influence of climate was further evidenced by the clustering patterns among the fungal assemblages of different host species, which revealed a clear separation between temperate and tropical palms. The fungal communities of temperate palms such as *C. humilis* and *Trachycarpus fortunei* exhibited a distinct enrichment in coelomycetous taxa, including *Keissleriella*, *Neodidymelliopsis*, *Neosetophoma*, and *Sclerostagonospora*, which are uncommon in TrPF assemblages. This pattern extended to the tropical palm *P. reclinata*, whose mycota closely resembled that of temperate palms. This similarity is likely facilitated by the close spatial proximity of the palms and other plant species in the mixed planting arrangements sampled. These findings underscore the influence of environmental and spatial factors on fungal community composition, particularly the role of spore exchange in densely planted urban landscapes. In contrast, tropical and subtropical palms, including *Chrysalidocarpus lutescens*, *P. canariensis,* and *P. dactylifera*, exhibited either reduced fungal diversity or an impoverishment in coelomycetes, with a higher prevalence of widespread, plurivorous taxa. This disparity suggests that certain fungal assemblages may be poorly adapted to temperate conditions or influenced by a long history of cultivation, resulting in a diminished community composition in non-native palms. The limited research on fungal assemblages associated with some palm species highlights significant knowledge gaps that need to be addressed to fully understand palm–fungal interactions, particularly in non-native environments. The paucity of previous research on fungal diversity in ornamental palms in temperate regions complicates direct comparisons with their tropical counterparts. However, it also provides an opportunity for future research to explore the unique ecological dynamics that arise when tropical plants are introduced into temperate landscapes. Such studies will be essential for uncovering how environmental factors, planting practices, and species interactions shape fungal communities and influence their ecological roles.

Foliar lesions are important indicators of underlying fungal activity and plant health. Understanding the fungal communities associated with these diseases is critical, as they can encompass a range of organisms, ranging from harmless or beneficial symbionts that promote plant growth to harmful pathogens that threaten palm integrity. This dynamic highlights the need for further research into the interactions among taxa within PLSF communities, as understanding these relationships can provide insights into the mechanisms driving disease outbreaks in ornamental palms. Investigating the conditions that trigger potential latent pathogens to transition into active disease agents will enhance the understanding of foliar disease ecology. Moreover, the hyperdiverse microhabitats of foliar lesions underscore the limitations of traditional assessment methods for accurate pathogen identification. This emphasises the need for integrative approaches that address the coexistence of multiple potential pathogens within a single lesion. By exploring these interactions, researchers can gain valuable insights to inform the development of targeted control measures and effective management strategies, ultimately preventing disease outbreaks and supporting the health of ornamental plantings.

This study contributes to the growing body of literature emphasising the influence of climate, host characteristics, and local environmental factors on plant-associated microbial communities. The introduction of palms into non-native, temperate environments not only shapes the observed fungal diversity observed but may also carry long-term ecological implications by altering plant–pathogen interactions. Future research should aim to further elucidate the dynamics of fungal community assembly in ornamental palms, with a particular focus on the role of coelomycetes and rare taxa as well as the implications for palm health and disease management in urban settings. These results have sizable implications for biodiversity assessments, ecological research, and phytosanitary management of ornamental palms. Future work should prioritise the integration of phylogenetic analyses and DNA metabarcoding to achieve precise species-level identifications and expand the geographic and taxonomic scope of studies on palm-associated fungi. Such efforts are essential for enhancing the understanding of fungal diversity and ecological roles, particularly in light of increasing biosecurity concerns driven by the global trade of ornamental plants.

## Figures and Tables

**Figure 1 jof-11-00043-f001:**
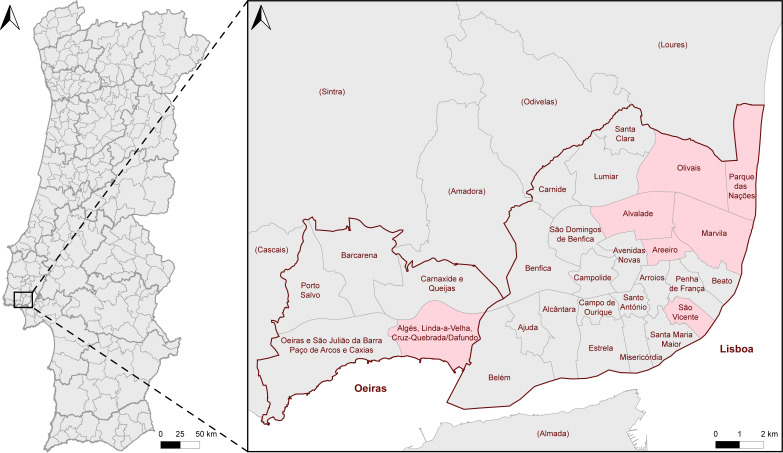
Map of sampling sites. The civil parishes of Lisbon and Oeiras that correspond to sampling sites are highlighted in pink. Source: original map created in QGIS version 3.38.1 “Grenoble” [80].

**Figure 2 jof-11-00043-f002:**
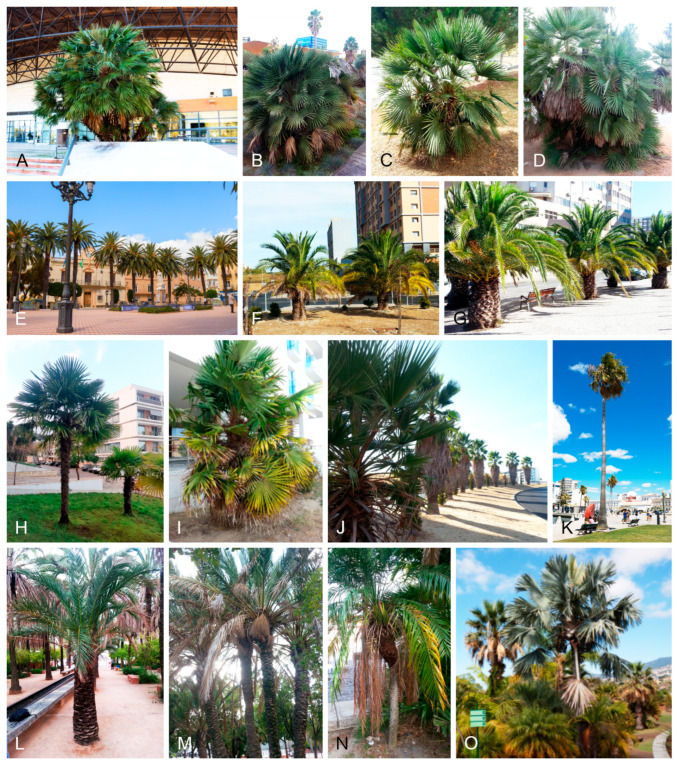
Illustrative photographs of ornamental palms in Portuguese cities. (**A**–**D**) *Chamaerops humilis*. (**E**–**G**) *Phoenix canariensis*. (**H**,**I**) *Trachycarpus fortunei*. (**J**,**K**) *Washingtonia filifera*. (**L**) *Phoenix dactylifera*. (**M**) *Phoenix reclinata*. (**N**) *Phoenix roebelenii*. (**O**) *Washingtonia filifera* and *P. roebelenii*.

**Figure 3 jof-11-00043-f003:**
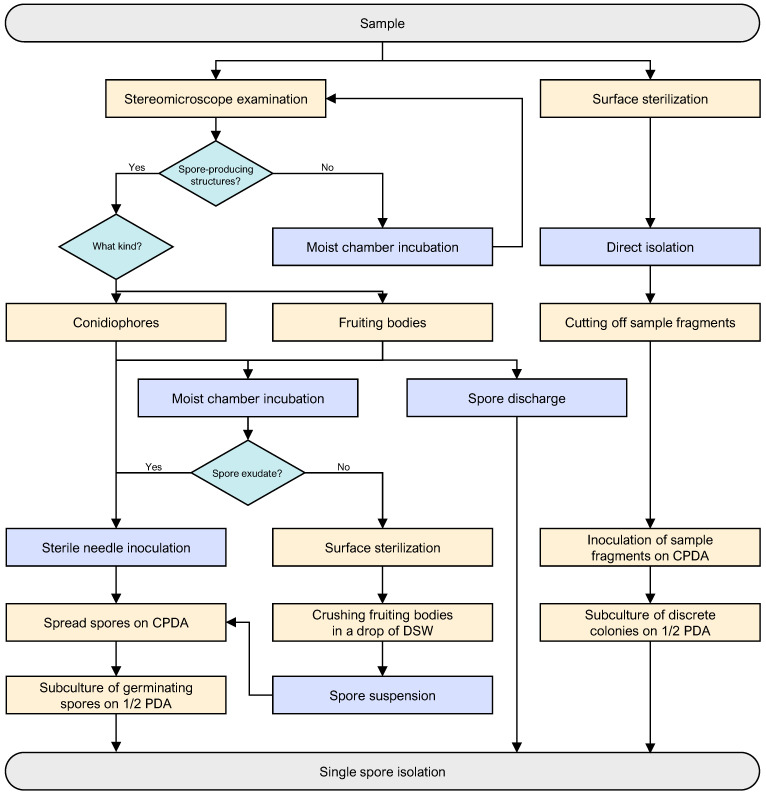
Fungal isolation flowchart. Coloured shapes represent different stages in the isolation process followed to establish the fungal communities of each sample. Grey rounded rectangles indicate either the beginning or the termination of the process. Green lozenges indicate decision points where the process is split into decisive options. Blue rectangles represent the methods used to isolate the fungal communities. Yellow rectangles represent methodological stages before an isolation method or the end of the process. C: chloramphenicol; DSW: distilled sterile water; PDA: potato dextrose agar.

**Figure 4 jof-11-00043-f004:**
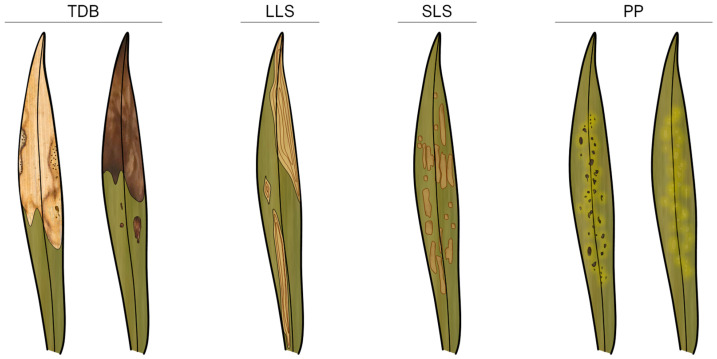
Schematic illustration of the general morphology of foliar lesion types. Illustrations not to scale. Illustrations by first author. TDB, tip die-back; LLS, large leaf spot; SLS, small leaf spot; PP, pinpoints and punctuations.

**Figure 5 jof-11-00043-f005:**
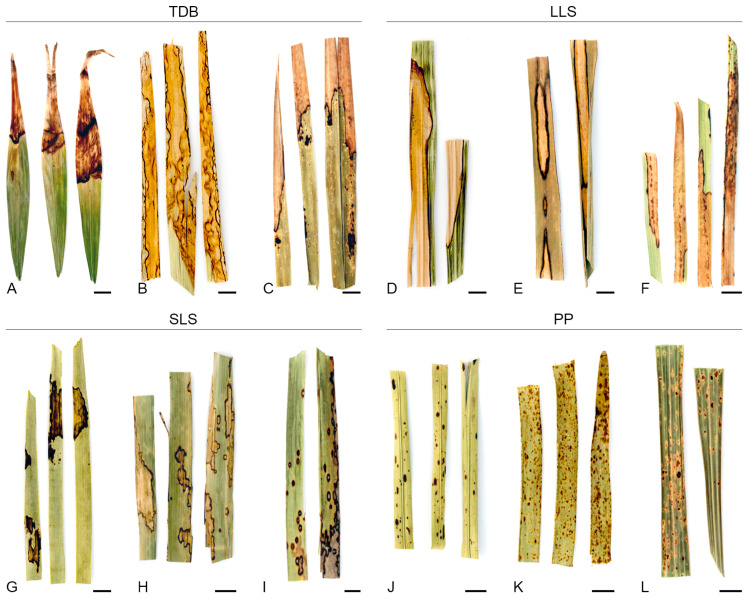
Representative samples of the foliar lesion types. (**A**) Tip die-back (TDB) foliar lesions on *Chamaedorea elegans* leaves (sample HDP 006). (**B**) TDB foliar lesions on *Phoenix reclinata* leaves (sample HDP 041). (**C**) TDB foliar lesions on *P. canariensis* leaves (sample HDP 018). (**D**) Large leaf spots (LLSs) on *Chrysalidocarpus lutescens* leaves (sample HDP 009). (**E**) LLSs on *Trachycarpus fortunei* leaves (sample HDP 042). (**F**) LLSs on *P. dactylifera* leaves (sample HDP 046/02). (**G**) Small leaf spots (SLSs) on *P. canariensis* leaves (sample HDP 033/02). (**H**) SLSs on *P. reclinata* leaves (sample HDP 050). (**I**) SLSs on *Chamaerops humilis* leaves (sample HDP 003). (**J**) Pinpoints and punctuations (PPs) on *P. canariensis* leaves (sample HDP 004/01). (**K**) PPs on *C. humilis* leaves (sample HDP 049). (**L**) PPs on *C. lutescens* leaves (sample HDP 007). Scale bars: 1 cm.

**Figure 6 jof-11-00043-f006:**
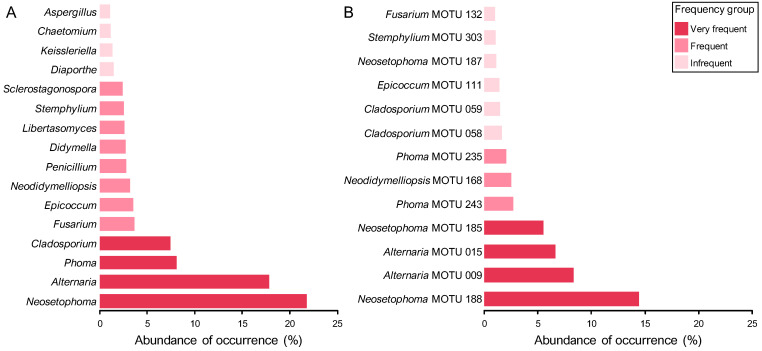
Comparison between the most commonly documented genera and molecular operational taxonomic units (MOTUs) by frequency groups. (**A**) Percentage abundance of occurrence (AO) of the most commonly documented genera. (**B**) Percentage AO of the most commonly documented MOTUs. Taxa with an AO ≤ 1% (taxa regarded as rare) were excluded. Colours represent frequency groups and are referred to in the chart legend.

**Figure 7 jof-11-00043-f007:**
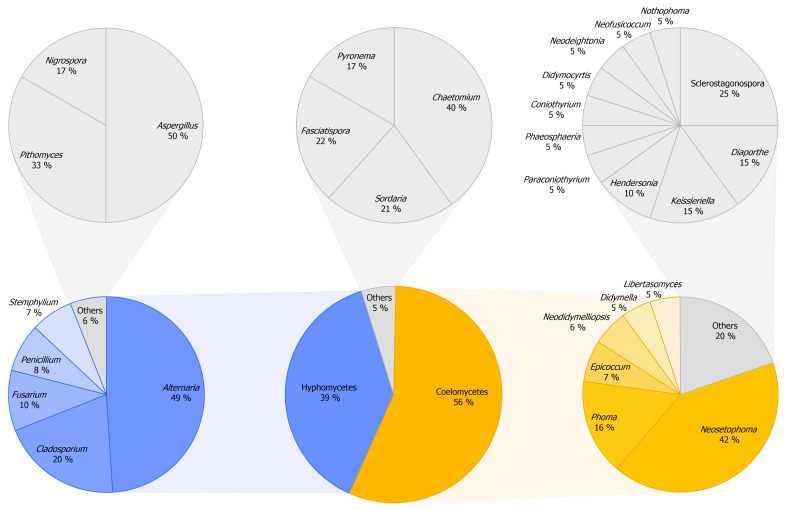
Distribution of fungal records by assemblage type and genus. The percentage abundance of occurrence (AO) of the main fungal genera isolated from foliar lesions is presented according to the assemblages of coelomycetes and hyphomycetes. Genera with an AO < 0.45% (fewer than 10 fungal records) were excluded.

**Figure 8 jof-11-00043-f008:**
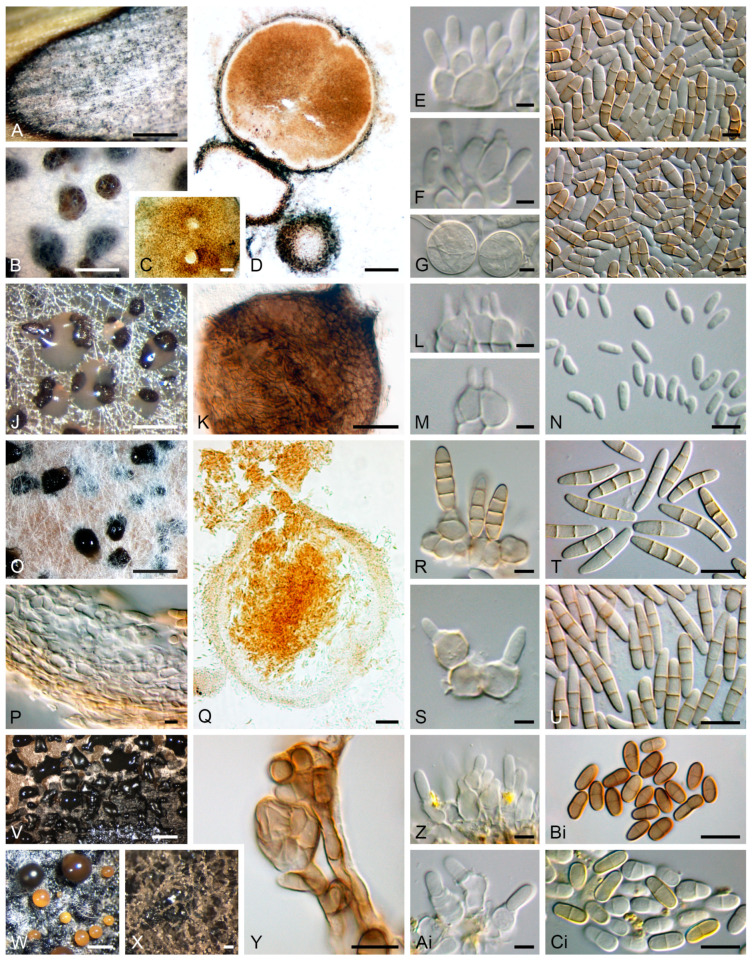
Overview of the morphological diversity of representatives of palm leaf spotting fungi (PLSF). (**A**–**N_ii_**) Coelomycetes. (**O_ii_**–**X_iii_**) Hyphomycetes. (**A**–**I**) *Neosetophoma*. (**L**–**N**) *Phoma*. (**O**–**U**) *Sclerostagonospora*. (**V**–**C_i_**) *Coniothyrium*. (**D_i_**–**G_i_**) *Libertasomyces*. (**I_i_**–**L_i_**) *Diaporthe*. (**M_i_**–**O_i_**) *Plenodomus*. (**P_i_**–**S_i_**) *Epicoccum*. (**T_i_**–**W_i_**) *Phyllosticta*. (**Xi**–**Zi**) *Xenocylindrosporium*. (**A_ii_**–**C_ii_**) *Neodeightonia*. (**D_ii_**–**N_ii_**) *Colletotrichum*. (**O_ii_**–**W_ii_**) *Alternaria*. (**X_ii_**–**E_iii_**) *Cladosporium*. (**F_iii_**–**H_iii_**) *Penicillium*. (**I_iii_**–**K_iii_**) *Aspergillus*. (**L_iii_**–**N_iii_**) *Scopulariopsis*. (**O_iii_**–**Q_iii_**) *Pithomyces*. (**R_iii_**,**S_iii_**) *Arthrinium*. (**T_iii_**–**X_iii_**) *Fusarium*. (**A**,**D_ii_**) Conidiomata formed on host tissue. (**B**,**J**,**O**,**V**–**X**,**D_i_**,**I_i_**,**T_i_**,**X_i_**,**E_ii_**,**T_iii_**) Conidiomata formed on WA. (**C**,**E_i_**) Ostioles. (**D**,**Q**,**J_i_**,**M_i_**,**U_i_**) Vertical section of conidiomata. (**E**,**F**,**L**,**M**,**R**,**S**,**Z**,**A_i_**,**F_i_**,**G_i_**,**K_i_**,**N_i_**,**V_i_**,**Y_i_**,**B_ii_**,**H_ii_**) Conidiogenous cells. (**H**,**I**,**N**,**T**,**U**,**B_i_**,**C_i_**,**H_i_**,**L_i_**,**R_i_**,**S_i_**,**W_i_**,**Z_i_**,**C_ii_**,**I_ii_**,**T_ii_**,**V_ii_**,**G_iii_**,**J_iii_**,**Q_iii_**,**W_iii_**,**X_iii_**) Conidia. (**G**,**Y**) Chlamydospores. (**K**) Optical section of conidioma. (**P**) Section of conidioma wall. (**P_i_**,**Q_i_**,**O_ii_–S_ii_**,**U_ii_**,**W_ii_**,**X_ii_–B_iii_**,**D_iii_**,**E_iii_**,**F_iii_**,**H_iii_**,**I_iii_**,**K_iii_**,**L_iii_–N_iii_**,**O_iii_**,**P_iii_**,**R_iii_**,**S_iii_**,**V_iii_**) Conidiophores and conidiogenous cells. (**A_ii_**) Conidiomata formed on palm leaf piece. (**F_ii_**) Acervulus with setae. (**G_ii_**) Seta with a conidiogenous tip. (**J_ii_**) Appressorium. (**K_ii_**) Vertical section of ascoma growing on host tissue. (**L_ii_**,**M_ii_**) Ascus. (**N_ii_**) Ascospores. (**C_iii_**) Mycelial coil. (**U_iii_**) Optical section of sporodochium. Scale bars: (**D_i_**,**I_i_**,**X_i_**,**A_ii_**) = 1 mm, (**D_ii_**,**E_ii_**,**T_iii_**) = 0.5 mm, (**A**,**B**,**J**,**O**,**V**–**X**,**T_i_**) = 0.25 mm, (**J_i_**,**M_i_**,**U_i_**,**F_ii_**,**K_ii_**) = 15 µm, (**C**,**K**,**Q**,**Y**,**B_ii_**,**O_ii_**,**R_ii_**,**U_ii_**,**W_ii_**,**E_ii_**,**U_iii_**,**W_iii_**) = 10 µm, (**D**,**P**,**G**,**H**,**I**,**N**,**T**,**U**,**B_i_**,**C_i_**,**K_i_**,**H_i_**,**L_i_**,**O_i_**,**P_i_**–**S_i_**,**V_i_**,**W_i_**,**Y_i_**,**Z_i_**,**C_ii_**,**G_ii_**,**H_ii_**–**I_ii_**,**L_ii_**–**N_ii_**,**P_ii_**,**Q_ii_**,**S_ii_**,**T_ii_**,**V_ii_**,**X_ii_**–**D_iii_**,**F_iii_**–**S_iii_**,**V_iii_**,**X_iii_**) = 5 µm, (**E**,**F**,**L**,**M**,**R**,**S**,**Z**,**A_i_**,**F_i_**,**G_i_**,**N_i_**) = 2.5 µm.

**Figure 9 jof-11-00043-f009:**
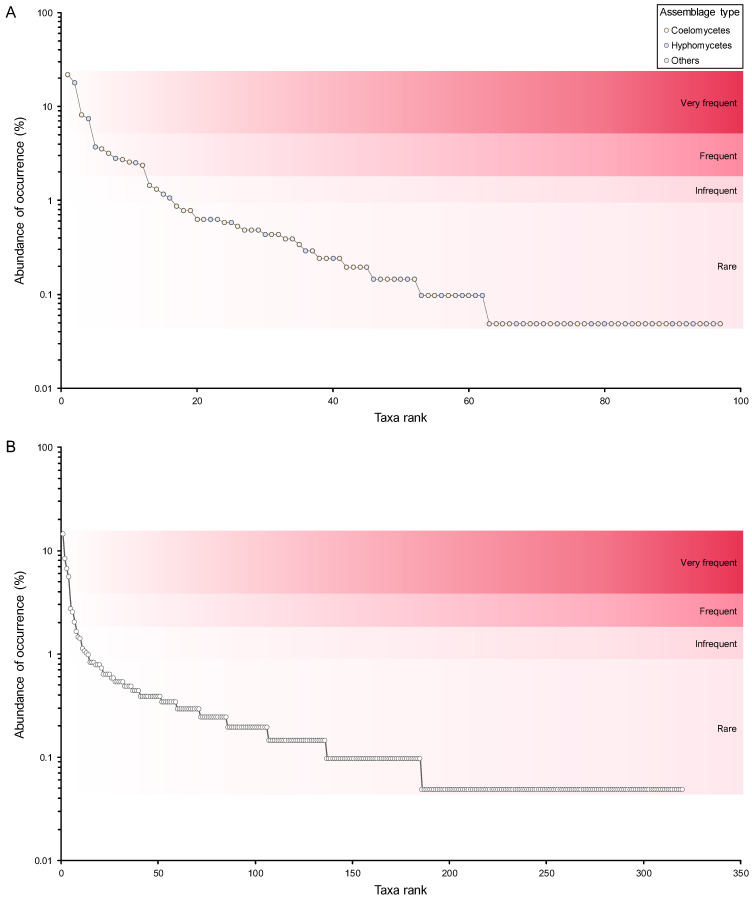
Abundance distributions for assemblages of palm leaf spotting fungi (PLSF) by taxonomic rank. (**A**) Genus abundance distribution for assemblages of PLSF. (**B**) Molecular operational taxonomic unit (MOTU) abundance distribution for assemblages of PLSF. The *y*-axis (on a logarithmic scale, base 10) represents the abundance of occurrence of each taxon, which is the relative importance of each genus or MOTU in the fungal assemblage as a percentage. The sum of all relative importance values is equal to 100%. Each taxon is ranked from the most to least abundant along the *x*-axis. Taxa are delimited with coloured blocks according to the frequency groups referred to on the right. Points in panel (**A**) are coloured by assemblage type and referred to in the chart legend. This information has been omitted in panel (**B**), as most points are barely visible due to the high number of taxa in each percentage abundance level.

**Figure 10 jof-11-00043-f010:**
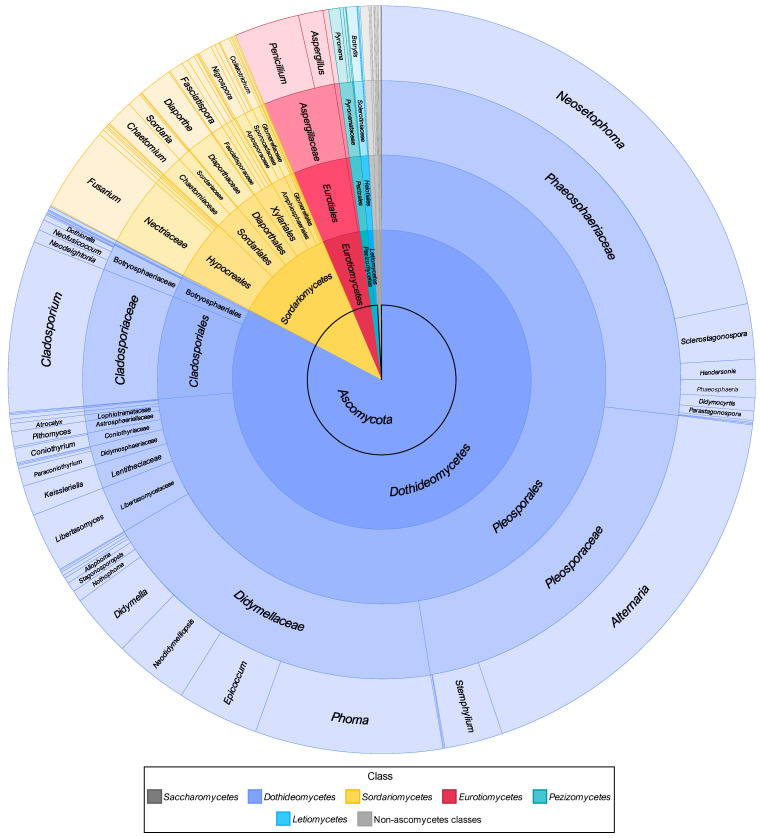
Nested pie chart of ranked fungal abundance, illustrating the taxonomic structure of palm leaf spotting fungi (PLSF). Each ring in the chart represents a taxonomic rank from the inner to outer ring: phylum, class, order, family, and genus. The size of the wedges within each ring indicates the relative abundance of the taxa within that rank. Within each divisible taxonomic rank, the taxa are ordered clockwise from the most to least abundant but conditioned by the previous rank. For reasons of legibility, taxa with a very low relative abundance have been omitted. Colours are according to class and referred to in the chart legend.

**Figure 11 jof-11-00043-f011:**
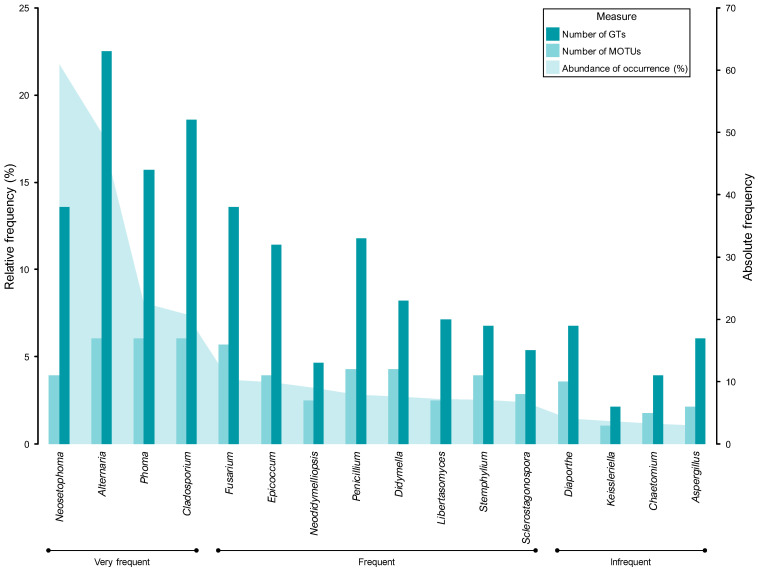
Comparison between the percentage abundance of occurrence (AO), the number of molecular operational taxonomic units (MOTUs), and the genetic diversity of the most commonly documented genera by frequency groups. Genetic diversity is presented in terms of genomic types (GTs). Taxa with an AO ≤ 1% (taxa regarded as rare) are excluded. Colours represent population status and diversity measures and are referred to in the chart legend.

**Figure 12 jof-11-00043-f012:**
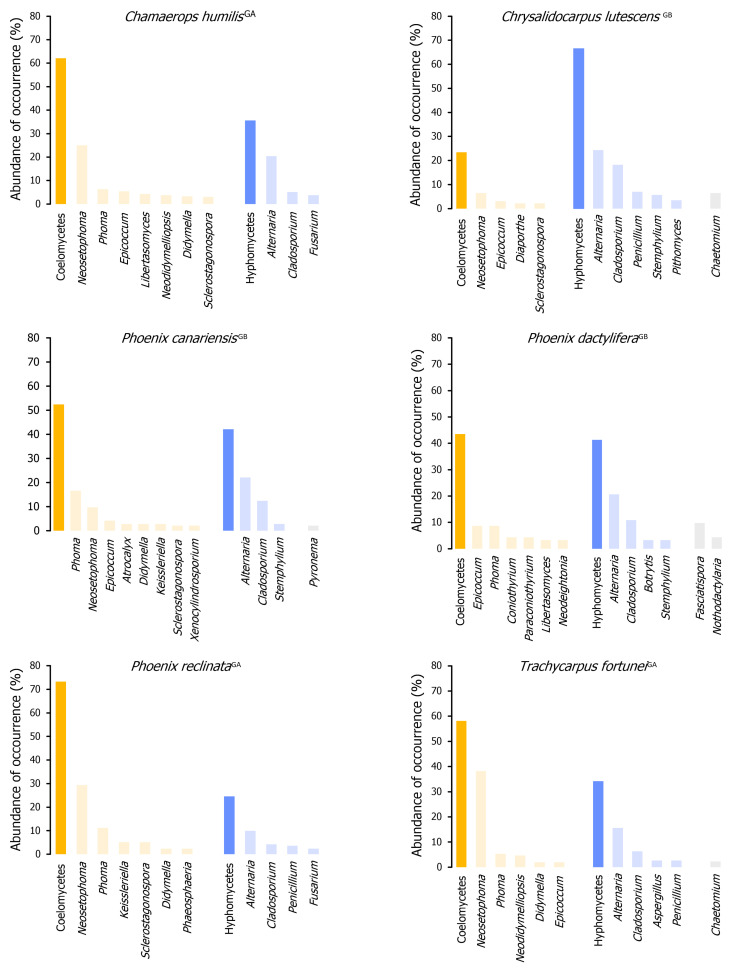
Top ten genera by assemblage type and host species. Each panel depicts the percentage abundance of occurrence (AO) of hyphomycete and coelomycete assemblages and the ten most documented genera found associated with foliar lesions by host species. Twelve genera are shown in *Phoenix canariensis* and *P. dactylifera*, as the tenth, eleventh, and twelfth most documented genera exhibited the same AO. Colours are according to assemblage types and are referred to in the first bar of each colour. Grey bars represent genera not counted as either coelomycetes or hyphomycetes (others). The groups defined according to trends in fungal biodiversity are identified by superscript abbreviations as group A (^GA^) and group B (^GB^) in the respective palm host species.

**Figure 13 jof-11-00043-f013:**
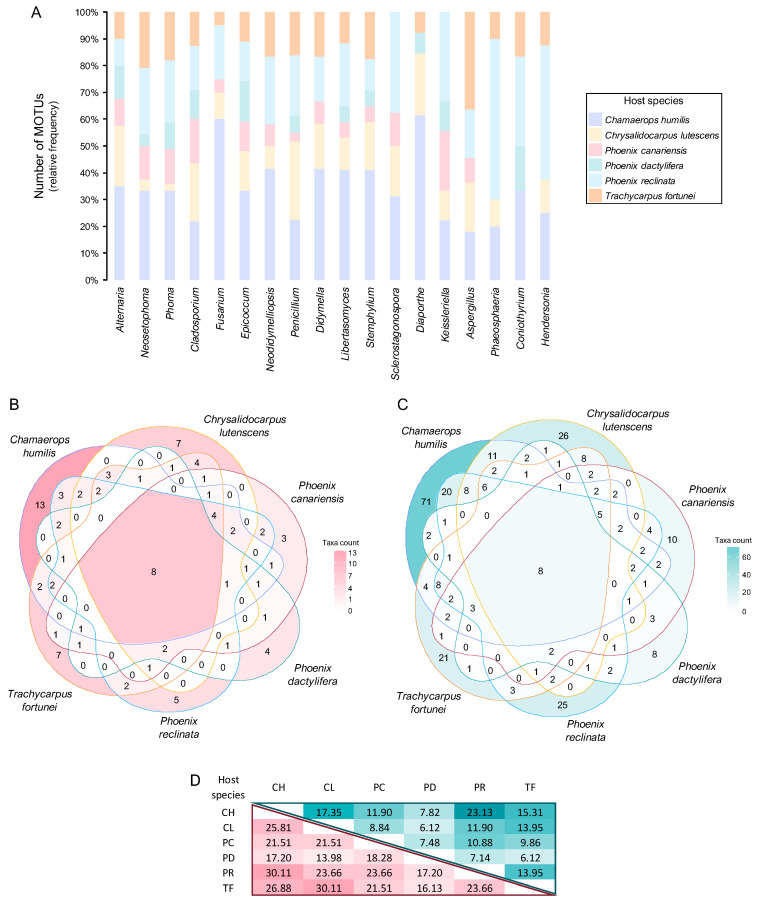
General biodiversity patterns of taxon richness and overlap by host species. (**A**) Genera recorded on at least four host species and corresponding number of MOTUs in relative frequency. Genera with an abundance of occurrence (AO) ≤ 0.45% were excluded. Colours are according to host species and are referred to in the chart legend. (**B**,**C**) Venn diagrams of unique and overlapping fungal taxa by host species at the genus level (N = 93) (panel (**B**)) and at the molecular operational taxonomic unit (MOTU) level (N = 294) (panel (**C**)). Colours of shapes are according to taxa count and are referred to in the scale legend. Colours of lines are according to host species. (**D**) Percentage of overlapping fungal taxa by two palm host species at the genus (lower triangle of the correlation table, N = 93) and MOTU (upper triangle, N = 294) levels. Colours of shapes are according to taxa count (darker shades signify higher proportions), and their relative proportions are displayed in each cell of the correlation table. CH, *Chamaerops humilis*; CL, *Chrysalidocarpus lutescens*; PC, *Phoenix canariensis*; PD, *Phoenix dactylifera*; PR, *Phoenix reclinata*; TF, *Trachycarpus fortunei*.

**Figure 14 jof-11-00043-f014:**
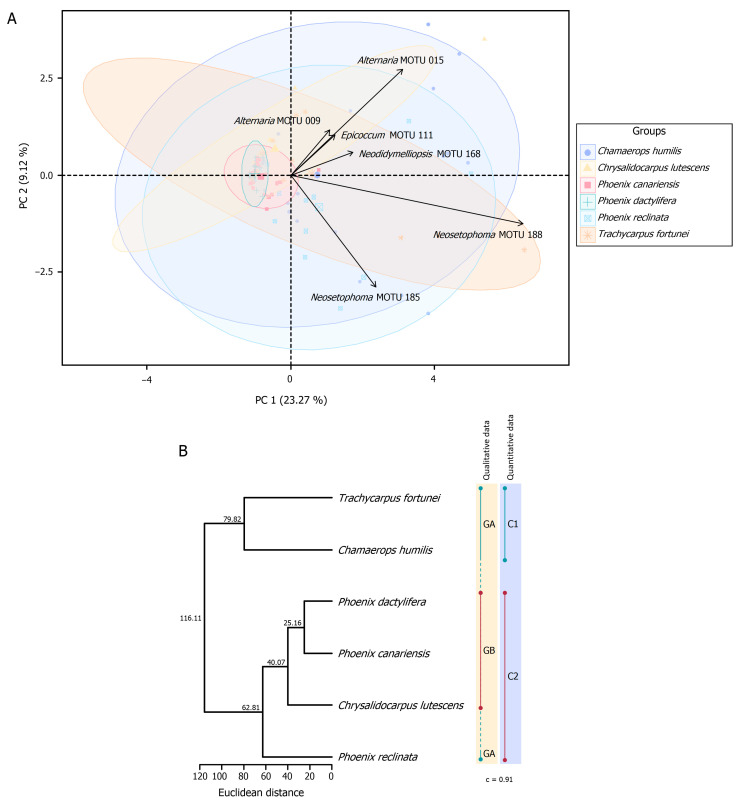
Host-associated clustering of communities of palm leaf spotting fungi (PLSF). (**A**) Principal component analysis (PCA)-biplot of communities of PLSF based on the distribution of samples from six host species. PCA was based on square-root-transformed abundance of occurrence (AO) of molecular operational taxonomic units (MOTUs) using samples with ≥5 fungal records. Explained variance by each principal component (PC) is given in parentheses. Samples are represented with coloured symbols according to host species and are referred to in the chart legend. Grouping of samples by host species is indicated by centroids (larger symbols) and ellipses of 95% confidence intervals (CIs) representing the true population mean of the bivariate distribution for groups. Vector arrows indicate the contribution of the six most important MOTUs (variables) to the variance explained on the first two PCs, as indicated by their direction and length. (**B**) Dendrograms obtained by hierarchical cluster analysis (HCA) of palm host species based on the AO of MOTUs using Euclidean distance and the unweighted pair group method with arithmetic mean (UPGMA) algorithm. Euclidean distance values are shown at the nodes. Coloured dots and dot-bounded bars highlight clusters (C) defined from HCA (quantitative data) and groups (G) defined according to trends in fungal biodiversity (qualitative data), and both are referred to in the right-hand columns, with the cophenetic correlation coefficient (c) noted below.

**Figure 15 jof-11-00043-f015:**
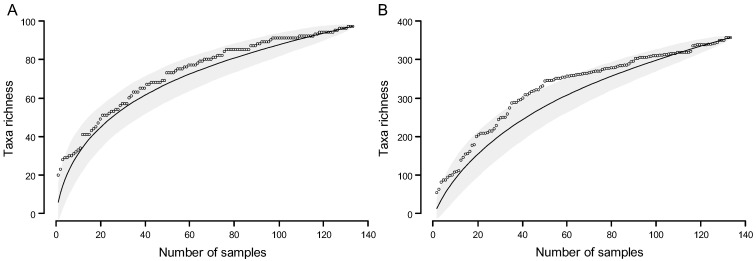
Species accumulation curves (SACs) for taxa isolated from the foliar lesions of palms. (**A**) SACs for genus richness. (**B**) SACs for molecular operational taxonomic unit (MOTU) richness. Taxon richness corresponds to the incremental increase in the number of taxa isolated plotted against the number of samples examined. Data were plotted in random order. Empty circles represent SACs, while solid lines represent the statistical expectation of the corresponding SACs (smoothed rarefaction curves) based on 1000 random resamplings, with shaded areas showing the 95% confidence intervals (CIs) of the expected (mean) taxon richness.

**Figure 16 jof-11-00043-f016:**
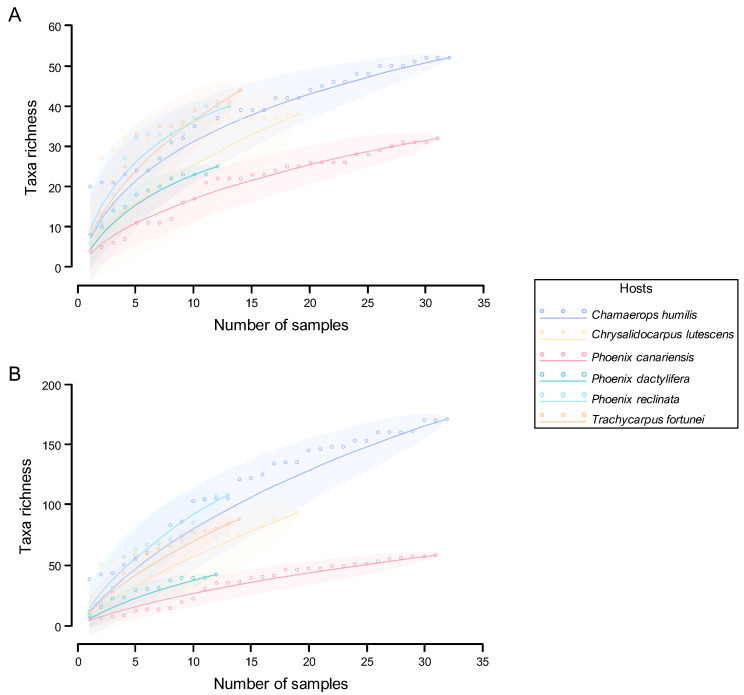
Species accumulation curves (SACs) for taxa isolated from the foliar lesions of palms by host species. (**A**) SACs for genus richness. (**B**) SACs for molecular operational taxonomic unit (MOTU) richness. Taxon richness corresponds to the incremental increase in the number of taxa isolated plotted against the number of samples examined. Data were plotted in random order. Empty circles represent SACs, while solid lines represent the statistical expectation of the corresponding SACs (smoothed rarefaction curves) based on 1000 random resamplings, with shaded areas showing the 95% confidence intervals (CIs) of expected (mean) taxon richness. Colours are according to host species and are referred to in the chart legend.

**Figure 17 jof-11-00043-f017:**
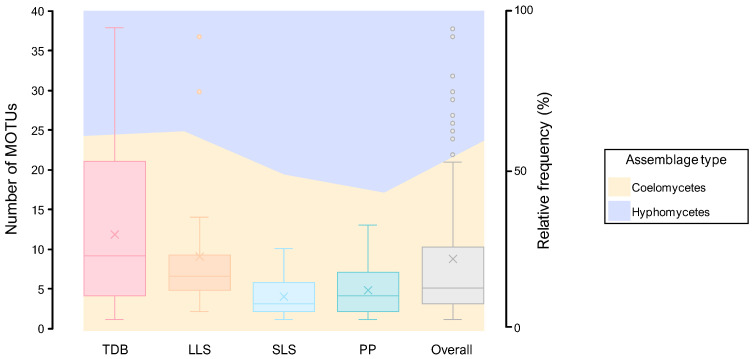
Combined boxplots and area chart of the distribution of molecular operational taxonomic units (MOTUs) and the relative frequency of assemblage types by foliar lesion type. The boxplots (left *y*-axis) represent the number of MOTUs in the four lesion types: tip die-back (TDB), large leaf spots (LLSs), small leaf spots (SLSs), and pinpoints and punctuations (PPs). A boxplot for the overall set of samples is also included. The line within each box represents the median, with the box representing the interquartile range (IQR) and the bar lines above and below the box indicating the minimum and maximum values. The mean is represented by a multiplication sign (×), while outliers are represented by circles. The area chart (right *y*-axis) displays the relative frequency of the fungal assemblage types in each foliar lesion type, with the corresponding colours referred to in the chart legend.

**Figure 18 jof-11-00043-f018:**
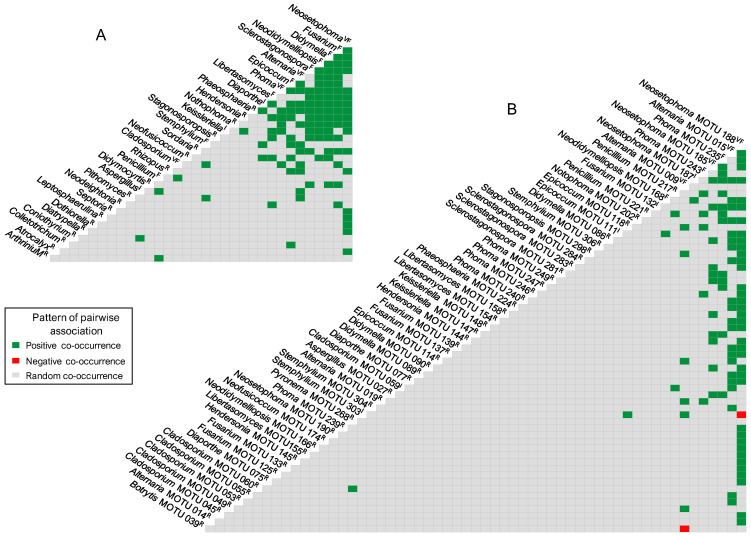
Patterns of pairwise taxon co-occurrence (CO) in the foliar lesions of palms. (**A**) Pairwise CO at the genus level. (**B**) Pairwise CO at the molecular operational taxonomic unit (MOTU) level. Each panel depicts a subset of taxa that exceeded a minimum threshold of expected CO (>1 sample). Taxon names are positioned to indicate the columns and rows that represent their pairwise relationships with other taxa. Colours are according to the pattern of pairwise association among taxa and are referred to in the chart legend. Positive CO indicates that two taxa are more likely to co-occur than expected by chance, while negative CO indicates that two taxa are less likely to co-occur than expected by chance. Random co-occurrence indicates no significant relationship in the COs of the two taxa. The frequency groups defined based on the abundance of occurrence (AO) of taxa are identified by superscript abbreviations as very frequent (^VF^), frequent (^F^), infrequent (^I^), and rare (^R^).

**Figure 19 jof-11-00043-f019:**
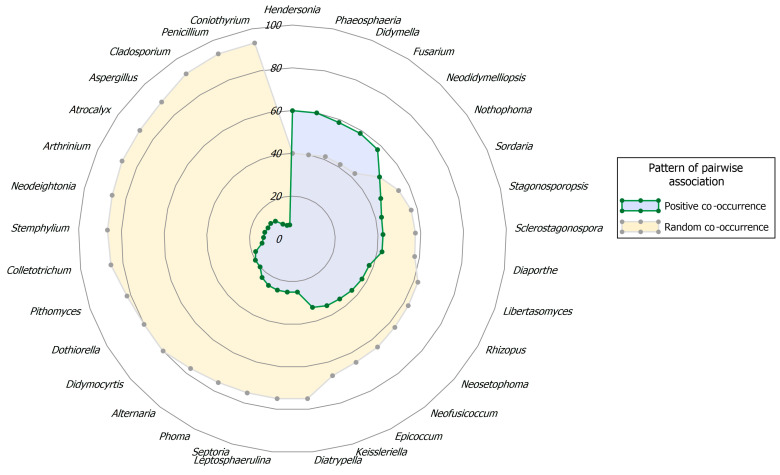
Individual contribution of genera to the positive associations. The chart illustrates the percentage of pairing for each genus that was classified as significantly positive and random. Colours are according to the pattern of pairwise associations among genera and are referred to in the chart legend. Positive co-occurrence (CO) indicates that pairwise associations for a given genus are more likely to co-occur than expected by chance, while random CO indicates no significant relationship in the COs of the pairwise associations for a given genus.

**Table 1 jof-11-00043-t001:** Collection details for the palm host species sampled.

Host	Number of Trees	Number of Samples ^1^				
		TDB	LLS	SLS	PP	Total
*Chamaedorea elegans*	2	2	0	0	0	2
*Chamaerops humilis*	23	17	2	3	10	32
*Chrysalidocarpus lutescens*	18	11	5	2	1	19
*Phoenix canariensis*	17	10	3	11	7	31
*Phoenix dactylifera*	6	2	3	3	4	12
*Phoenix reclinata*	10	6	4	2	1	13
*Phoenix roebelenii*	3	2	0	1	0	3
*Trachycarpus fortunei*	13	10	3	1	0	14
*Washingtonia filifera*	8	5	2	1	0	8
Total	100	65	22	24	23	134

^1^ TDB: tip die-back; LLS: large leaf spot; SLS: small leaf spot; PP: pinpoints and punctuations. Note: a sample corresponds to set of discrete, localised spots on diseased host leaves, which have been termed foliar lesions (see text for further explanation).

**Table 2 jof-11-00043-t002:** Distribution of fungal records by assemblage types. The figures represent the absolute frequency of fungal records isolated from the foliar lesions collected.

Assemblage Type ^1^	Number of Records	Number of Genera	Number of MOTUs ^2^
Coelomycetes	1163	47	160
Hyphomycetes	795	23	114
Ascomycetes	90	19	35
Basidiomycetes	8	7	8
Zygomycetes	8	1	3
Total	2064	97	320

^1^ Ascomycetes and basidiomycetes refer to fungal records in which no signs of asexual sporulation were observed, preventing classification as coelomycetes or hyphomycetes. In some cases, sexual morphs were observed, while in others, the mycelium remained sterile and was identified solely through molecular methods. ^2^ MOTU: molecular operational taxonomic unit.

**Table 3 jof-11-00043-t003:** Distribution of fungal genera by frequency groups. Genera are grouped based on their abundance of occurrence as *very* frequent (>5%), frequent (>2–5%), infrequent (>1–2%), or rare (≤1%).

Frequency Group/Genus ^1^	Measures of Population Status ^2^		
	AO (%)	FO (%)	S_MOTU_
Very frequent			
*Neosetophoma*	21.75	32.83	11
*Alternaria*	17.83	61.19	17
*Phoma*	8.09	49.25	17
*Cladosporium*	7.41	58.20	17
Frequent			
*Fusarium*	3.68	16.42	16
*Epicoccum*	3.54	23.88	11
*Neodidymelliopsis*	3.20	14.18	7
*Penicillium*	2.81	25.37	12
*Didymella*	2.71	14.93	12
*Libertasomyces*	2.57	16.42	7
*Stemphylium*	2.52	21.64	7
*Sclerostagonospora*	2.37	17.16	8
Infrequent			
*Diaporthe*	1.45	13.43	10
*Keissleriella*	1.31	11.94	3
*Chaetomium*	1.16	8.96	5
*Aspergillus*	1.07	12.69	6
Rare			
*Hendersonia*	0.87	6.72	4
*Phaeosphaeria*	0.78	6.72	7
*Paraconiothyrium*	0.78	4.48	5
*Sordaria*	0.63	5.22	6
*Pithomyces*	0.63	5.22	5
*Coniothyrium*	0.63	6.72	2
*Fasciatispora*	0.63	5.22	1
*Didymocyrtis*	0.58	6.72	6
*Nigrospora*	0.58	5.97	5
*Neodeightonia*	0.53	6.72	2
*Neofusicoccum*	0.48	5.97	4
*Pyronema*	0.48	7.46	2
*Nothophoma*	0.48	5.97	1
*Colletotrichum*	0.44	3.73	4
*Stagonosporopsis*	0.44	5.97	2
*Botrytis*	0.44	5.97	1
*Rhizopus*	0.39	4.48	3
*Parastagonospora*	0.39	2.99	1
*Atrocalyx*	0.34	4.48	2
*Arthrinium*	0.29	4.48	3
*Nothodactylaria*	0.29	3.73	2
*Sarocladium*	0.24	3.73	4
*Dothiorella*	0.24	2.99	3
*Xenocylindrosporium*	0.24	3.73	2
*Allophoma*	0.24	2.99	1
*Preussia*	0.19	2.99	3
*Neopestalotiopsis*	0.19	2.99	3
*Morinia*	0.19	2.99	2
*Plenodomus*	0.19	2.99	2
*Leptosphaerulina*	0.15	2.24	2
*Plicaria*	0.15	2.24	2
*Acremonium*	0.15	1.49	2
*Pseudopithomyces*	0.15	1.49	2
*Diatrypella*	0.15	2.24	1
*Dimorpha*	0.15	2.24	1
*Septoria*	0.15	2.24	1
Rare (doubletons)	0.10	1.49	1–2
*Coniochaeta*, *Ectophoma*, *Foliophoma* *, *Lecanicillium* *, *Neosulcatispora*, *Phanerochaete* *, *Scopulariopsis* *, *Tricharina* *, *Trichoderma*, *Wallemia* *			
Rare (singletons)	0.05	0.75	1
*Blastobotrys*, *Daldinia*, *Lanzia*, *Neoeutypella*, *Neurospora*, *Palmeiromyces*, *Peziza*, *Schizothecium*, *Thielavia*, *Antrodia*, *Coprinopsis*, *Coriolopsis*, *Cylindrobasidium*, *Phlebia*, *Trametes*, *Aplosporella*, *Ascochyta*, *Bartalinia*, *Botryosphaeria*, *Cryptovalsa*, *Cytospora*, *Didymosphaeria*, *Diplodia*, *Neostagonospora*, *Paraphaeosphaeria*, *Phyllosticta*, *Pseudoconiothyrium*, *Sardiniella*, *Tamaracicola*, *Wojnowiciella*, *Bipolaris*, *Harzia*, *Monilia*, *Pseudogymnoascus*, *Stachybotrys*			

^1^ Rare genera represented by doubletons (two fungal records) or singletons (one fungal record) are listed in alphabetic order. Doubletons represented by two molecular operational taxonomic units (MOTUs) are noted by a superscript asterisk (*). ^2^ AO: abundance of occurrence; FO: frequency of occurrence; SMOTU: number of MOTUs. Note: genera are listed in descending order of AO.

**Table 4 jof-11-00043-t004:** Diversity measures of hyphomycete and coelomycete assemblages.

Assemblage Type	Taxonomic Rank ^1^	Diversity Measures ^2^					
		D	H′	J′	α	DMg	DMn
Coelomycetes	Genera	0.82	1.05	0.63	9.83	6.52	1.38
	MOTU	0.92	1.60	0.73	50.25	22.53	4.69
Hyphomycetes	Genera	0.72	0.77	0.57	4.43	3.29	0.82
	MOTU	0.92	1.51	0.74	36.46	16.92	4.04

^1^ MOTU: molecular operational taxonomic unit; ^2^ D: Simpson diversity index; H′: Shannon diversity index; J′: Shannon evenness index; α: logseries alpha; DMg: Margalef species richness index; DMn: Menhinick species richness index.

**Table 5 jof-11-00043-t005:** Summary of the fit metrics for the five competing species abundance distribution (SAD) models that were used to evaluate the biodiversity patterns of the assemblage of palm leaf spotting fungi (PLSF).

Testing Method			Logseries	Neutral Model	Lognormal	Geometric Series	Broken Stick
GOF test ^1^	χ^2^ (*p*-value)		99.00 (0.24) *	99.00 (0.24) *	99.00 (0.24) *	90.75 (0.22) *	90.75 (0.22) *
	D (*p*-value)		0.18 (0.99) *	0.18 (0.99) *	0.27 (0.81) *	0.27 (0.81) *	0.27 (0.81) *
SOF test ^2^	log (L)		−755	−755.1	−771.8	−890.3	−922.3
	AIC	AICc (ΔAICc)	1512.0 (0.0)	1514.2 (2.2)	1547.6 (35.7)	1782.6 (270.7)	1844.6 (332.7)
		Weight	0.75	0.25	<0.001	<0.001	<0.001
	BIC	BIC (ΔBIC)	1515.7 (0.0)	1521.6 (5.9)	1555.1 (39.4)	1786.4 (270.7)	1844.6 (328.9)
		Weight	0.95	0.05	<0.001	<0.001	<0.001

^1^ GOF: goodness of fit; χ^2^: Pearson’s chi-squared statistic; D: Kolmogorov–Smirnov statistic. ^2^ SOF: strength of fit; log (L): natural logarithm of the maximum likelihood (log-likelihood); AIC: Akaike information criterion; AICc: AIC corrected for small sample size; ΔAIC = [AIC − min(AIC)]; BIC: Bayesian information criterion; ΔBIC = [BIC − min(BIC)]. * Model accepted by the statistical test, *p* > 0.05. Note: models are listed in descending order of the AIC and BIC values.

**Table 6 jof-11-00043-t006:** Richness and diversity measures of the communities of palm leaf spotting fungi (PLSF) by host species.

Host	Richness Measures ^1^		Diversity Measures ^2^					
	IR	MOTUs (Per Sample)	D	H′	J′	α	DMg	DMn
*Chamaerops humilis*	22.00	172 (5.38)	0.95	1.66	0.74	72.56	26.08	6.48
*Chrysalidocarpus lutescens*	12.11	94 (4.95)	0.97	1.72	0.87	59.32	17.10	6.20
*Phoenix canariensis*	4.68	58 (1.87)	0.94	1.51	0.85	35.83	11.45	4.82
*Phoenix dactylifera*	7.67	42 (3.50)	0.96	1.46	0.90	29.87	9.07	4.38
*Phoenix reclinata*	25.38	108 (8.31)	0.96	1.71	0.84	55.90	18.45	5.95
*Trachycarpus fortunei*	21.50	86 (6.14)	0.86	1.37	0.71	40.22	14.89	4.95

^1^ IR: isolation rate; MOTU: molecular operational taxonomic unit. MOTU richness per sample was computed as the number of MOTUs documented relative to the number of samples analysed for a given host species. ^2^ D: Simpson diversity index; H′: Shannon diversity index; J′: Shannon evenness index; α: logseries alpha; D_Mg_: Margalef species richness index; D_Mn_: Menhinick species richness index. All diversity measures were computed in relation to MOTUs.

**Table 7 jof-11-00043-t007:** Extrapolated taxon richness for the communities of palm leaf spotting fungi (PLSF).

Richness Estimator ^1^	Extrapolated Taxon Richness ^2^	
	Genera ± SE [95% CI]	MOTUs ± SE (95% CI)
Ŝ_Chao1_	158.22 ± 29.40 [122.07, 246.50]	505.88 ± 43.76 [437.91, 613.03]
Ŝ_ACE_	148.46 ± 17.68 [123.74, 196.03]	535.95 ± 38.12 [473.20, 624.41]
Ŝ_jk1_	131.98 ± 8.36 [119.04, 152.53]	454.94 ± 16.43 [426.38, 491.14]
Ŝ_jk2_	156.96 ± 14.48 [134.60, 192.62]	540.88 ± 28.44 [491.79, 603.99]
Ŝ_boot_	111.83 ± 3.72 [104.53, 119.125]	381.69 ± 13.51 [355.21, 408.17]
Ŝ_asymptotic_ *	103.77	324.86

^1^ Ŝ_Chao1_: Chao 1 estimator; Ŝ_ACE_: abundance-based coverage estimator; Ŝ_jk1_: first-order jackknife estimator; Ŝ_jk2_: second-order jackknife estimator; Ŝ_boot_: bootstrap estimator; Ŝ_asymptotic_: asymptote of the asymptotic curve-fitting approach estimator. ^2^ SE: standard error; CI: confidence interval; MOTU: molecular operational taxonomic unit. * The value presented for Ŝ_asymptotic_ corresponds to the asymptote of the fitted Weibull distribution.

**Table 8 jof-11-00043-t008:** Extrapolated taxon richness for the communities of palm leaf spotting fungi (PLSF) by host species.

Host Species	$\mathrm{Observed}/\mathrm{Extrapolated}\;\mathrm{Taxon}\;\mathrm{Richness}\;(\bar{x})$	
	Genera	MOTUs
*Chamaerops humilis*	52/60–81 (71)	172/212–326 (276)
*Chrysalidocarpus lutescens*	38/47–79 (65)	94//119–310 (201)
*Phoenix canariensis*	32/38–56 (50)	58/73–127 (104)
*Phoenix dactylifera*	25/29–42 (36)	42/53–106 (79)
*Phoenix reclinata*	40/46–56 (50)	108/135–213 (181)
*Trachycarpus fortunei*	44/53–92 (74)	86/111–241 (172)

Note: the ranges and mean ($\bar{x}$) of extrapolated taxon richness are based on the Chao 1 estimator, abundance-based coverage (ACE) estimator, first- and second-order jackknife estimators, and bootstrap estimator.

**Table 9 jof-11-00043-t009:** Distribution of fungal occurrences on the foliar lesions of palms. The figures represent the number of samples supporting n molecular operational taxonomic units (MOTUs) per sample.

Number of MOTUs (n) Per Sample	Break-Up of Number of Samples Examined	Foliar Lesion Type ^1^			
		TDB	LLS	SLS	PP
n = 1	11	4	0	4	3
n = 2	17	4	3	4	5
n = 3	15	7	1	5	2
n = 4	15	8	1	3	3
n = 5	12	4	5	2	1
n = 6	7	2	1	2	2
n = 7	5	2	1	1	1
n = 8	4	0	2	1	1
n = 9	9	4	3	1	1
n = 10	6	3	1	1	1
10 < n ≤ 20	14	10	2	0	2
20 < n ≤ 30	15	14	1	0	0
30 < n ≤ 40	4	3	1	0	0

^1^ TDB: tip die-back; LLS: large leaf spots; SLS: small leaf spots; PP: pinpoints and punctuations.

**Table 10 jof-11-00043-t010:** Comparison of the most frequently recorded fungal taxa associated with palm foliar lesions in Portugal and those found in fungal assemblages on tropical palms.

Taxonomic Rank	Palm Leaf Spotting Fungi (PLSF)	Tropical Palm Fungi (TrPF)		
Genus	*Alternaria* *Cladosporium* *Didymella* * *Epicoccum* *Fusarium* *Libertasomyces* * *Neodidymelliopsis* * *Neosetophoma* * *Penicillium* *Phoma* *Sclerostagonospora* * *Stemphylium*	*Acremonium-like taxa* *Anthostomella* *Apioclypea* *Apiospora* *Appendicospora* *Arecomyces* *Astrocystis* *Brunneiapiospora* *Capsulospora* *Cocoicola* *Endocalyx* *Fasciatispora*	*Frondispora* *Hypoxylon* *Lachnum* *Leptosporella* *Linocarpon* *Myelosperma* *Nectria-like taxa* *Neolinocarpon* *Oxydothis* *Palmicola* *Pemphidium* *Phaeochora*	*Phaeochoropsis* *Phomatospora* *Serenomyces* *Sorokinella* *Trematosphaeria* *Xylaria*
Family	*Aspergillaceae* *Cladosporiaceae* *Didymellaceae* * *Nectriaceae* *Phaeosphaeriaceae* * *Pleosporaceae*	*Apiosporaceae* *Appendicosporaceae* *Astrosphaeriellaceae* *Cainiaceae* *Clypeosphaeriaceae* *Fasciatisporaceae*	*Leptosporellaceae* *Linocarpaceae* *Oxydothidaceae* *Phaeochoraceae* *Phomatosporaceae* *Xylariaceae*	
Order	*Cladosporiales* *Eurotiales* *Hypocreales* *Pleosporales* *	*Amphisphaeriales* *Chaetosphaeriales* *Phomatosporales* *Phyllachorales* *Xylariales*		
Class	*Dothideomycetes*	*Sordariomycetes*		
Phylum	*Ascomycota*	*Ascomycota*		

* Taxa that appear to have a higher prevalence in association with palm foliar lesions than previously known, which are addressed here as components of an assemblage of potential typical temperate palm fungi (TePF). Note: many TrPF commonly associated with palms have been excluded from this table for simplification. For a more comprehensive overview of the typical assemblage of palm fungi, refer to Pereira and Phillips [45]. The PLSF genera listed here include only those categorised as very frequent and frequent mycota (see Section 3.1 for further details).

## Data Availability

All data generated and analysed in this study are included in this article and its Appendix A.

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
