# Peer review of "Exploring the Diversity and Ecological Dynamics of Palm Leaf Spotting Fungi—A Case Study on Ornamental Palms in Portugal"

_jof, 2025, doi:10.3390/jof11010043_

Round 1
Reviewer 1 Report
The authors have chosen a very important and timely research topic—exploring the diversity and ecological dynamics of palm leaf spotting fungi (PLSF). Considering the academic value and practical significance of this study, the paper is suitable for publication in the Journal of Fungi. However, the following points need to be addressed:
1. Are there non-culturable fungi in this study? How does MSP-PCR handle fungi that cannot be cultured?
2. Sampling location coordinates were not provided; it is recommended to include georeferencing points.
3. Lines 2657–2661 mention the clarification of the complex interactions of climate, host, and ecological factors affecting PLSF communities. It is suggested to revise this section to align with the overall focus and findings of the article.
4. It is recommended to indicate whether the molecular identification data have been deposited in a public database (e.g., GenBank) to enhance transparency.
5. Where do the data on typical fungal communities from tropical palms come from?
The authors have chosen a very important and timely research topic—exploring the diversity and ecological dynamics of palm leaf spotting fungi (PLSF). Considering the academic value and practical significance of this study, the paper is suitable for publication in the Journal of Fungi. However, the following points need to be addressed:
1. Are there non-culturable fungi in this study? How does MSP-PCR handle fungi that cannot be cultured?
2. Sampling location coordinates were not provided; it is recommended to include georeferencing points.
3. Lines 2657–2661 mention the clarification of the complex interactions of climate, host, and ecological factors affecting PLSF communities. It is suggested to revise this section to align with the overall focus and findings of the article.
4. It is recommended to indicate whether the molecular identification data have been deposited in a public database (e.g., GenBank) to enhance transparency.
5. Where do the data on typical fungal communities from tropical palms come from?
Author Response
The authors have chosen a very important and timely research topic—exploring the diversity and ecological dynamics of palm leaf spotting fungi (PLSF). Considering the academic value and practical significance of this study, the paper is suitable for publication in the Journal of Fungi. However, the following points need to be addressed:
We would like to express our gratitude for the time and effort invested in the review of our manuscript. Your comments have undoubtedly contributed to the enhancement of the manuscript.
- Are there non-culturable fungi in this study? How does MSP-PCR handle fungi that cannot be cultured?
Response: Thank you for your questions. The subject in question is thoroughly expounded upon in the M&M section. Within the section entitled "Fungal Isolation", the following assertion is made: "A comprehensive isolation flowchart (Figure 3) was employed to capture the maximum culturable diversity of palm leaf spotting fungi (PLSF) present in each sample" (lines 191–193). It is therefore evident that the focus has been on the consideration of only culturable fungi. Furthermore, the discussion section explicitly advocates that the study is only preliminary, and that next-generation sequencing methodologies are required to reveal the undetected mycota, which may include non-culturable fungi, as well as many others that may have been overlooked.
- Sampling location coordinates were not provided; it is recommended to include georeferencing points.
Response: We appreciate your comment. You are right, this is something we missed in our collections as they were done randomly in Lisbon and Oeiras, Portugal. Unfortunately, we do not have this information, but we do not believe that it compromises the study and its conclusions.
- Lines 2657–2661 mention the clarification of the complex interactions of climate, host, and ecological factors affecting PLSF communities. It is suggested to revise this section to align with the overall focus and findings of the article.
Response: Thank you for your comment. Many of the subsections in the discussion were devoted to discussing all of the points contained in that sentence. In short, we have reflected extensively on how the differences between typical tropical palm fungi and the fungi detected in our study (in a temperate context), as well as differences that have long been discussed in the literature [e.g. Journal of Biogeography 2000, 27 (2), 297-310]. Similarly, the differences in communities found in six different hosts have been extensively detailed, indicating the influence of host species on PLSF communities. Several aspects of the potential ecology of these communities have been addressed, from climate (abiotic factors) to statistically significant positive and negative interactions between pairs of taxa (biotic factors).
- It is recommended to indicate whether the molecular identification data have been deposited in a public database (e.g., GenBank) to enhance transparency.
hank you for your comment. We have not deposited our ITS sequences in public databases as we consider our identifications to be tentative/presumptive (line 360 in section 2.5) and they were only used as a proxy to confirm our morphological examination to identify the isolates to genus level. For the sake of consistency, we have also used this method for isolates that we have already identified to species level and that are currently published. For example, Morinia and Bartalinia isolates have been identified to species level and published previously [see Mycological Progress 2021, 20 (2), 83-94]. However, in the present study, these isolates were subjected to the same methodology as all others, so that the final results would not be biased. Many of the isolates discussed here will be fully identified by phylogenetic means in future studies and published accordingly.
- Where do the data on typical fungal communities from tropical palms come from?
Response: Thank you for your question. All the information provided has been referenced throughout the discussion. In the notes section of Table 10, which compares our results with typical tropical palm fungi, we state “For a more comprehensive overview of the typical assemblage of palm fungi, refer to Pereira & Phillips [45]”. Thus, although there is an enormous amount of work on palm fungi, we have recently published a review in the Journal of Fungi that presents a comprehensive compilation of the taxonomic structure of palm fungi [see Journal of Fungi 2023, 9 (11), 1121].
Reviewer 2 Report
The manuscript titled “ Exploring the diversity and ecological dynamics of palm leaf spotting fungi – a case study on ornamental palms in Portugal” is dedicated to research on fungi associated with the Arecaceae family and affecting ornamental palms.
The present study conducted a preliminary assessment of the diversity and ecology of potential phytopathogenic fungi associated with foliar lesions on various ornamental palm species in Portugal. This was done through morphological examination, PCR-based genomic fingerprinting, and biodiversity data analysis.
A total of 134 foliar lesions were sampled from 100 palm trees, resulting in a collection of 2,064 palm leaf spotting fungi. These fungi represent a diverse fungal assemblage consisting of 320 molecular operational taxonomic units (MOTUs) across 97 genera.
Overall, the fungal community composition revealed a distinct assemblage dominated by Neosetophoma, Alternaria, Phoma, and Cladosporium. There were also numerous infrequent and rare taxa present, consistent with a logarithmic distribution.. Positive co-occurrence patterns among both common and uncommon taxa suggest the potential for synergistic interactions that enhance fungal colonisation, persistence, and pathogenicity. The taxonomic structure of the population differs markedly from that of tropical palm fungi, particularly in the prevalence of Pleosporalean coelomycetes from the Didymellaceae and Phaeosphaeriaceae families, including recently introduced or undocumented genera on Arecaceae plants. This novel community suggests that climatic conditions shape the structure of palm fungal communities, leading to distinct temperate and tropical assemblages. Additionally, fungal communities vary significantly across palm host species, with native temperate palms hosting more diverse and coelomycete-enriched communities. These findings highlight the importance of foliar lesions as highly diverse microhabitats that support intricate fungal interactions, influenced by climatic, host, and ecological factors.. With climate change affecting environmental conditions, it is crucial to identify fungi that thrive or inhabit these microhabitats in order to predict shifts in pathogen dynamics and mitigate future fungal disease outbreaks. Understanding these complex ecological interactions is essential for identifying potential threats to ornamental plant health and developing effective management strategies. The quality of research is high, the manuscript is large, carefully written, and can be published after cleaning some mistyping and text errors, including :
1) Line 74 Only a few fungal pathogens are lethal to palms, with most causing minor damage that can be easily managed” and Line 75 “Common lethal or potentially lethal fungal diseases in ornamental palms include Fusarium wilt (caused mainly by Fusarium oxysporum)…”
Note – "Lethal" factor is mostly applied to animals as a factor causing rapid death. The word "deadly" is more appropriate for plants.
2) Line 540 “hierarchical cluster analyses “ and Line 541 “Principal component analysis (PCA)”. Several times hierarchical cluster analyses (HCA) and Principal component analysis (PCA) are typed as abbreviation and several times – as both full name and abbreviation. Please, carefully check all abbreviations used in the manuscript. The full name followed by the abbreviation should be used the first time it appears in the text, and the abbreviation alone should be used thereafter, except in the legends for figures and tables.
The quality of research is high, the manuscript is large, carefully written, and can be published after cleaning some mistyping and text errors, including :
1) Line 74 Only a few fungal pathogens are lethal to palms, with most causing minor damage that can be easily managed” and Line 75 “Common lethal or potentially lethal fungal diseases in ornamental palms include Fusarium wilt (caused mainly by Fusarium oxysporum)…”
Note – "Lethal" factor is mostly applied to animals as a factor causing rapid death. The word "deadly" is more appropriate for plants.
2) Line 540 “hierarchical cluster analyses “ and Line 541 “Principal component analysis (PCA)”. Several times hierarchical cluster analyses (HCA) and Principal component analysis (PCA) are typed as abbreviation and several times – as both full name and abbreviation. Please, carefully check all abbreviations used in the manuscript. The full name followed by the abbreviation should be used the first time it appears in the text, and the abbreviation alone should be used thereafter, except in the legends for figures and tables
Author Response
The manuscript titled “ Exploring the diversity and ecological dynamics of palm leaf spotting fungi – a case study on ornamental palms in Portugal” is dedicated to research on fungi associated with the Arecaceae family and affecting ornamental palms.
We appreciate the time and effort you have taken to review our manuscript. Your comments have certainly helped us to improve the manuscript.
The present study conducted a preliminary assessment of the diversity and ecology of potential phytopathogenic fungi associated with foliar lesions on various ornamental palm species in Portugal. This was done through morphological examination, PCR-based genomic fingerprinting, and biodiversity data analysis.
A total of 134 foliar lesions were sampled from 100 palm trees, resulting in a collection of 2,064 palm leaf spotting fungi. These fungi represent a diverse fungal assemblage consisting of 320 molecular operational taxonomic units (MOTUs) across 97 genera.
Overall, the fungal community composition revealed a distinct assemblage dominated by Neosetophoma, Alternaria, Phoma, and Cladosporium. There were also numerous infrequent and rare taxa present, consistent with a logarithmic distribution.. Positive co-occurrence patterns among both common and uncommon taxa suggest the potential for synergistic interactions that enhance fungal colonisation, persistence, and pathogenicity. The taxonomic structure of the population differs markedly from that of tropical palm fungi, particularly in the prevalence of Pleosporalean coelomycetes from the Didymellaceae and Phaeosphaeriaceae families, including recently introduced or undocumented genera on Arecaceae plants. This novel community suggests that climatic conditions shape the structure of palm fungal communities, leading to distinct temperate and tropical assemblages. Additionally, fungal communities vary significantly across palm host species, with native temperate palms hosting more diverse and coelomycete-enriched communities. These findings highlight the importance of foliar lesions as highly diverse microhabitats that support intricate fungal interactions, influenced by climatic, host, and ecological factors.. With climate change affecting environmental conditions, it is crucial to identify fungi that thrive or inhabit these microhabitats in order to predict shifts in pathogen dynamics and mitigate future fungal disease outbreaks. Understanding these complex ecological interactions is essential for identifying potential threats to ornamental plant health and developing effective management strategies. The quality of research is high, the manuscript is large, carefully written, and can be published after cleaning some mistyping and text errors, including :
1) Line 74 Only a few fungal pathogens are lethal to palms, with most causing minor damage that can be easily managed” and Line 75 “Common lethal or potentially lethal fungal diseases in ornamental palms include Fusarium wilt (caused mainly by Fusarium oxysporum)…”
Note – "Lethal" factor is mostly applied to animals as a factor causing rapid death. The word "deadly" is more appropriate for plants.
Response: Thank you very much for this correction. The correction has been done (lines 72, 74, 75 and 76).
2) Line 540 “hierarchical cluster analyses “ and Line 541 “Principal component analysis (PCA)”. Several times hierarchical cluster analyses (HCA) and Principal component analysis (PCA) are typed as abbreviation and several times – as both full name and abbreviation. Please, carefully check all abbreviations used in the manuscript. The full name followed by the abbreviation should be used the first time it appears in the text, and the abbreviation alone should be used thereafter, except in the legends for figures and tables.
Response: Thank you for your comment. Considering the number of pages in the paper, we have used the full name followed by the abbreviation in each section (we mean Introduction, M&M, Results and Discussion). There is no real consensus on how to do this. Some people define acronyms and abbreviations once (when they first appear), while others repeat them in each section. In our understanding, we tend to restate acronyms after a significant gap in their use, which in this paper happens between each section, to facilitate the reader, as they are quite large. If there is a rule in the Journal of Fungi that states otherwise, we will be happy to format the paper accordingly. One option would be to provide a list of abbreviations of acronyms at the beginning of the paper, but we do not know if this is an option for Journal of Fungi. We really appreciate your comment and will contact the editor about this issue.